# High-resolution genomic history of early medieval Europe

Leo Speidel[1,2,3✉], Marina Silva[1], Thomas Booth[1], Ben Raffield[4], Kyriaki Anastasiadou[1], Christopher Barrington[5], Anders Götherström[6,7], Peter Heather[8] & Pontus Skoglund[1✉]

Many known and unknown historical events have remained below detection thresholds of genetic studies because subtle ancestry changes are challenging to reconstruct. Methods based on shared haplotypes[1,2] and rare variants[3,4] improve power but are not explicitly temporal and have not been possible to adopt in unbiased ancestry models. Here we develop Twigstats, an approach of time-stratified ancestry analysis that can improve statistical power by an order of magnitude by focusing on coalescences in recent times, while remaining unbiased by population-specific drift. We apply this framework to 1,556 available ancient whole genomes from Europe in the historical period. We are able to model individual-level ancestry using preceding genomes to provide high resolution. During the first half of the first millennium CE, we observe at least two different streams of Scandinavian-related ancestry expanding across western, central and eastern Europe. By contrast, during the second half of the first millennium CE, ancestry patterns suggest the regional disappearance or substantial admixture of these ancestries. In Scandinavia, we document a major ancestry influx by approximately 800 CE, when a large proportion of Viking Age individuals carried ancestry from groups related to central Europe not seen in individuals from the early Iron Age. Our findings suggest that time-stratified ancestry analysis can provide a higher-resolution lens for genetic history.

Ancient genome sequencing has revolutionized our ability to reconstruct expansions, migrations and admixture events in the ancient past and understand their impact on human genetic variation today. However, tracing history using genetic ancestry has remained challenging, particularly in historical periods for which the richest comparative information from history and archaeology often exists. This is because ancestries in many geographical regions are often so similar as to be statistically indistinguishable with current approaches. One example is northern and central Europe since the start of the Iron Age around 500 BCE, a period for which many long-standing questions remain, such as the nature of large-scale patterns of human migration during the fourth to sixth centuries CE, their impact on the Mediterranean world and later patterns of human mobility during the Viking Age (around 750–1050 CE).

Several recent studies have documented substantial mobility and genetic diversity in these time periods, suggesting stable population structure despite high mobility[5], and have revealed genetic variation in Viking Age Scandinavia[6–8], early medieval England[3,9], early medieval Hungary[10,11] and Iron Age and medieval Poland[12]. However, previous studies mostly used large modern cohorts to study ancestry change through time and space. This is because the differentiation between Iron Age groups in central and northern Europe is an order of magnitude lower (fixation index ($F_{ST}$) = 0.1–0.7%; Extended Data Fig. 1) than, for example, the more commonly studied hunter-gatherer, early farmer and steppe-pastoralist groups that shaped the ancestry landscape of Stone Age and Bronze Age Europe[13–16] ($F_{ST}$ = 5–9% (refs. 13,17)). Modern populations provide more power to detect differences, but their genetic affinity to ancient individuals may be confounded by later gene flow, that is, after the time of the ancient individual(s)[18]. The most principled approach is thus to build ancestry models in which source and 'outgroup/reference' populations are older than, or at least contemporary with, the target genome or group that we are trying to model[18]. However, this has been challenging, due to the limited statistical power offered by the thousands-fold lower sample sizes and reduced sequence quality of ancient genomes.

Reconstructing genetic histories and ancestry models from ancient DNA (aDNA) data commonly uses methods based on $f$-statistics[13,19–22]. Their popularity is rooted in a number of favourable properties, such as enabling analyses of lower-quality aDNA data, relative robustness to ascertainment and theoretical guarantees of unbiasedness, including in the presence of population bottlenecks[21,23]. Approaches derived from $f$-statistics, such as qpAdm[13], are close to unique in enabling the unbiased fitting of admixture models, including identifying the number of such events and the closest representatives of sources[13,14,23]. However, $f$-statistics have not always had sufficient power to reconstruct events that involve closely related ancestries, despite increasing sample sizes[6,24]. Methods that identify haplotypes, or shared segments of DNA that are not broken down by recombination, have previously been shown to have more power than those using individual

[1]Ancient Genomics Laboratory, Francis Crick Institute, London, UK. [2]Genetics Institute, University College London, London, UK. [3]iTHEMS, RIKEN, Wako, Japan. [4]Department of Archaeology and Ancient History, Uppsala University, Uppsala, Sweden. [5]Bioinformatics and Biostatistics, Francis Crick Institute, London, UK. [6]Centre for Palaeogenetics, Stockholm University, Stockholm, Sweden. [7]Department of Archaeology and Classical Studies, Stockholm University, Stockholm, Sweden. [8]Department of History, King's College London, London, UK. ✉e-mail: leo.speidel@riken.jp; pontus.skoglund@crick.ac.uk

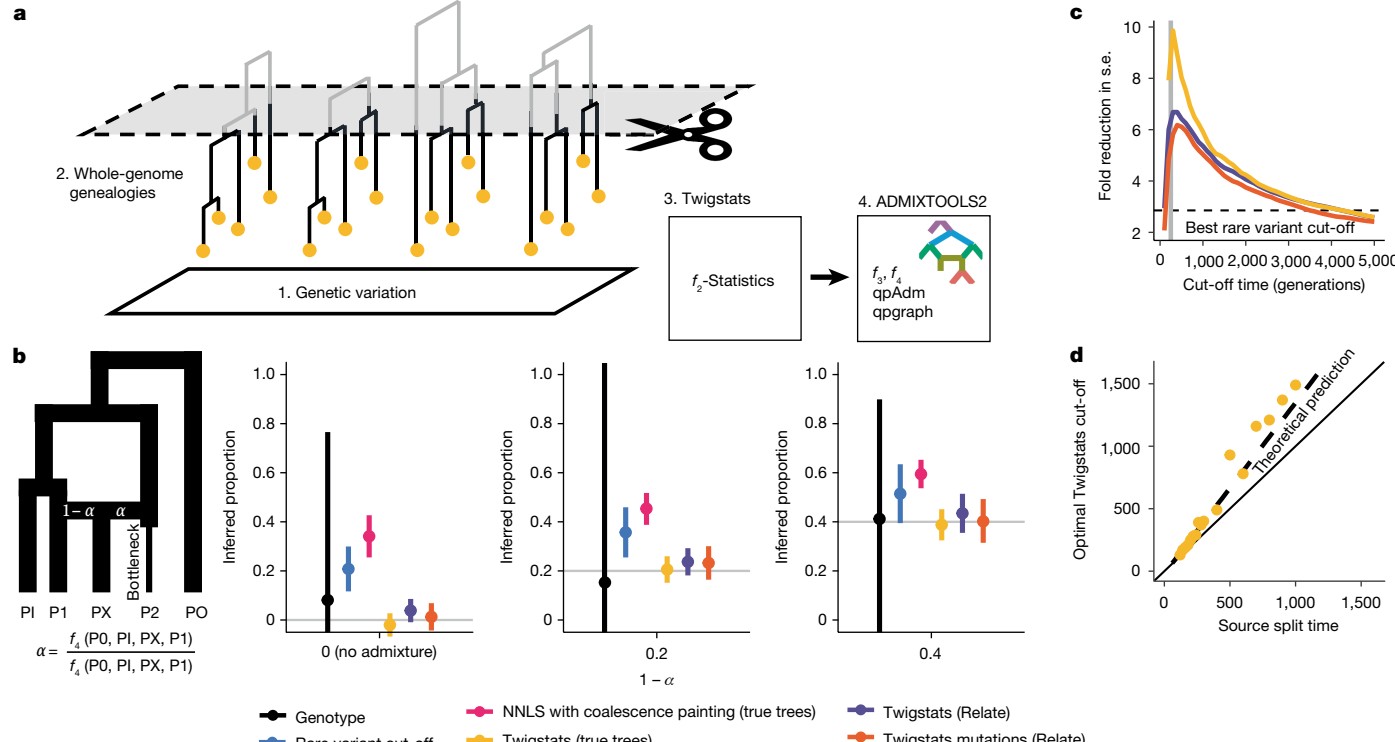

**Fig. 1 | Twigstats performance on simulated data. a**, A diagram of the Twigstats approach. We first construct genealogies from genetic variation data and then use Twigstats to compute $f_2$-statistics between pairs of groups to be used by ADMIXTOOLS2. **b**, Admixture proportions inferred from an $f_4$-ratio statistic or non-negative least squares method. Source groups P1 and P2 split 250 generations ago and mix 50 generations ago, where P2 contributes proportion $\alpha$ and P1 contributes $1 - \alpha$. Effective population sizes are equal and constant except for a recent bottleneck in P2 (see Methods for simulation details). The Twigstats cut-off is set to 500 generations, the rare variant cut-off is set to 5%, and we additionally infer admixture proportions by generating 'first coalescence profiles' for each population and modelling PX as a mixture

of sources P1 and P2 using non-negative least squares (NNLS) (Methods). We sample 20 haploid sequences from each population. Data are mean ± 2 s.e. around the point estimate. **c**, The fold improvement of s.e. relative to the genotype case as a function of the Twigstats cut-off time, for the same simulation as in **b** and averaged across different true admixture proportions. The dashed line shows the best fold improvement of s.e. when ascertaining genotypes by frequency, when evaluated at different frequency cut-offs. **d**, The optimal Twigstats cut-off, defined as the largest reduction in s.e. relative to the genotype case, as a function of source split time in simulations using true trees. The dashed line indicates our theoretical prediction (Supplementary Note).

single-nucleotide polymorphism (SNP) markers, but this information has not been accessible in combination with the advantages of $f$-statistics[2,6,25,26]. Furthermore, the overwhelming majority of available aDNA is from a panel of 1.2 million SNPs[27], and few clear advantages have been demonstrated for analysis of the more than 50 million SNPs available with whole-genome shotgun data.

One class of methods that use haplotype information is full genealogical tree inference[28,29], which can now readily be applied to many thousands of modern and ancient whole genomes[30–35]. Such methods have been successfully applied to boost the detection of positive selection[32,36–38], population structure[31,33,35,39], geographical locations of ancestors[34,40], demography[31,32] and mutation rate changes[31]. Genealogical trees can be thought of as containing essentially full, time-resolved information about genetic ancestry, including information typically captured by recent haplotype sharing or identity by descent. Genetic ancestry here refers to the full collection of genetic ancestors of individuals[41], and genealogical trees reveal how and when these are shared across individuals. By contrast, rare variant ascertainment, haplotypes or chromosome blocks can be thought of as subsets or summaries of the information available in genealogies.

Here, we propose an approach that we refer to as 'time-stratified ancestry analysis' to boost the statistical power of $f$-statistics several-fold by using inferred genome-wide genealogies (Fig. 1a) and apply our method to reconstruct the genetic history of northern and central Europe from around 500 BCE to 1000 CE.

## Genealogies improve ancestry modelling

By definition, $f$-statistics count the occurrence of local genealogical relationships that are implied by how mutations are shared between individuals[42]. This inherent relationship between $f$-statistics and local genealogies makes it straightforward to compute $f$-statistics directly on inferred genealogies[43]. Instead of computing $f$-statistics on observed mutations, they are now calculated on the inferred branches of these genealogies, some of which may not be directly tagged by mutations but are inferred by resolving the local haplotype structure (Methods).

We develop mathematical theory and simulate a simple admixture model, in which the ancestry proportion is constrained in a single ratio of two $f_4$-statistics[19], to test this approach (Fig. 1b and Supplementary Note). While unbiased, we find that using $f$-statistics computed on genealogies by itself does not yet yield a large improvement in statistical power to quantify admixture events. However, we show, through both theoretical prediction and simulation, that large improvements in power can be gained without bias by restricting to recent coalescences, which are most informative for recent admixture events (Fig. 1c,d and Extended Data Figs. 2 and 3). We show that coalescences older than the time of divergence of the sources carry no information with respect to the admixture event and only add noise to the $f$-statistics. Excluding these therefore increases statistical power, without introducing bias, in principle.

We implement this idea of studying the 'twigs' of gene trees in a tool, Twigstats (Fig. 1a and Methods), which we demonstrate in simulations

reduces standard errors (s.e.) by up to tenfold and potentially more, depending on sample sizes and details of the genetic history model. The approach does not produce detectable bias in estimates of admixture proportions (Fig. 1b–d and Extended Data Fig. 3). Furthermore, we demonstrate that computing $f$-statistics on genotypes ascertained for young mutation ages produce a power gain nearly equal to that produced when using full genealogies in many examples, while adding flexibility by allowing lower-quality genomes to be grafted onto a genealogy reconstructed with higher-quality genomes[31].

We further confirm with simulations that genealogy-based $f$-statistics estimates are robust to sequencing and phase-switch errors of expected magnitude (Extended Data Fig. 3b). In fact, although sequence errors can affect SNP-based population-genetic approaches substantially, errors can be 'corrected' in genealogies as they take all variants in a region into account[32].

Previous studies have suggested ascertaining rare mutations as a proxy for recent history[3,4], but we show that this approach is prone to bias when effective population sizes vary between populations, and that using full time-restricted genealogies is both unbiased and more powerful (Fig. 1b and Extended Data Fig. 3). We attribute this to the observation that mutation age is not fully predictive of allele frequency (Extended Data Fig. 4) and that the genealogy-based approach gains power from the inclusion also of higher-frequency young mutations that 'tag' recent coalescences by closely pre-dating them. We demonstrate that a widely used 'chromosome painting' approach, and any conceptually similar modelling based on identity by descent, that finds the nearest neighbours between chromosomal segments in a sample and model groups using a non-negative least squares of genome-wide painting profiles[2] is also prone to bias, when source groups have undergone strong drift since the admixture event (Fig. 1b and Extended Data Fig. 3b).

We next test the Twigstats time-restricted genealogy approach on a range of empirical examples. First, we boost pairwise outgroup $f_3$-statistics[44] to quantify fine-scale population structure; we demonstrate this improvement using a previously proposed simulation[39] (Extended Data Fig. 5a). When applied to published genomes from Neolithic Europe (Methods and Supplementary Table 1), we can replicate the previously suggested fine-scale structure between individuals buried in megalithic structures in Ireland compared with others[45], a relationship that is not apparent from SNP data alone (Extended Data Fig. 5b). For the well-studied example of three major ancestries contributing to prehistoric Europe, that is, Mesolithic hunter-gatherers, early farmers and steppe populations[13–16], we obtain unbiased estimates and an approximately 20% improvement in standard errors in an already well-powered qpAdm model[46] (Extended Data Fig. 5c).

Finally, we demonstrate that Twigstats can be used to resolve competing models of punctual admixture and long-standing gene flow, or constrain the time of admixture. For instance, it has previously been suggested that long-standing deep structure and gene flow between Neanderthals and early modern humans in Africa may produce genetic patterns that resemble a punctual admixture event some 60,000 years ago[47–49], casting doubt on the model of Neanderthal admixture into ancestors of Eurasians[49–51]. However, whereas such long-standing deep substructure would confound SNP-based $f$-statistics to produce patterns similar to Neanderthal admixture, we demonstrate, in simulations, that Twigstats can clearly distinguish this history from recent admixture (Extended Data Fig. 5d). Application of Twigstats on empirical whole genomes produces results inconsistent with deep substructure alone, but consistent with punctual admixture.

## Ancestry models of early medieval Europe

Having demonstrated that the Twigstats approach can effectively improve resolution and statistical power to test ancestry models and estimate proportions, we turn to the history of early medieval Europe.

In the first half of the first millennium CE, Roman historians such as Tacitus and Ammianus Marcellinus described the geographical distribution and movements of groups beyond the imperial frontier and suggested a potential role for them in the fall of the western Roman Empire[52]. However, the exact nature and scale of these historically attested demographic phenomena—and their genetic impact—have been questioned[53], and have been difficult to test with genetic approaches owing to the close relations shared between many groups that were ostensibly involved. Less is understood at further distances from the Roman frontier owing to a lack of historical accounts. The improved statistical power of time-restricted ancestry in Twigstats thus offers an opportunity to revisit these questions.

To develop an ancestry model for early medieval individuals (Supplementary Table 1), we first need a broad characterization of the ancestry of the earlier sources from the early Iron Age (EIA) and Roman periods. We use hierarchical UPGMA clustering based on pairwise clade testing between all individuals, and formally test the cladality of proposed ancestry groups with qpWave[5] (cladality in this sense means whether they are consistent with being symmetrically related to all other tested groups; Methods). This resulted in a set of model ancestry sources that included Iron Age and Roman Britain ($n = 11$), the Iron Age of central European regions of mostly Germany, Austria and France ($n = 10$), Roman Portugal ($n = 4$), Roman Italy ($n = 10$), Iron Age Lithuania ($n = 5$), the EIA Scandinavian Peninsula (Sweden and Norway, $n = 10$) and several other more eastern groups dating to the Bronze Age and EIA ($n = 25$) (Fig. 2a and Extended Data Fig. 1). We then use a rotational qpAdm approach[54] to narrow down the set of contributing sources from this larger pool of putative sources.

We additionally perform non-parametric multidimensional scaling (MDS) on outgroup-$f_3$ statistics[44] computed using Twigstats, the results of which do not depend on any modelling assumptions and which show increased resolution compared with conventional outgroup-$f_3$ statistics (Fig. 2a,b, Extended Data Fig. 6 and Supplementary Table 2). Encouragingly, the MDS model supports regional fine-scale genetic structures reflected in our source groups, such as the separation of predominantly Norwegian and northern Swedish EIA individuals from southern Peninsular Scandinavia (Fig. 2a); this relationship is not detected without Twigstats. In this MDS analysis, we note a close affinity of wide-ranging individuals from Portugal, France, Germany, Austria and Britain. We hypothesize that this corresponds to areas associated with the Celtic-speaking world, and that their close genetic affinity is due to earlier expansions. Sparse sampling limits our understanding of the full extent of regional ancestry variation in central Europe and some other regions, but the continental ancestries differentiated in the MDS model suggests that major ancestry variation across Europe in this period is relatively well captured.

## Expansions of Scandinavian-like ancestry

We assembled time transects using available aDNA data across several geographical regions in Europe, and infer their ancestry using a model with the EIA or Roman Iron Age sources previously defined (shown in Fig. 2a). Our modelling provides direct evidence of individuals with ancestry originating in northern Germany or Scandinavia appearing across Europe as early as the first century CE (Figs. 2b,c and 3 and Supplementary Table 3).

In the region of present-day Poland, our analysis suggests several clear shifts in ancestry. First, in the Middle to Late Bronze Age (1500 BCE to 1000 BCE), we observe a clear shift away from preceding ancestry originally associated with Corded Ware cultures[55] (Fig. 3a). Second, in the first to fifth century CE, individuals associated with Wielbark culture[5,12] show an additional strong shift away from the preceding Bronze Age groups, and can only be modelled with a >75% component attributed to the EIA Scandinavian Peninsula. Multiple individuals, especially from earlier Wielbark cemeteries, have approximately 100%

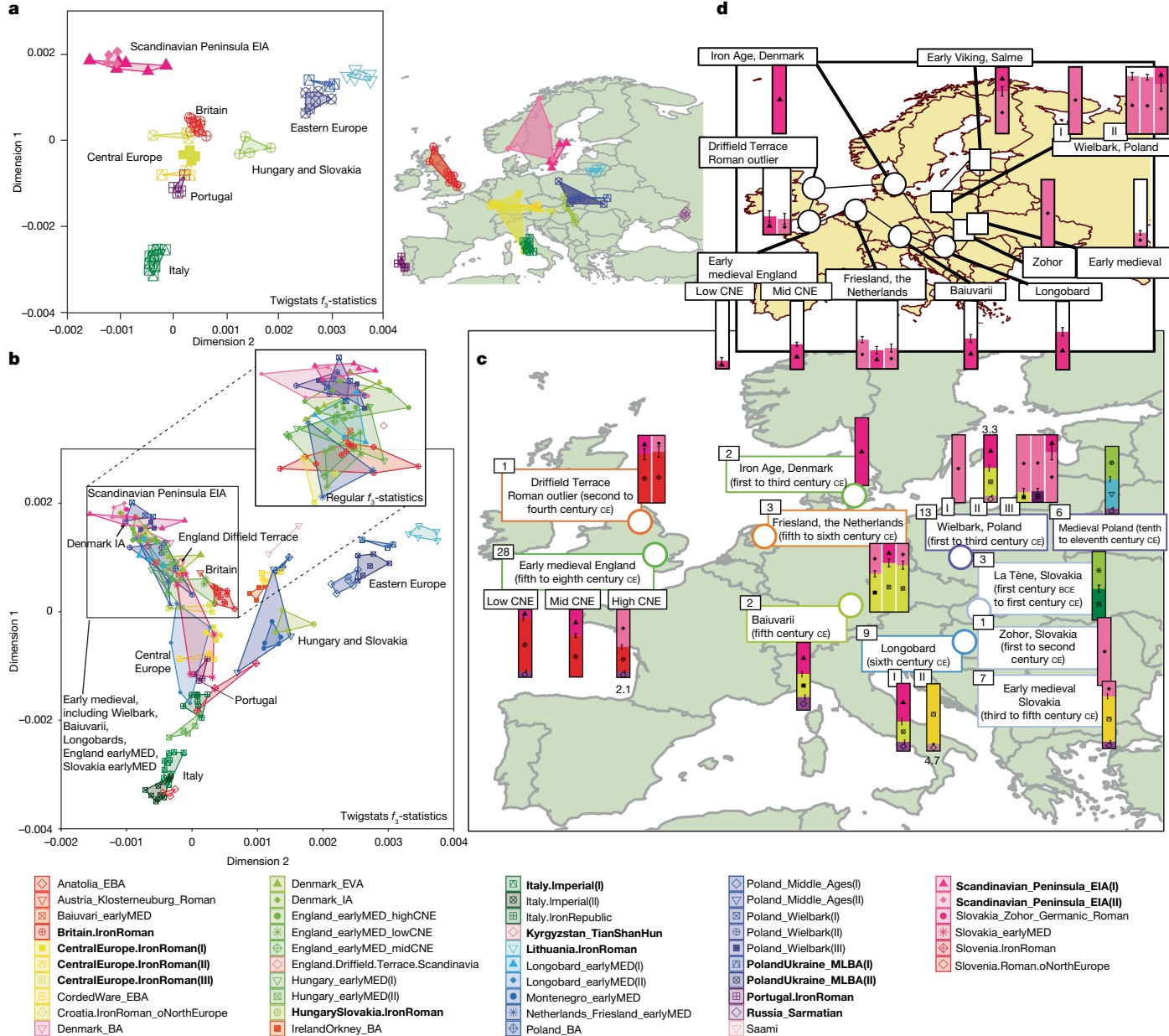

**Fig. 2 | Ancestry from the Iron Age to the early medieval period in Europe.**
**a**, Source groups used for qpAdm modelling of early medieval Europe. MDS is computed jointly with individuals from later periods using pairwise outgroup $f_3$ statistics (outgroup: Han Chinese people). These are calculated using Twigstats on Relate genealogies with a cut-off of 1,000 generations. The geographical map shows sampling locations of these individuals. **b**, The genetic structure of ancient groups predominantly from early medieval contexts shown on the same MDS as in **a**. The magnified inset shows an MDS computed without Twigstats on the same samples as the Twigstats MDS and focusing on early medieval or later individuals. **c**, Ancestry models of early medieval (EM) groups across Europe computed using qpAdm. Sample sizes are shown in black boxes. Sources are highlighted in **a** and marked as bold in the key, and were used in a rotational qpAdm scheme. For each target group, we remove models with infeasible admixture proportions (falling outside [0, 1]) and use a Twigstats cut-off of 1,000 generations. All models satisfy $P > 0.01$, unless a $-\log_{10}[P\text{ value}]$ is shown next to the model. If models satisfy $P > 0.05$, we show all such models; otherwise, we show only the model with the largest $P$ value. **d**, The ancestry proportion derived from EIA Scandinavia in groups with a non-zero component of this ancestry. We show groups modelled in **c** that have a feasible model ($P > 0.01$). In **c**,**d**, we show one s.e. BA, Bronze Age; CNE, continental northern Europeans; EBA, early Bronze Age; EVA, early Viking Age; IA, Iron Age; MED, medieval; MLBA, middle/late Bronze Age; VA, Viking Age.

ancestry related to EIA Scandinavian Peninsula (Fig. 2c). The Wielbark archaeological complex has been linked to the later Chernyakhov culture to the southeast and to early Goths, an historical Germanic group that flourished in the second to fifth centuries CE[56]. Our modelling supports the idea that some groups that probably spoke Germanic languages from Scandinavia expanded south across the Baltic into the area between the Oder and Vistula rivers in the early centuries CE, although whether these expansions can be linked specifically with historical Goths is still debatable. Moreover, since a considerable

proportion of Wielbark burials during this period were cremations, the possible presence of individuals with other ancestries cannot be strictly rejected if they were exclusively cremated (and are therefore invisible in the aDNA record).

A previous study could not reject continuity in ancestry from the Wielbark-associated individuals to later medieval individuals from a similar region[12]. With the improved power of Twigstats, models of continuity are strongly rejected, with no one-source model of any preceding Iron Age or Bronze Age group providing a reasonable fit for the

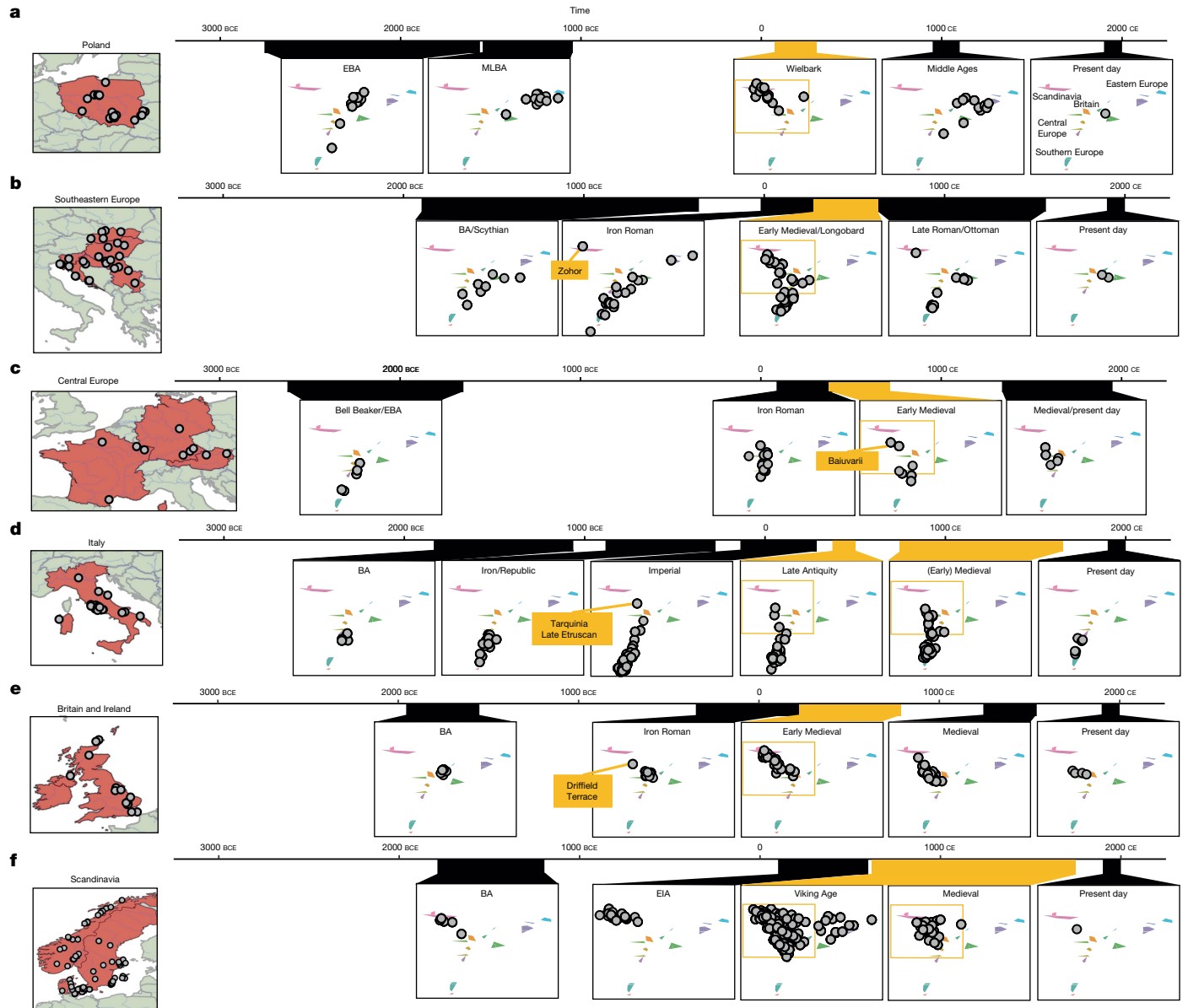

**Fig. 3 | Time transects across six geographical regions in Europe.**
**a**–**f**, Ancestry change visualized over a time transect spanning from the Bronze Age to the present day in Poland (**a**), southeastern Europe (**b**), central Europe (**c**), Italy (**d**), Britain and Ireland (**e**) and Scandinavia (**f**). The maps show sample locations of all available ancient genomes with at least 0.5× coverage from these regions (Supplementary Table 1). Their ancestry is shown on the same MDS model as in Fig. 2a for each time period. For each geographic region, the early medieval period is highlighted in orange and the area in the MDS corresponding to Scandinavian and central European ancestries is highlighted in an orange box.

medieval individuals ($P \ll 1 \times 10^{-32}$). Instead, the majority of individuals from medieval Poland can be modelled only as a mixture of ancestries related to Roman Iron Age Lithuania, which is similar to ancestries of individuals from middle to late Bronze Age Poland (44%, 95% confidence interval 36–51%), an ancestry component related to Hungarian Scythians or Slovakian La Tène individuals (49%, 95% confidence interval 41–57%) and potentially a minority component of ancestry related to Sarmatians from the Caucasus ($P = 0.13$) (Fig. 2c). Four out of twelve individuals from medieval Poland, three of whom are from the late Viking Age[6], carried detectable Scandinavian-related ancestry. Some of the ancestry detected in individuals from later medieval Poland may have persisted during the late first millennium CE in the cremating portion of the population, but regardless, this points to large-scale ancestry transformation in medieval Poland (Fig. 3a). Future data could shed light on the extent to which this reflects the influence of groups speaking Slavic languages in the region.

In present-day Slovakia, individuals associated with the Iron Age La Tène period appear close to Hungarian Scythians in the two dimensions of our MDS analysis, and are modelled as a mixture of central and eastern European ancestry. However, a first-century CE burial of a 50–60-year-old woman from Zohor is modelled only with Scandinavian-related ancestry, providing evidence of ancestry related to the Scandinavian EIA appearing southwest of the range of the Wielbark archaeological complex[5,57] (Fig. 3b). Later early medieval individuals from Slovakia have partial Scandinavian-related ancestry, providing evidence for the integration between expanding and local groups.

Nearby, in present-day Hungary, we observe Scandinavian-related ancestry components in several burials dating to the sixth century CE associated with Longobards (Longobard_earlyMED(I))[10] (Fig. 2c). This is consistent with the original study[10], which reported affinity to present-day groups from northwestern Europe (GBR, CEU and FIN in the 1000 Genomes Project (1000GP))[10] but which we can resolve with

higher resolution using earlier genomes. Several other individuals from these Longobard burials (Longobard_earlyMED(II)) show no detectable ancestry from northern Europe and, instead, are more closely related to Iron Age groups in continental central Europe, putatively representing descendants of local people buried in a Longobard style. Our results are consistent with attestations that the Longobards originated in the areas of present-day northern Germany or Denmark, but that by the sixth century CE they incorporated multiple different cultural identities, and mixed ancestries. Present-day populations of Hungary do not appear to derive detectable ancestry from early medieval individuals from Longobard contexts, and are instead more similar to Scythian-related ancestry sources (Extended Data Fig. 6), consistent with the later impact of Avars, Magyars and other eastern groups[58].

In southern Germany, the genetic ancestry of individuals from early medieval Bavaria probably associated with the historical Germanic-language-speaking Baiuvarii[59] cannot be modelled as deriving ancestry solely from earlier groups in Iron Age central Germany ($P \ll 1 \times 10^{-36}$). The Baiuvarii probably appeared in the region in the fifth century CE[59], but their origins remain unresolved. Our current best model indicates a mixture with ancestry derived from EIA Peninsular Scandinavia and central Europe, suggesting an expansion of Scandinavian-related ancestry producing a regional ancestry shift (Figs. 2c and 3c).

In Italy, southward expansions of northern and central European ancestries appear by the Late Antiquity (approximately fourth century CE), where a clear diversification of ancestry can be observed compared with preceding time periods (Fig. 3d). However, no individuals with near 100% Scandinavian ancestry can be observed in the sampling data available so far.

In Britain, the ancestries of Iron Age and Roman individuals form a tight cluster in our MDS analysis (Fig. 3e), shifted relative to available preceding Bronze Age individuals from Ireland and Orkney, and adjacent to, but distinct from, available individuals in Iron Age and Roman central Europe. However, two first- to second-century CE burials from a Roman military fortress site in Austria (Klosterneuburg)[5] carry ancestry that is currently indistinguishable from Iron Age or Roman populations of Britain, to the exclusion of other groups (qpWave cladality $P = 0.11$). One option is that they had ancestry from Britain; alternatively, currently unsampled populations from western continental Europe carried ancestries similar to Iron Age southern Britain.

Twigstats substantially improves models of admixture between ancestries from Iron Age Britain and northern Europe in early medieval England[9], halving standard errors from 9% with SNPs to 4% when using time stratification (point estimates 80% and 79% Iron Age Britain-related ancestry, respectively). We used this improved resolution to demonstrate that an earlier Roman individual (6DT3) dating to approximately second to fourth century CE from the purported gladiator or military cemetery at Driffield Terrace in York (Roman *Eboracum*), England[60], who was previously identified as an ancestry outlier[61,62], specifically carried approximately 25% EIA Scandinavian Peninsula-related ancestry (Fig. 2c). This documents that people with Scandinavian-related ancestry already were in Britain before the fifth century CE, after which there was a substantial influx associated with Anglo-Saxon migrations[9]. Although it is uncertain whether this individual was a gladiator or soldier, individuals and groups from northern Europe are indeed recorded in Roman sources both as soldiers and as enslaved gladiators[63,64].

Across Europe, we see regional differences in the southeastern and southwestern expansions of Scandinavian-related ancestries. Early medieval groups from present-day Poland and Slovakia carry specific ancestry from one of the Scandinavian EIA groups—the one with individuals primarily from the northern parts of Scandinavia in the EIA—with no evidence of ancestry related to the other primary group in more southern Scandinavia (Fig. 2d). By contrast, in southern and western Europe, Scandinavian-related ancestry either derives from

EIA southern Scandinavia—as in the cases of the probable Baiuvarii in Germany, Longobard-associated burials in Italy and early medieval burials in southern Britain—or cannot be resolved to a specific region in Scandinavia. If these expansions are indeed linked to language, this pattern is remarkably concordant with the main branches of Germanic languages, with the now-extinct eastern Germanic spoken by Goths in Ukraine on the one hand, and western Germanic languages such as Old English and Old High German recorded in the early medieval period on the other hand.

## Influx into pre-Viking Age Scandinavia

In EIA Scandinavia (<500 CE), we find evidence for broad genetic homogeneity. Specifically, individuals from Denmark (100 CE–300 CE) were indistinguishable from contemporary people in the Scandinavian Peninsula (Fig. 2c). However, we observe a clear shift in genetic ancestry already in the eighth century CE (Late Iron Age/early Viking Age) on Zealand (present-day Denmark) for which a 100% EIA ancestry model is rejected ($P = 1 \times 10^{-17}$ using Twigstats; $P = 7.5 \times 10^{-4}$ without). This shift in ancestry persists among later Viking Age groups in Denmark, where all groups are modelled with varying proportions of ancestry related to Iron Age continental groups in central Europe (Figs. 3f and 4c). A non-parametric MDS of Viking Age individuals suggests that variation between individuals forms a cline spanning from the EIA Scandinavian Peninsula individuals to ancestry characteristic of central Europe (Fig. 4e). The observed shift in ancestry in Denmark cannot be confounded by potentially earlier unknown gene flow into Iron Age source groups in Austria, France and Germany, but such gene flow could affect the exact ancestry proportions.

These patterns are consistent with northward expansion of ancestry, potentially starting before the Viking Age, into the Jutland peninsula and Zealand island towards southern Sweden. The geographical origin of this ancestry is currently difficult to discern, as the available samples from Iron Age central Europe remain sparse. The timing of this expansion is constrained only by the samples available: this ancestry is not observed in individuals from the Copenhagen area of Denmark (around 100 CE–300 CE)[6], an individual from the southern tip of Sweden (around 500 CE)[16], individuals from the Sandby Borg massacre site on Öland in present-day Sweden (around 500 CE)[7] and 31 individuals from the mid-eighth century Salme ship burials in present-day Estonia (Extended Data Fig. 9), who probably originated in central Sweden[6]. Therefore, this ancestry transformation most likely postdated these individuals in each particular region and mostly occurred in the second half of the first millennium CE.

To assess the full extent of the impact of this ancestry influx into Scandinavia, we next aimed to understand the ancestry of individuals in Scandinavia during the Viking Age. Previous studies have suggested that there was a diversity of ancestries in Scandinavia during this period[6,7,65], due to increased maritime mobility, but have not reported per-individual ancestry estimates based on preceding ancestry. We analysed each individual's ancestry using a rotational qpAdm scheme (Fig. 4a, Extended Data Fig. 9 and Supplementary Table 4), which showed increased power in distinguishing models when restricted to recent coalescences with Twigstats (more than 80% of accepted one-source models in Twigstats were also accepted one-source models using all SNPs, compared with less than 17% for the inverse).

We investigated regional differences in non-local ancestry across Scandinavia. In Denmark, 25 out of 53 Viking Age individuals had detectable ($z$-score > 1) central European-related ancestry (CentralEurope. IronRoman or Portugal.IronRoman) in their best accepted qpAdm models. In Sweden 20 out of 62 individuals had detectable central European-related ancestry, concentrated almost entirely in southern regions (Fig. 4a,d). By contrast, in Norway, this ancestry was observed in only 2 out of 24 individuals, indicating a wide-ranging impact of incoming ancestry in southern Scandinavia and suggesting more

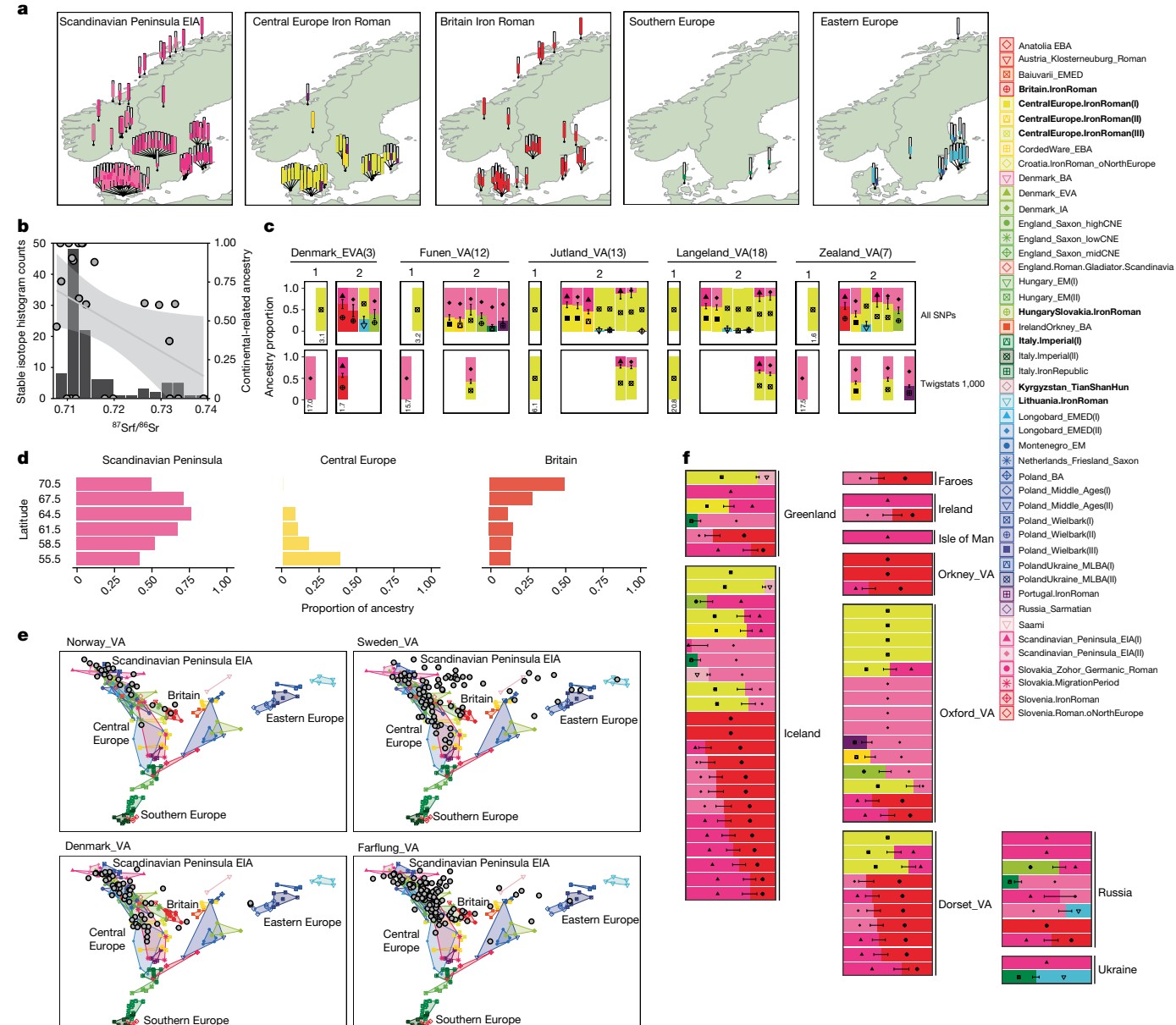

**Fig. 4 | Ancestry in the Viking world. a**, Map showing ancestry carried by Scandinavian Viking Age individuals as inferred using the best-fitting qpAdm model. These are chosen by either choosing the one-source model with largest *P* value and *P* > 0.01 or the two-source model with the largest *P* value and *P* > 0.01. Extended Data Fig. 7 shows the same map with all accepted models. **b**, Stable isotope data indicating the geology of childhood origin. The histogram shows the ratio of strontium isotopes 87 to 86 measured in 109 individuals in Öland[69]. For individuals included in our ancestry modelling, we plot Iron Age central European-related ancestry against their stable isotope values (grey circles, *r* = −0.39, *P* = 0.075). Shared area corresponds to the 95% confidence band around the regression line. **c**, The ancestry shift observed in Viking Age Danish groups using qpAdm on all SNPs or Twigstats. We show the best one-source and all two-source models with *P* > 0.05. For models with *P* < 0.05, the −log₁₀[*P* value] is shown under the plot. Sample sizes for each group are shown in brackets. **d**, The ancestry proportion across Viking Age individuals in Denmark, Sweden and Norway grouped by latitude. **e**, Viking Age genetic variation (grey circles) visualized on the same MDS as in Fig. 2a,b. **f**, The best-fitting qpAdm ancestry model for far-flung Viking individuals. Detailed models for all individuals are shown in Extended Data Figs. 9 and 10. In **c** and **f**, we show one s.e. Rotating qpAdm sources are marked in bold in the key.

continuity from the EIA in Norway and northern Sweden (Fig. 4a). When considered collectively, the individuals who show evidence of central European-related ancestry are mostly observed in regions historically within the Danish sphere of influence and rule. Currently, no such individuals, for example, are noted in eastern central Sweden, which was a focus of regional power of the Svear (Fig. 4a). The difference in distribution could suggest that the central European-related ancestry was more common in regions dominated by the historical Götar and groups inhabiting the lands on the borders of the Danish kingdom.

To test the extent to which the variation in ancestry was consistent with mobility during the lifetime of the individuals or, alternatively,

that of established groups, we focused on the island of Öland in southeast Sweden, where 23 individuals for whom we could reconstruct ancestry portraits also had associated strontium stable isotope data[66]. Strontium isotope data from dental enamel reflect the geology of the region where an individual grew to maturity, and there are considerable differences in expectations between Öland and many other regions in northern Europe. The full range of strontium isotope ratios in 109 individuals show two modes, a majority group with low ratios and a second minority group with high ratios falling outside the expected range of local fauna (Fig. 4b). Among 23 individuals with genomes in our data, all 5 individuals with 100% ancestry relating to central Europe

(including one with ancestry related to Britain) are part of the majority strontium values, consistent with them having grown up locally. By contrast, the six most clearly non-local individuals based on the stable isotopes all have 50% or more EIA Scandinavian Peninsula-related ancestry, although three individuals with wholly EIA Scandinavian Peninsula-related ancestry also had local values. This suggests that the presence of central European-related ancestry was not a transient phenomenon, but an ancestry shift that occurred at some point after about 500 CE, the period to which individuals from the massacre site at Sandby Borg ringfort on Öland were dated; these individuals all have strictly EIA Scandinavian-related ancestry. Indeed, one hypothesis is that the massacre at Sandby Borg could represent conflict associated with movements of people that contributed to later ancestry change, although other scenarios are possible and further synthesis of biomolecular and archaeological data is necessary to test this hypothesis.

## Viking Age mobility into Scandinavia

Previous studies had suggested a major influx of ancestry related to Britain into Viking Age Scandinavia[6,7]. Although we detect this ancestry in some individuals (7 individuals in Norway, 14 in Denmark and 14 in Sweden), including some individuals whose ancestry appears to be entirely derived from Iron Age Britain, its overall impact appears reduced compared with previous reports. Our analysis indicates a proportionally larger impact of ancestry from Iron Age Britain in northern Norway, with southern Scandinavia predominantly influenced by continental central European ancestries (Fig. 4d). We hypothesize that our estimates of ancestry from Britain are reduced relative to previous studies because ancestry related to Britain and continental central Europe may have been indistinguishable. This could be due to a lack of statistical power to distinguish these closely related sources with standard methods, as well as through potential biases introduced by using modern surrogate populations that have since been influenced by later gene flow (such as gene flow into Britain). We illustrate this by replicating the analyses previously described[6,7] (Extended Data Fig. 8).

Similarly, a previous study has suggested that individuals at sites such as Kärda in southern Sweden carried ancestry from southern Europe[6]. In our models, two Kärda individuals fit with central European-related ancestry, but none of the individuals has a substantial proportion of ancestry related to southern European sources (Extended Data Fig. 9). Instead, we detect ancestry from southern European sources in only three individuals from Scandinavia, and in relatively small proportions (Fig. 4a).

Interestingly, we detect ancestry from Bronze and Iron Age sources from Eastern Europe (present-day Lithuania and Poland), concentrated in southeastern parts of Sweden, particularly the island of Gotland (14 individuals; Fig. 4a). This is consistent with previous genetic studies[6,7]. We find that this ancestry is enriched in male individuals (Extended Data Fig. 7d), suggesting male-biased mobility and/or burial. The closest match tends to be Roman Iron Age Lithuanian genomes associated with Balts, which would be consistent with mobility across the Baltic Sea, but we caution that the geographical representation of available genomes is still limited.

## Viking Age expansion from Scandinavia

Traditionally, historical perspectives on what is now often referred to as the Viking diaspora placed an emphasis on the movements and settlements of population groups from various parts of Scandinavia[67]. Our explorative MDS analysis again indicates mixed ancestries related to the Scandinavian EIA, with regional differences that point to varied local admixture (Fig. 4e and Extended Data Fig. 10).

In Britain, most of the individuals recovered from the two late Viking Age mass graves identified at Ridgeway Hill, Dorset, and St John's College, Oxford[6], show ancestries typical of those seen in Viking Age southern Scandinavia (Fig. 4f). Further west, North Atlantic Viking Age individuals in the Faroe Islands, Iceland and Greenland carry ancestry from the Scandinavian Peninsula, with several individuals showing the continental central Europe-related ancestry signal found in southern Scandinavia (Fig. 4f) and others who share substantial ancestry with Iron Age Britain. In contrast to previous hypotheses[68], we found a marginal enrichment of ancestry related to Britain and Ireland in men (15 out of 17 men and 3 out of 6 women with at least one accepted model involving Iron or Roman Age Britain as source; Fisher's exact test $P = 0.089$) (Extended Data Fig. 7c,e). However, sampling of additional individuals to improve distinction between early English- and Norse-related ancestries would be required to fully test this hypothesis.

In eastern Europe, we observe EIA Scandinavian ancestries in a Viking Age burial from Ukraine, and these ancestries are overrepresented in Viking Age burials from present-day Russia. At Staraya Ladoga in western Russia, we observe several individuals with EIA Scandinavian Peninsula-related ancestry and at least one individual dated to the eleventh century with apparent ancestry related to Iron Age Britain. The relative absence of Iron Age central European ancestry, which was largely restricted to southern Scandinavia during the Viking Age, is thus indicative that these individuals may have originated in the central/northern parts of Sweden or Norway, where Viking Age individuals show the most similar ancestry profiles to them.

## Conclusions

Our approach, Twigstats, transfers the power advantage of haplotype-based approaches to a fully temporal framework, which is applicable to $f$-statistics and enables previously unavailable unbiased and time-stratified analyses of admixture. We demonstrated that Twigstats enables fine-scale quantitative modelling of ancestry proportions, revealing wide-ranging ancestry changes that affect northern and central Europe during the Iron, Roman and Viking ages. We reveal evidence of the southward and/or eastward expansion of individuals who probably spoke Germanic languages and who had Scandinavian-related ancestry in the first half of the first millennium CE. We note that 'Scandinavian-related' in this context relates to the ancient genomes available, and so it is entirely possible that these processes were driven, for example, from regions in northern-central Europe. This could be consistent with the attraction of the greater wealth, which tended to build up among Rome's immediate neighbours and may have played a major role in vectors of migration internal to communities in Europe who lived beyond the Roman frontier[52]. Later, patterns of gene flow seem to have turned northwards, with the spread of Iron Age Central Europe-related ancestry into Scandinavia. Overall, our approach can be used for the reconstruction of new high-resolution genetic histories around the world.

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

## Methods

### Twigstats

Twigstats takes the Relate[32] output format as input and allows the computation of $f$-statistics directly on genealogies, by using the inferred expected number of mutations on each branch as input, which is computed as the product of a prespecified average mutation rate per base per generation, the branch length and the number of bases each tree persists[43]. Importantly, Twigstats computes $f_2$-statistics ascertained by an upper date threshold, such that only branches younger than this threshold are used. If a branch crosses the threshold, we use only the proportion of the branch underneath the threshold. Twigstats additionally enables us to specify a minimum derived allele frequency and lower date threshold. Twigstats can also compute $f_2$-statistics on age-ascertained mutations, which is particularly convenient for individuals not built into the genealogies.

The computed $f_2$-statistics are fed into ADMIXTOOLS2[70] to compute derived statistics. ADMIXTOOLS2 implements computation of genome-wide $f_2$-, $f_3$- and $f_4$-statistics, as well as qpgraph and qpAdm models. We implement the sample size correction as detailed in ref. 21. The $f_2$-statistics are computed in blocks, typically of prespecified centimorgan size or of prespecified physical distance. These blocks are used downstream in ADMIXTOOLS2 to compute standard errors using a block-jackknife approach. By default, we compute $f$-statistics only on internal branches and exclude singleton tip branches to increase robustness against sample age.

The optimal Twigstats time cut-off is a priori unknown; however, we develop a theory that predicts the optimal choice in a simple two-way admixture as a function of the admixture date, source split time and admixture proportion (Supplementary Note). In this case, the optimal cut-off equals approximately 1.4 times the split time between admixing source groups, depending on exact parameters in the model (Fig. 1b,c and Extended Data Fig. 2).

**Non-negative least squares ancestry modelling.** We implement an approach that uses genealogies to emulate the chromosome painting technique of identifying closest genetic relatives along the genome[1,2] to fit admixture weights. When applied to true genealogies in simulations, this approach represents an idealized version of this idea.

We implement this function in Twigstats, which, given known assignment of each sample to a population, identifies, at each position in the genome, the population with which a sample coalesces first. Our implementation takes a list of reference populations as input, such that any coalescences that do not involve these reference populations are ignored when traversing back in time through genealogical trees. If the first coalescence involves multiple different reference populations, this coalescence event will be assigned to each population with a weight proportional to the number of samples in each population involved in that event.

We then implement a second function in Twigstats to compute, for each target population and putative source populations, the proportion of the genome 'painted' by each of the reference populations. Given $k$ reference populations, we denote by $\mathbf{a}_i$ the vector of length $k$ storing these proportions for population $i$. We fitted our target population as a mixture of putative source populations using a non-negative least squares approach that finds a solution to the optimization problem $\min_{0 \leq \Sigma \beta_\ell \leq 1} \|\mathbf{a}_{\text{target}} - \mathbf{A}\boldsymbol{\beta}\|_2$, where $\mathbf{A}$ is a matrix storing $\mathbf{a}_\ell$ for putative source populations as its column vectors with $\ell$ indexing source populations and $\boldsymbol{\beta}$ are non-negative mixture weights.

**Admixture simulations.** We use msprime[71] to simulate genetic variation data to test our approach. All simulation scripts are available at https://github.com/leospeidel/twigstats_paper.

**$f_4$-ratio admixture simulation.** Our simulation in Fig. 1b and Extended Data Fig. 3b simulates five populations named PI, PO, P1, P2 and PX, where PO splits from all other populations 10,000 generations ago, P1 and P2 represent two proxy source groups that split from each other at 250 generations or 500 generations ago, PI splits from P1 100 generations ago and PX emerges from a pulse admixture between P1 and P2 50 generations ago. All populations have a constant diploid population size of 5,000, a variable human-like recombination map, in which our simulation only covers chromosome 1, and a human-like mutation rate of $1.25 \times 10^{-8}$ mutations per base per generation. We additionally have a modified simulation with a lower mutation rate of $4 \times 10^{-9}$ mutation per base per generation, emulating a transversions-only dataset, and a simulation in which P2 has a diploid population size of 1,000 in the last 50 generations, emulating a recent bottleneck in this population. We sample 20 haploid sequences from all populations. The 'large sample size' simulation samples 100 haploid sequences from all populations.

**$f_4$-ratio admixture simulation with genotype and phasing errors.** We emulate the data quality we expect in imputed ancient genomes (Extended Data Fig. 3b). We implement a simple error model in which every haploid genotype at any segregating site can switch with a certain error probability. We can theoretically compute the predicted squared correlation coefficient ($r^2$) between the true simulated genotypes and the genotypes that include error, stratified by minor allele frequency, to generate a plot similar to those used for evaluating imputation accuracy using downsampled high-coverage ancient genomes[72] (Extended Data Fig. 3a). As imputation accuracy varies for each individual in real settings, we randomly sample the error probability for each individual uniformly between $1 \times 10^{-4}$ and $1 \times 10^{-3}$ (errors per SNP per haplotype). This yields $r^2$ curves that are comparable to those observed in real data. We additionally simulate a high error case, for which we sample error probabilities between $1 \times 10^{-3}$ and $1 \times 10^{-2}$.

In real settings, we are additionally required to computationally phase genomes. We emulate this by combining two haploid sequences to construct a diploid individual. We then computationally rephase these diploid individuals without a reference panel. This approach is expected to result in suboptimal phasing and should therefore be well suited to test robustness to phase-switch errors.

**qpAdm simulation.** Our simulation in Extended Data Fig. 3c uses the simulation model and script provided with ref. 23, although we changed this script to use the human hotspot recombination map. We simulate only chromosome 1. In the original simulation model, admixing sources split 1,200 generations ago, with admixture occurring 40 generations ago. We additionally simulate a version in which all population split times and admixture times are reduced by a factor of 5. We sample 20 haploid sequences per population.

**Stepping-stone separation by distance simulation.** We adapt the simulation model provided previously[23] to simulate a stepping-stone model of nine populations organized on a 1D grid, in which individuals are able to migrate between adjacent populations (Extended Data Fig. 3d). We changed this script to use the human hotspot recombination map and simulate only chromosome 1. We simulate under migration rates of 0.001 and 0.005, corresponding to average $F_{\text{ST}}$ values of 0.01 and 0.002, respectively[23]. We sample 20 haploid sequences per population. We then fitted population 4 using pairs of other populations as sources in a rotational qpAdm scheme such that unused populations are assigned to the reference set.

We expect that this simulation model violates qpAdm assumptions of no (or limited) gene flow after admixture between sources and reference groups. Consistent with this idea, qpAdm models are rejected ($P = 4 \times 10^{-38}$ for migration rates of 0.001 and $P = 5 \times 10^{-8}$ for migration rates of 0.005) when using Twigstats with a cut-off of 1,000 generations. However, these are not rejected using regular qpAdm, including

when migration rates are high (and, therefore, $F_{ST}$ is low), indicating that Twigstats is better powered to detect such scenarios of continued migration. Encouragingly, a model that involves the two immediately adjacent populations is selected in all replicates as the 'best' model (highest qpAdm $P$ value) using Twigstats, whereas this is the case in only 80% (migration rate of 0.001) and 30% (migration rate of 0.005) of all replicates using regular qpAdm.

**Neanderthal admixture and deep structure simulation.** Our simulation in Extended Data Fig. 5d emulates Neanderthal admixture, in which Neanderthals and ancestors of modern humans split 25,000 generations ago and admixture occurs 2,000 generations ago. The resulting admixed non-African-like population coexists with the non-admixed African-like population until the present day. Furthermore, two Neanderthal populations split from each other 7,000 generations ago, which can be interpreted as emulating the Altai and Vindija Neanderthal populations, with Vindija being closer to the admixing source.

We simulate an alternative model with two subgroups emulating ancestral modern humans in Africa that have a non-zero symmetric migration rate, ranging from $4 \times 10^{-5}$ to $2 \times 10^{-4}$ per generation, up until 3,000 generations before present. One of these subgroups gives rise to a present-day African-like population, while the other gives rise to a present-day non-African-like population. We further sample two Neanderthal populations that split 7,000 generations ago and merge 25,000 generations ago with the same ancestral modern human subgroup that will eventually give rise to a non-African-like population.

We simulate whole genomes with human-like recombination rates and a mutation rate of $1.25 \times 10^{-8}$ mutations per base per generation. Diploid effective population sizes are set to 10,000 except on the Neanderthal lineage, in which it is set to 3,000. We sample 2 haploid sequences for each Neanderthal population and 20 haploid sequences for the target admixed population and African non-admixed population.

**Fine-scale structure simulation.** Our simulation in Extended Data Fig. 5a emulates the emergence of a fine-scale population structure and is adapted from ref. 39. In this simulation, populations split 100 generations ago into 25 subpopulations followed by a period in which individuals are allowed to migrate at a rate of 0.01 between adjacent populations in a $5 \times 5$ grid. The diploid effective population size is 500 in each of the 25 populations, and 10,000 in the ancestral population. We simulate ten replicates of chromosome 10, with a human-like mutation rate of $1.25 \times 10^{-8}$ and hotspot recombination map. We sample two diploid individuals from each population. Furthermore, we sample 100 individuals from an ancestral population that splits from the 25 target populations 100 generations ago, before the emergence of structure in these 25 populations. Relate trees are inferred assuming true mutation rates, recombination rates and average coalescence rates across all samples.

**Ancient sample selection.** A full list of ancient genomes can be found in Supplementary Table 1. Published ancient shotgun genomes provided by refs. 7,8 were only available aligned against the GRCh38 reference sequence. These data were realigned to the GRCh37d5 reference sequence using bwa aln (v. 0.7.17-r1188).

We select genomes with average autosomal coverage above 0.5×, except for VK518, which has previously been suggested to be of Saami ancestry[6] and which had a coverage of 0.438. We included VK518 in our panel to capture this ancestry. Genomes above a coverage cut-off of 0.5× have previously been shown to result in reliable imputation results[72]. We exclude samples with evidence of contamination. We remove any duplicate individuals, such as individuals who were resequenced, choosing the file with the highest coverage. We then filter out any relatives annotated in the Allen Ancient DNA Resource v. 54.1[27], retaining the individual with the highest coverage in each family clade.

Our final dataset includes 1,556 ancient genomes.

**Imputation of ancient genomes.** We follow the recommended pipeline of GLIMPSE[73] and first call genotype likelihoods for each genome in the 1000GP, segregating sites using bcftools mpileup with filter -q 20, -Q 20 and -C 50. We subsequently impute each genome separately using GLIMPSE v. 1.1.1 using the 1000GP phase 3 reference panel[74] downloaded from https://ftp.1000genomes.ebi.ac.uk/vol1/ftp/release/20130502/. These imputed genomes are merged into a single VCF (variant call format) for further downstream processing.

We filter any site for which more than 2% of sites have an imputation posterior of less than 0.8 and retain all remaining sites so as not to have any missing genotypes at individual SNPs.

**Relate-inferred genealogies.** We merge imputed ancient genomes with a subset of the 1000GP dataset, including all European populations (CEU, Utah residents with northern and western European ancestry; CHB, Han Chinese in Beijing, China; FIN, Finnish in Finland; GBR, British in England and Scotland; BS, Iberian populations in Spain; TSI, Toscani in Italy; YRI, Yoruba in Ibadan, Nigeria). We create a second dataset in which we merge imputed genomes with the Simons Genome Diversity Project[75] (SGDP) downloaded from https://sharehost.hms.harvard.edu/genetics/reich_lab/sgdp/phased_data2021/. These two datasets contain, respectively, a total of 2,270 and 1,834 modern and ancient individuals.

We then infer genealogies for the joint dataset of ancient and modern genomes using Relate v. 1.2.1. We restrict our analysis to transversions only and assume a mutation rate of $4 \times 10^{-9}$ mutations per base per generation and input sample dates as shown in Supplementary Table 1. We use coalescences rates pre-inferred for the 1000GP and SGDP datasets.

**MDS analysis.** We compute $f_2$-statistics using the Twigstats function f2_blocks_from_Relate between all pairs of individuals and between all individuals and an outgroup (Han Chinese people in SGDP) using the Relate genealogies of SGDP modern and imputed ancient genomes. We set the argument $t$ to specify a time cut-off and set the argument use_muts to FALSE to compute these $f$-statistics on branches of the genealogy and to TRUE to compute these only on the mutations. We use these to compute $f_3$(outgroup, indiv1, indiv2) = $0.5 \times (f_2$(outgroup, indiv1) + $f_2$(outgroup, indiv2) − $f_2$(indiv1, indiv2)) for every pair of individuals, and store $1 - f_3$(outgroup, indiv1, indiv2) in a symmetric $N \times N$ matrix (where $N$ is the number of individuals) for which we then compute an MDS using the R function cmdscale.

**qpAdm modelling.** In brief, qpAdm models are a generalization of $f_4$-ratios, for which one-, two- and three-source models can be tested as hypotheses and admixture components and their s.e. obtained with a block jackknife[13]. A qpAadm model is fully specified by a set of putative source groups and additional 'outgroups' that are used to distinguish source ancestries. We used a rotating approach in which we iteratively selected a subset of source groups and used all remaining putative sources as outgroups. This approach penalizes models where true contributing sources are used as outgroups. With sufficient statistical power, qpAdm models will be statistically rejected if true contributing sources are used as outgroups. If statistical power is more limited, several models will fit the data, but the correct model is expected to be preferred over wrong models. Throughout, we use the Relate genealogies of SGDP modern and imputed ancient genomes in our qpAdm modelling and first compute $f_2$-statistics using the Twigstats function f2_blocks_from_Relate between all populations involved, which we then feed to the ADMIXTOOLS2 package[70].

**Clustering using qpwave.** To overcome challenges with hand-curating source groups used in qpAdm modelling, we follow ref. 5 and run qpwave using Twigstats between pairs of ancient individuals. We use Han Chinese individuals from Beijing and five European populations from the 1000GP as reference groups. This approach tests whether two

individuals form a clade with respect to reference groups. The reason why this is a principled approach despite the 1000GP groups post-dating the ancient individuals is that if a group of ancient individuals are truly homogeneous, they will be so also with respect to later individuals.

We then define clusters by running UPGMA (unweighted pair group method with arithmetic mean) on $-\log_{10}[P$ values$]$ obtained from qpwave between all pairs of individuals and cut the resulting dendrogram at a height corresponding to a $P$ value of 0.01. We then further subdivide clusters by requiring all samples to be within 500 years of the mean cluster age.

To choose the source groups shown in Fig. 2a and Extended Data Fig. 1d, we run this algorithm on samples from Iron and Roman Age Europe (Supplementary Table 1). We retain groups that have at least three individuals and, therefore, exclude clusters of size one or two.

This approach results in two clusters in the Scandinavian Peninsula, approximately separating northern from southern Scandinavia, three clusters in Poland and Ukraine that separate samples temporally between the early and later Bronze Age, a cluster combining the Hungarian Scythian and Slovakian La Tène-associated individuals, and a cluster each for Iron and Roman Age Portugal, Italy and Lithuania. In present-day Austria, Germany and France, this approach identifies three clusters, with each cluster spanning multiple archaeological sites in different countries, indicating genetic diversity in this region in the first millennium CE. Encouragingly, these clusters separate in our non-parametric MDS analysis (Fig. 2a), indicating that we are capturing real genetic differences between groups using this approach.

**Fine-scale structure in Neolithic Europe.** To quantify fine-scale structure in Neolithic Europe (Extended Data Fig. 5b), we aimed to select individuals in Neolithic Europe who have not yet been affected by the arrival of Steppe ancestry and do not show excess hunter-gatherer ancestry. We infer distal ancestry sources using Balkan_N, Yamnaya and Western Hunter-gatherers as source groups and reference groups according to a previously proposed qpAdm setup[46] (Supplementary Table 1). For this analysis, we infer ancestry using qpAdm applied to 1.2 million SNP sites of imputed genomes. We retain only Neolithic individuals with $P > 0.01$, $z < 2$ for Yamnaya ancestry, and $z < 2$ or proportion $<0.25$ for Western Hunter-gatherer ancestry.

### Reporting summary

Further information on research design is available in the Nature Portfolio Reporting Summary linked to this article.

## Data availability

All aDNA data used in this study were publicly available, and accession codes are listed in Supplementary Table 1.

## Code availability

Twigstats is freely available under an MIT licence through GitHub (https://github.com/leospeidel/twigstats), and detailed documentation, as well as example data, is available at https://leospeidel.github.io/twigstats/. The code has also been deposited at Zenodo (https://zenodo.org/records/13833120)[76]. All scripts to reproduce simulations, and to run Relate on imputed ancient genomes, and downstream analyses, including computation of $f$-statistics and running qpAdm models, are available through GitHub (https://github.com/leospeidel/twigstats_paper).

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

**Acknowledgements** L.S. was supported by a Sir Henry Wellcome Fellowship (220457/Z/20/Z). P.S. was supported by the European Molecular Biology Organization, the Vallee Foundation, the European Research Council (852558), the Wellcome Trust (217223/Z/19/Z) and Francis Crick Institute core funding (FC001595) from Cancer Research UK, the UK Medical Research Council and the Wellcome Trust. B.R. was supported by the Swedish Research Council (2021-03333).

**Author contributions** P.S. supervised the study. L.S. and P.S. developed the method. L.S, M.S. and P.S. curated the dataset. L.S. and P.S. analysed the data and wrote the manuscript. L.S., M.S., T.B., B.R., K.A., C.B., A.G., P.H. and P.S. interpreted the results and edited the manuscript.

**Funding** Open Access funding provided by The Francis Crick Institute.

**Competing interests** The authors declare no competing interests.

**Additional information**
**Correspondence and requests for materials** should be addressed to Leo Speidel or Pontus Skoglund.

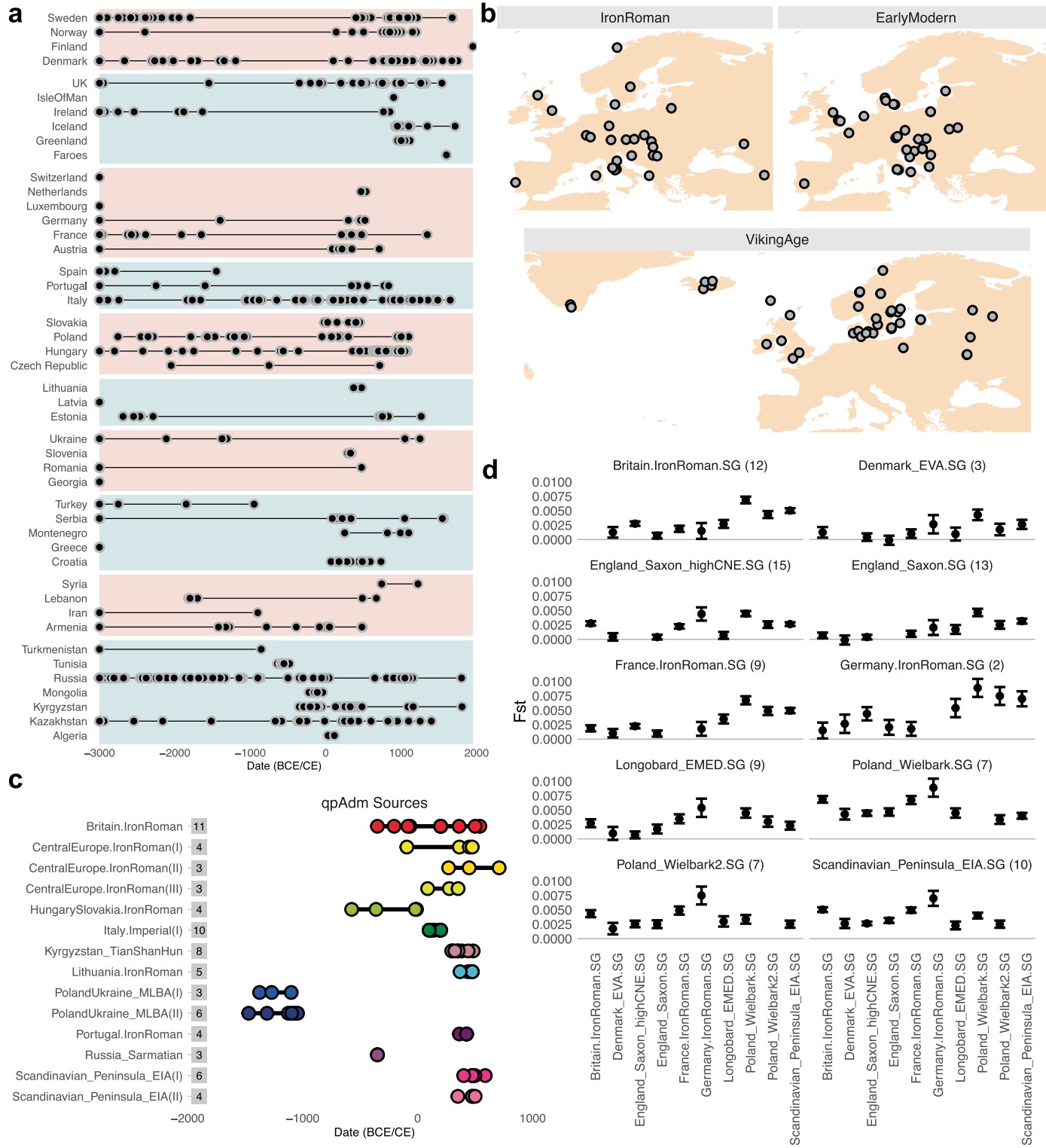

**Extended Data Fig. 1 | Collection of ancient genomes used in this study.**
**a**, Ancient DNA samples included in this study (Supplementary Table 1). Samples older than 3000 BCE are shown at 3000 BCE. **b**, Map showing mean coordinates of groups in the Iron, Early Modern, and Viking Ages. **c**, Source groups used in qpAdm modelling of Metal Age and early Medieval Europe (Figs. 2, 3 and 4), showing sample ages. Sample sizes are shown in grey boxes. **d**, $F_{ST}$ between Metal Age and early Medieval groups computed using popstats[77] using options --FST --informative. Sample sizes are shown in brackets and we show one standard error.

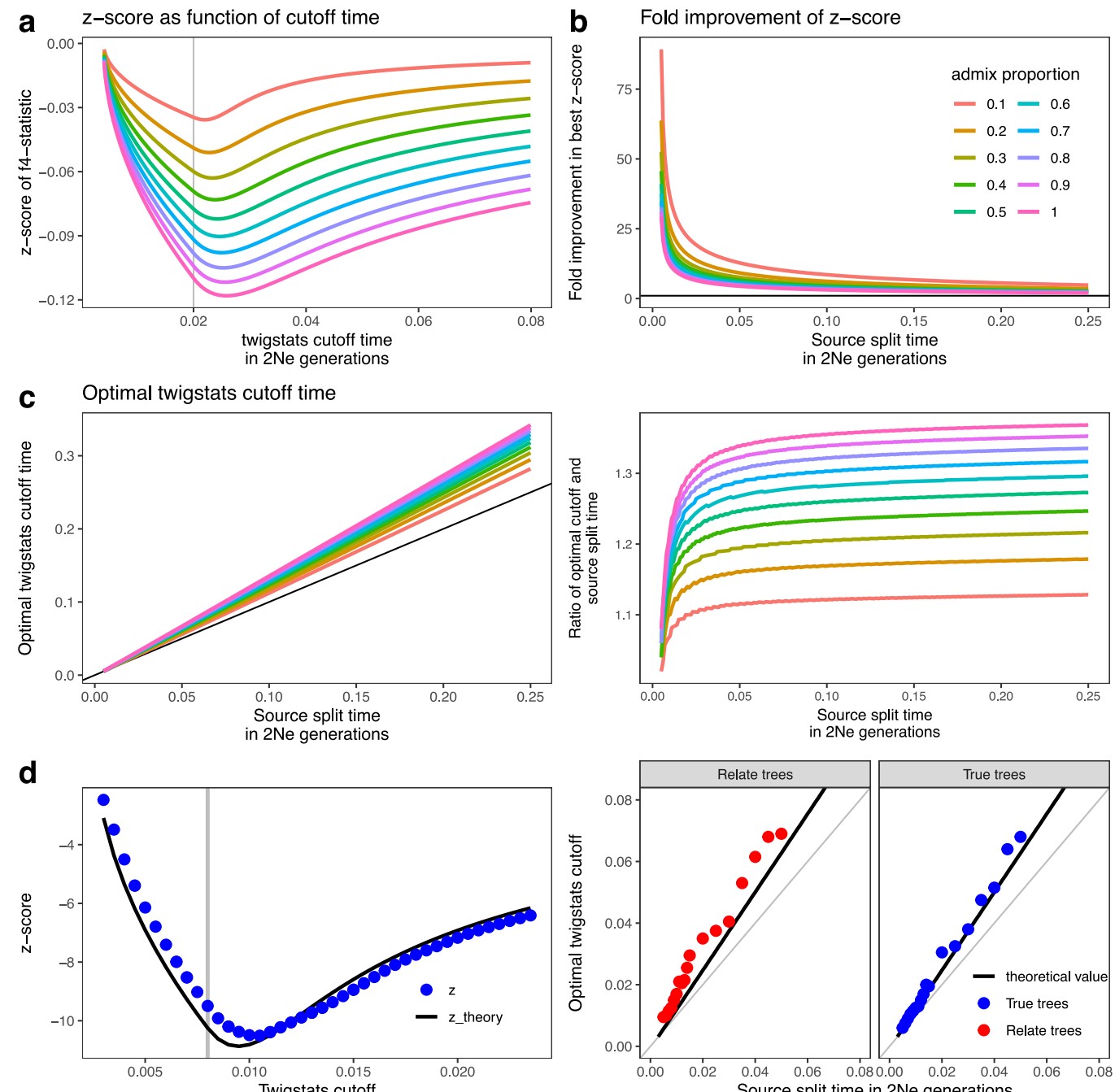

**Extended Data Fig. 2 | Twigstats optimal cutoff. a**, Theoretically computed z-score of $f_4$(PO,P1,PX,P2) at a single genomic locus (Supplementary Note), assuming PX is admixted between P1 and P2 at time 0.004 (in units of $2N_e$ generations), e.g. corresponding to 100 generations with $2N_e$ of 25,000. Sources split at time 0.02. **b**, The theoretical fold-improvement of the best Twigstats z-score of $f_4$(PO,P1,PX,P2) relative to the z-score obtained with regular $f_4$-statistics. We use the same parameters as in **a**, but vary source split times to illustrate the improved power for mixtures involving more closely related groups. **c**, The optimal Twigstats cutoff time as a function of the source split time and the ratio between the optimal cutoff time and source split time. **d**, Comparison of z-scores computed using Twigstats to the corresponding theoretical values shown in **a**.

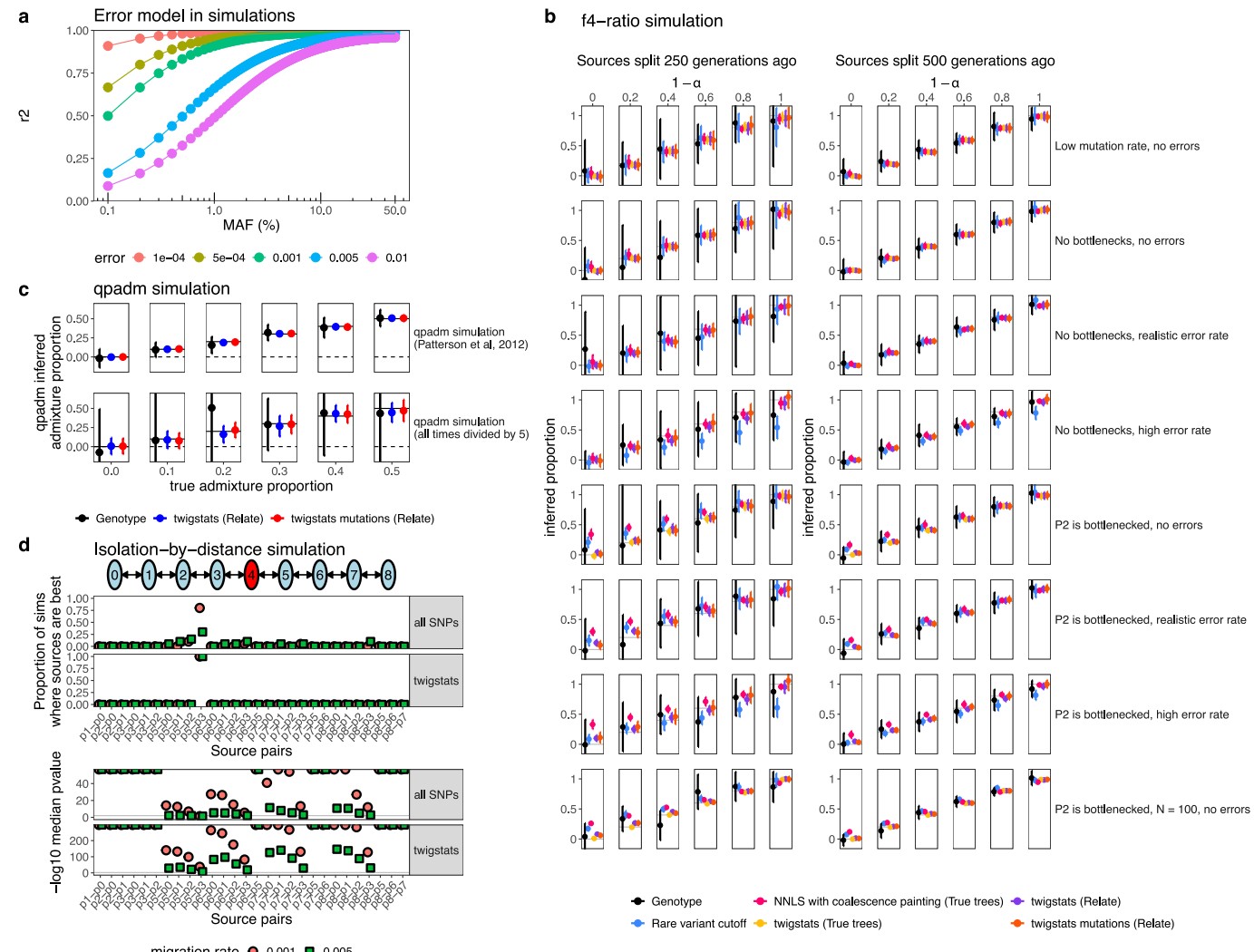

**Extended Data Fig. 3 | Simulations. a**, Theoretically computed $r^2$ stratified by minor allele frequency between true genotypes and genotypes with randomly introduced errors (Methods), emulating imputation accuracy plots. **b**, Admixture simulation where sources P1 and P2 split 250 or 500 generations ago and a pulse admixture event gives rise to a target population PX 50 generations ago. We vary demographic history, error rates, and sample sizes and simulate 20 replicates for each scenario (see Methods for simulation details). Admixture proportions are computed using an $f_4$-ratio statistic and the Twigstats cutoff is set to twice the source split time and the rare variant cutoff is 5%. We plot two standard errors around the mean. **c**, qpAdm simulation taken from[21,23], as well as an adapted version where all population split times and the admixture date are divided by 5. The Twigstats cutoff time is chosen to be

1200 generations (top) and 600 generations (bottom). We simulate 10 replicates and plot two standard errors around the mean. **d**, Simulation adapted from[23] of a stepping stone model with 9 populations organised on a 1-dimensional grid as shown, where individuals are able to migrate between adjacent fields. We run a rotational qpAdm to fit population 4 using other pairs of populations to the left and right as sources. We run 50 replicates and set the p-value of models with inferred proportions outside of [0,1] to 0. We then compute the proportion where a given pair achieves the best p-value (top) and show the median p-value across these replicates (bottom). In all simulations in **b, c, d**, we sample N = 20 haploid sequences per population, except for one simulation in **b**, where we sample N = 100 sequences.

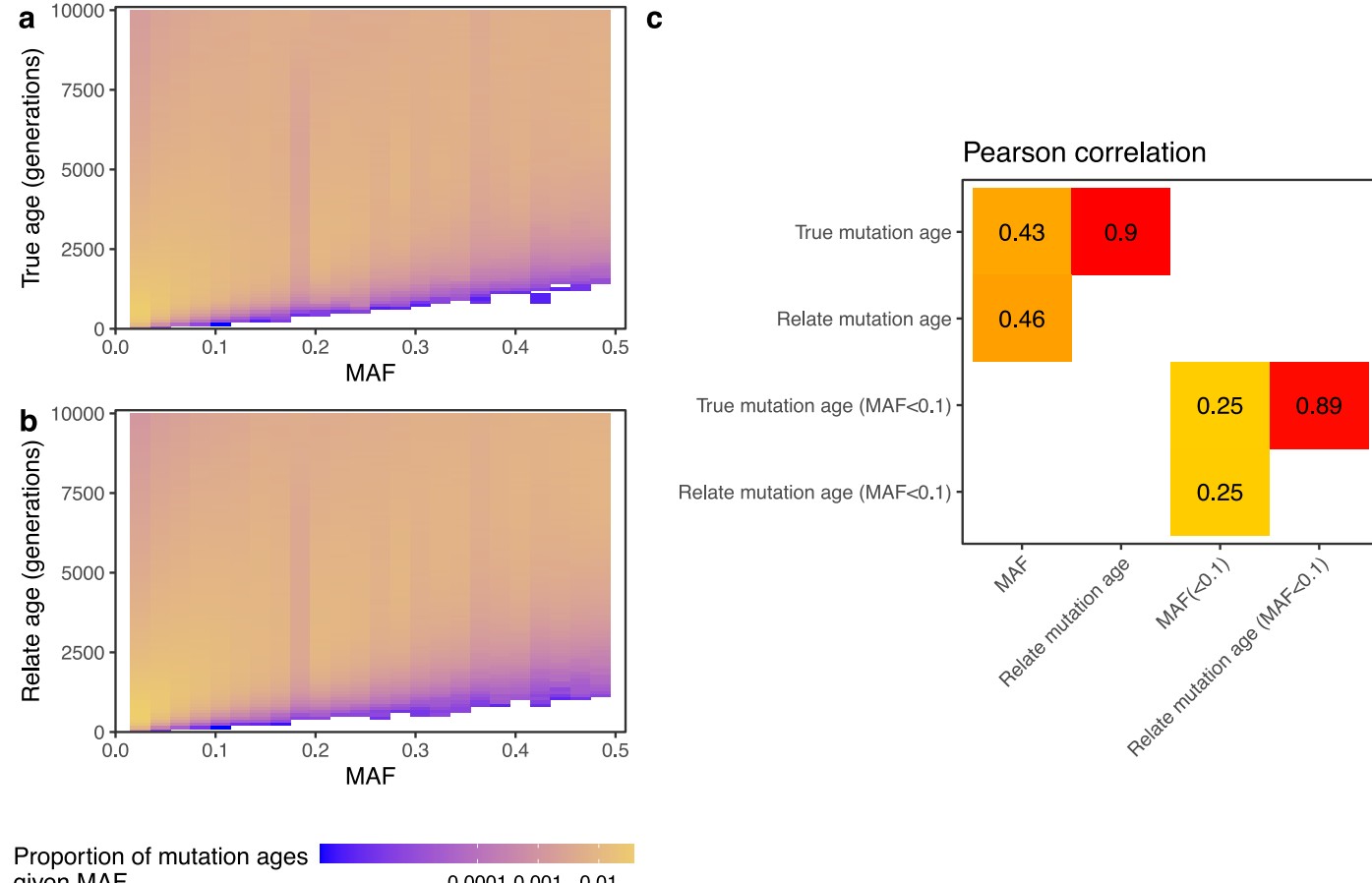

**Extended Data Fig. 4 | Relationship of mutation age and MAF. a**, Heatmap showing the distribution of mutation ages for each minor allele frequency (MAF) bin. To account for the uncertainty in when the mutation arose on a branch, we sample a random date between the lower and upper ends of the branch onto which it maps. We use 20 replicates of the simulation of Fig. 1b, where sources split 500 generations ago and the admixture proportion is 0%. **b**, Same as **a** but using mutation ages determined by Relate-inferred genealogies.

We place mutations at the same relative height between the lower and upper ends of a branch as in the true trees to remove the uncertainty in when on the branch the mutation occurred, so that we would recover the true allele age from a correctly inferred genealogy. **c**, Pearson correlation between MAF, true mutation age, and Relate mutation ages, as well as the same comparisons when restricting to mutations of MAF less than 0.1.

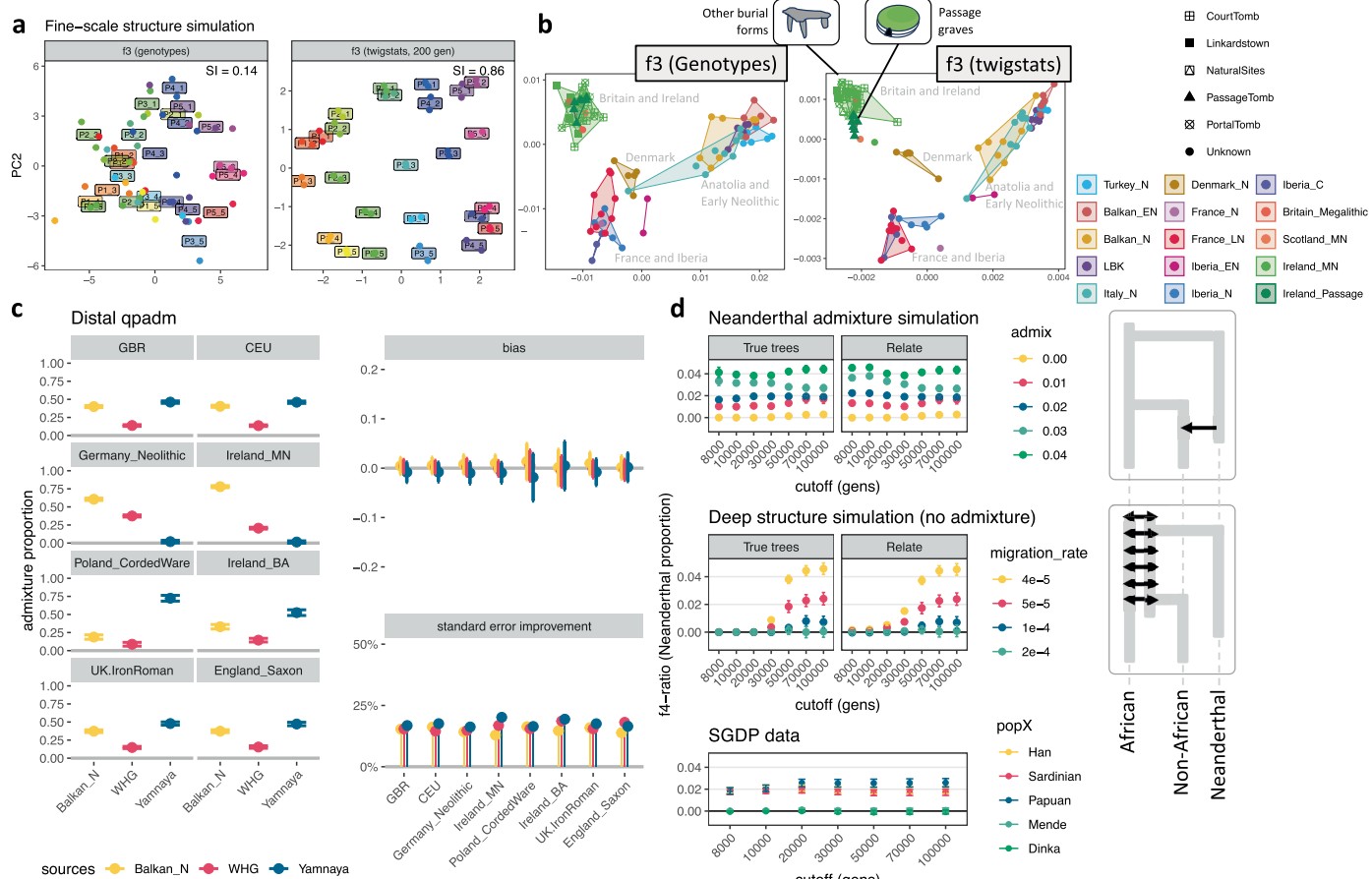

**Extended Data Fig. 5 | Three examples of applying Twigstats. a** Fine-scale population structure simulation emulating ref. 39 (see Methods for simulation details). First two principal components are computed from pairwise outgroup $f_3$ statistics on the genotypes directly and on Relate trees inferred from the 50 target individuals. Labels in plots show the average coordinates of members of that population. For each panel, we calculate a separation index (SI) as in[39], which we define as the proportion of individuals for which the closest individual (by the Euclidean distance in PC space) is in the same population. **b**, Fine-scale genetic structure in Neolithic Europe quantified using an MDS calculated on a symmetric matrix that contains all pairwise outgroup $f_3$ statistics (outgroup: YRI) between individuals. These are either calculated directly on genotypes or calculated using Twigstats on Relate genealogies with a cutoff of 1000 generations. Individuals were selected by filtering based on Steppe and Western Hunter-gatherer ancestry (Methods). **c**, Admixture proportions inferred using qpAdm with three distal sources of Western

Hunter-gatherers, early European farmers, and Yamnaya Steppe people[46]. We show results for Twigstats-5000. Bias is measured as the difference in admixture proportions obtained from Twigstats-5000 and all SNPs, and we show standard errors of the latter. We plot two standard errors around the mean. The standard error improvement shown is one minus the ratio of standard errors obtained from Twigstats-5000 and using all SNPs. **d**, Neanderthal admixture proportion inferred using an $f_4$-ratio of the form $f_4$(outgroup, Altai, target, Mbuti)/$f_4$(outgroup, Altai, Vindija, Mbuti). We compute these on genetic variation data from the Simon's Genome Diversity Project (SGDP)[75] and use the high-coverage Altai and Vindija Neanderthals[78,79]. We also compute equivalent $f_4$-ratio statistics in a simulation emulating Neanderthal admixture 50,000 years ago and a second simulation involving no Neanderthal admixture but deep structure that leads to a similar inference unless deep coalescences are ignored by Twigstats. We plot two standard errors around the mean.

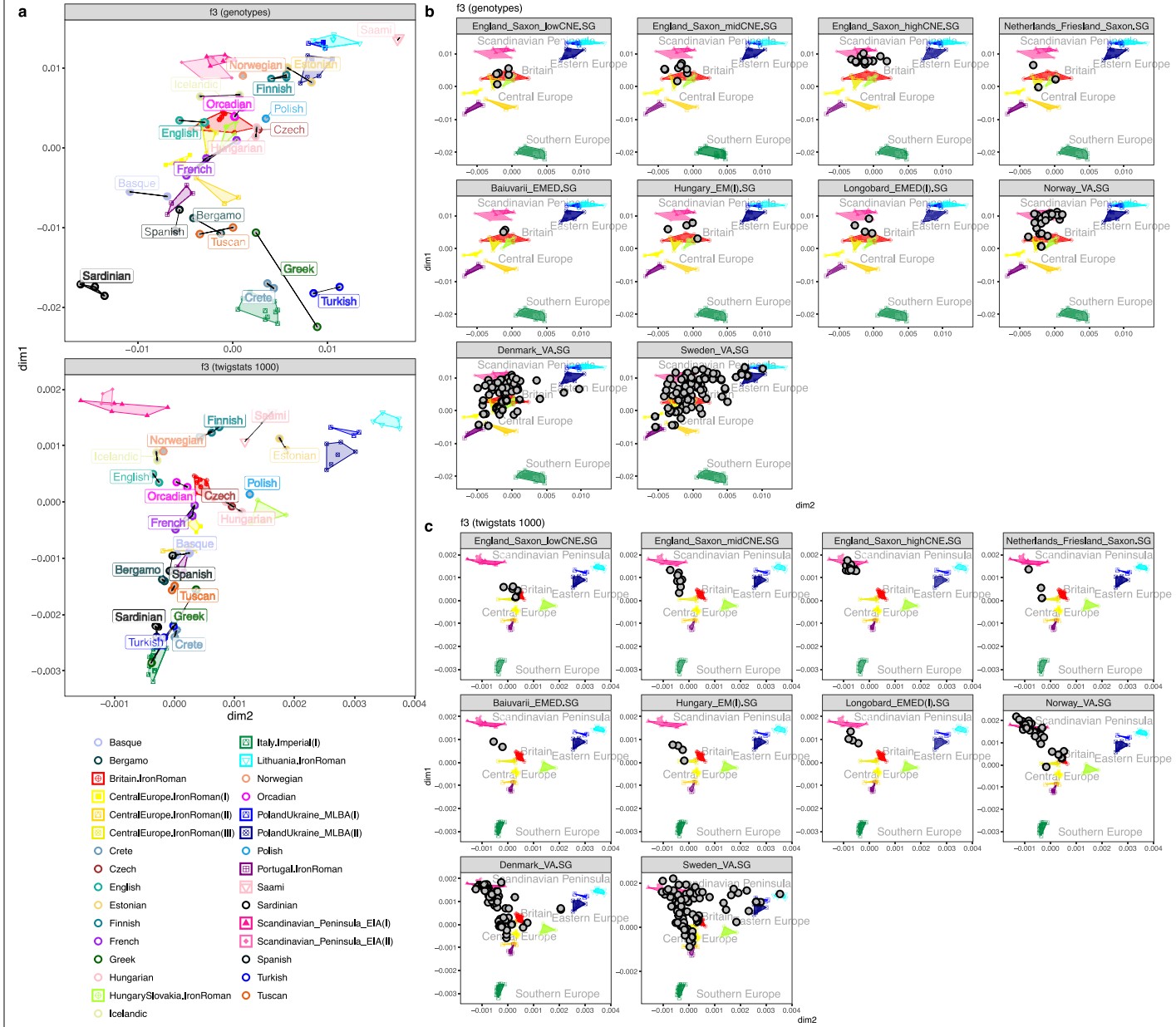

**Extended Data Fig. 6 | MDS of ancient and modern genomes. a**, Same MDS as in Fig. 2 but only showing qpAdm source groups of Fig. 2a and modern groups in the Simons Genome Diversity Project (labelled) computed using genotypes (top) or Twigstats (bottom). **b**, MDS computed using genotypes showing one early medieval or Viking age group per facet. **c**, MDS computed using Twigstats showing one early medieval or Viking age group per facet.

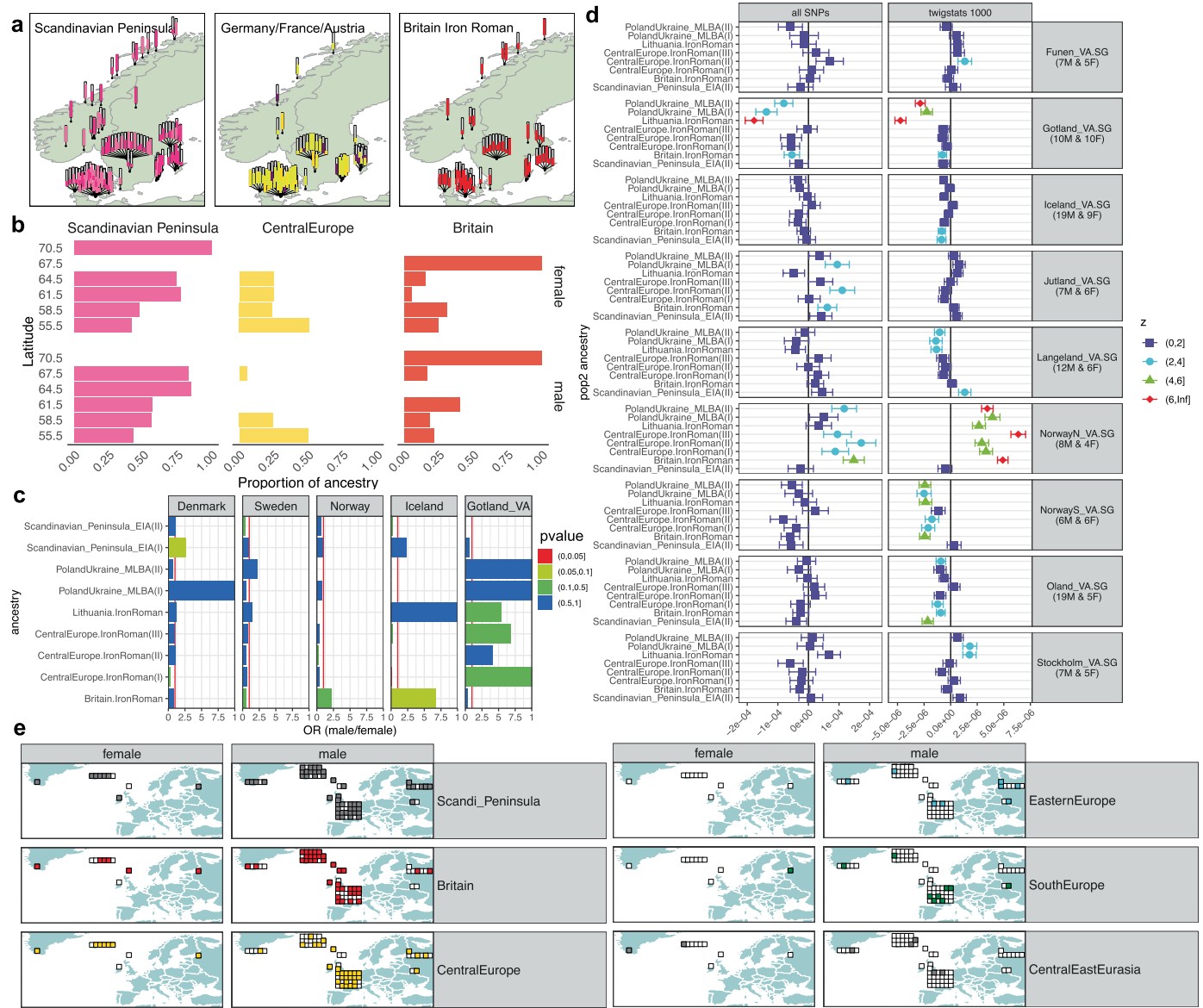

**Extended Data Fig. 7 | Ancestry estimates stratified by genetic sex. a**, Map showing ancestry carried by each Scandinavian Viking age individual. **b**, Ancestry proportions across individuals grouped by Latitude and genetic sex. **c**, Odds ratio and p-values calculated using a two-sided Fisher's exact test on the number of males and females carrying each ancestry in Viking Age Denmark, Sweden, Norway, Iceland, and Gotland. **d**, $F_4$ values of the form $f_4$(Scandinavian_Peninsula_EIA(I), alternative source group, males in Viking group, females in Viking group) computed using all SNPs and Twigstats. A significantly positive value is

evidence of attraction of females with pop2 or males with Scandinavian_Peninsula_EIA(I). Number of males and females is shown in each facet title and we restrict to groups with at least four males and females. We plot one standard error. **e**, Map showing 'far flung' Viking individuals grouped by ancestry and genetic sex. In contrast to Fig. 4a and d where we showed results for the 'best' qpAdm model, here in panels **a, b, c,** and **e**, an individual is assigned an ancestry group, if it has **any** accepted model (p > 0.01) where that ancestry features.

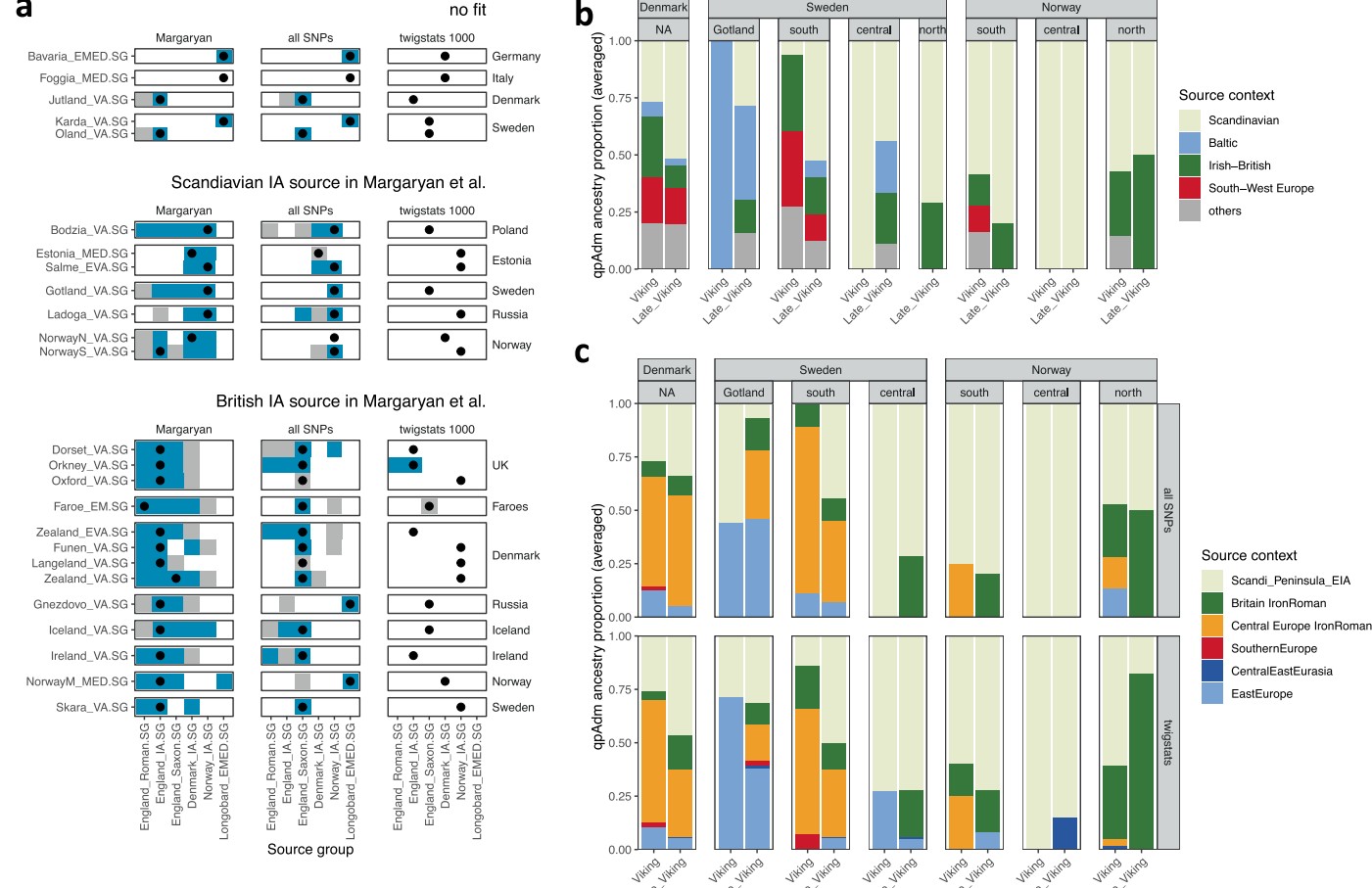

**Extended Data Fig. 8 | Replication previous Viking Age ancestry modelling.**
**a**, P-values of 1-source qpAdm models with target groups shown as rows and source groups shown as columns, replicating Extended Data Fig. 5a of ref. 6 Left column uses p-values obtained from ref. 6. Middle and right column correspond to newly computed p-values in a qpAdm using, respectively, all SNPs and Twigstats-2000. Outgroups are YRI, CHB, DevilsCave_N.SG, WHG, EHG, Anatolia_N, Yamnaya, Estonia_CordedWare.SG (Supplementary Table 1). We excluded Denmark_IA.SG and England_Roman.SG from the rotational scheme as these groups overlap in ancestry with England_IA.SG and Norway_IA, respectively. Only samples with coverage exceeding 0.5 are used. For each target group, the source group with the largest p-value is shown with a black circle.

**b**, qpAdm models of ref. 7 where modern populations are used as sources. As in ref. 7, we show ancestry proportions averaged over individuals in each group, where for each individual the model with the smallest number of sources and largest p-value is chosen. **c**, Replication using the same target samples as in **b**. We fit a maximum of two sources and choose the model with the smallest number of sources and largest p-value, requiring p > 0.01 for 1 source and p > 0.001 for 2 source models. The set of individuals used in **b** and **c** are identical and are comprised of targets with an accepted model in all SNPs and Twigstats-1000, removing 15 of 167 individuals. We additionally remove 17 individuals that did not have a feasible model in ref. 7.

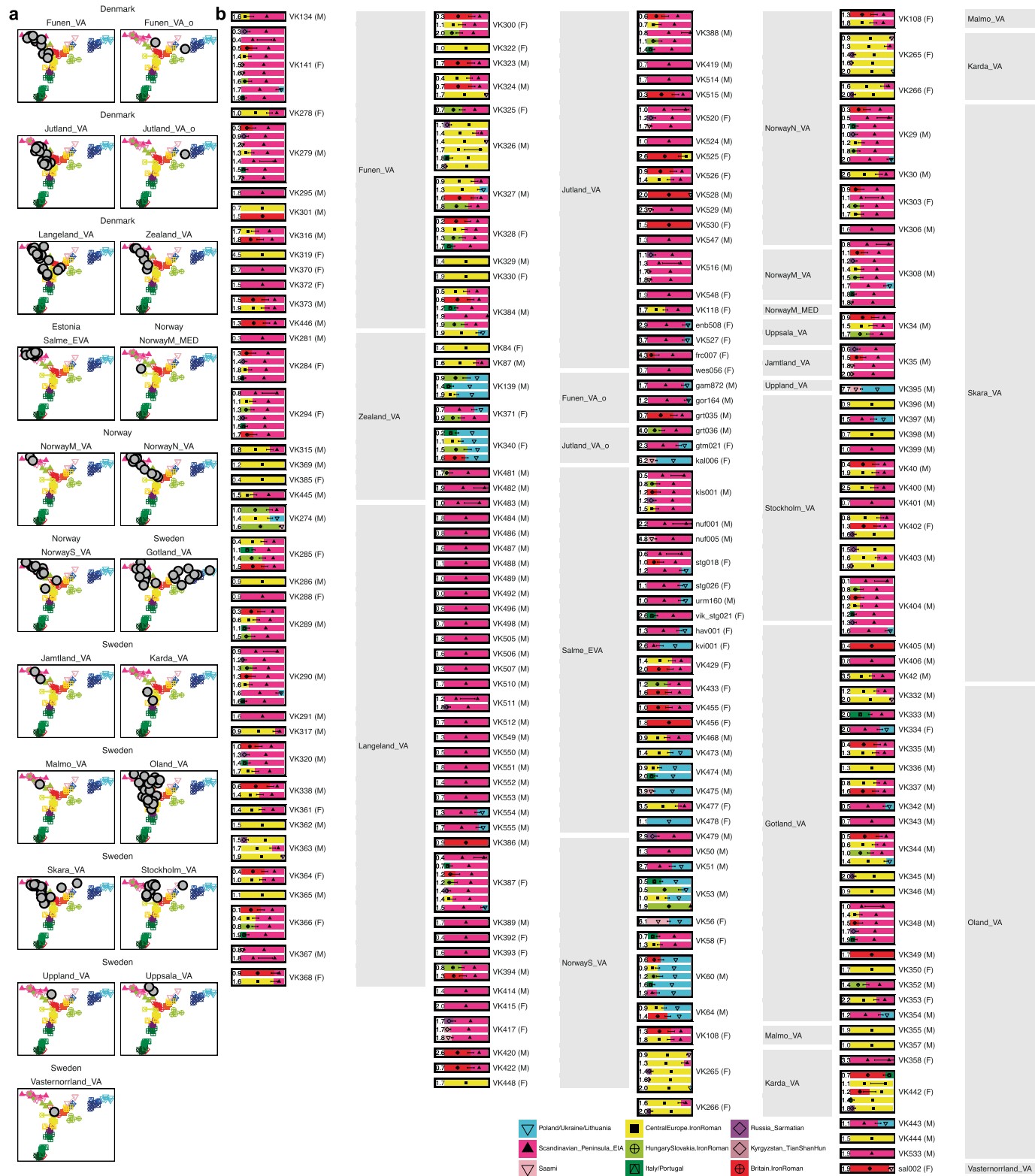

**Extended Data Fig. 9 | Ancestry models of Viking Age individuals in Scandinavia. a**, MDS of each Scandinavian Viking group plotted on top of preceding Iron age and Roman individuals. **b**, All accepted qpAdm models using Twigstats-1000 for every Scandinavian Viking individual in Denmark, Sweden, and Norway, computed in a rotational qpAdm with source groups identical to Fig. 4. We only retain models with feasible admixture proportions, standard errors of <0.25, and show models with 1 source and a p-value greater than 0.01

or otherwise with 2 sources and a p-value greater than 0.01. If several models satisfy p > 0.05, we show all such models, otherwise we select the model with the largest p-value. The -log10 p-values are shown to the left of each model. We combine models involving related sources, if they exist, by averaging their respective admixture proportions, standard errors, and p-values. We plot one standard error.

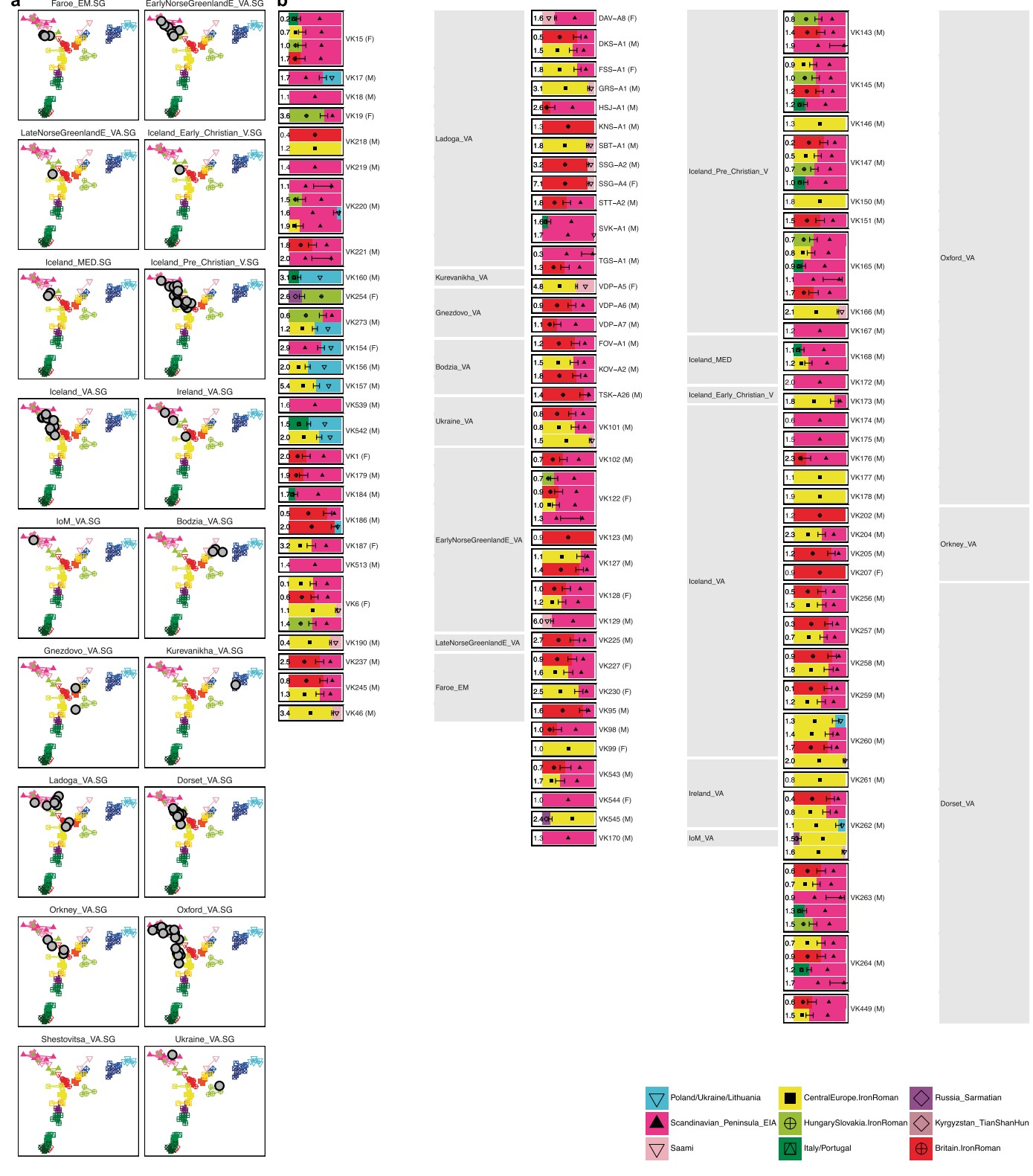

**Extended Data Fig. 10 | Ancestry models of far flung Viking individuals.**
**a**, MDS of each far flung Viking group plotted on top of preceding Iron age and Roman individuals. **b**, All accepted qpAdm models using Twigstats-1000 for every non-Scandinavian Viking individual computed in a rotational qpAdm with source groups identical to Fig. 4. We plot one standard error.

# Reporting Summary

## Statistics

For all statistical analyses, confirm that the following items are present in the figure legend, table legend, main text, or Methods section.

| n/a | Confirmed | |
|---|---|---|
| ☐ | ☒ | The exact sample size (*n*) for each experimental group/condition, given as a discrete number and unit of measurement |
| ☐ | ☒ | A statement on whether measurements were taken from distinct samples or whether the same sample was measured repeatedly |
| ☐ | ☒ | The statistical test(s) used AND whether they are one- or two-sided<br>*Only common tests should be described solely by name; describe more complex techniques in the Methods section.* |
| ☐ | ☒ | A description of all covariates tested |
| ☐ | ☒ | A description of any assumptions or corrections, such as tests of normality and adjustment for multiple comparisons |
| ☐ | ☒ | A full description of the statistical parameters including central tendency (e.g. means) or other basic estimates (e.g. regression coefficient) AND variation (e.g. standard deviation) or associated estimates of uncertainty (e.g. confidence intervals) |
| ☐ | ☒ | For null hypothesis testing, the test statistic (e.g. *F*, *t*, *r*) with confidence intervals, effect sizes, degrees of freedom and *P* value noted<br>*Give P values as exact values whenever suitable.* |
| ☐ | ☒ | For Bayesian analysis, information on the choice of priors and Markov chain Monte Carlo settings |
| ☐ | ☒ | For hierarchical and complex designs, identification of the appropriate level for tests and full reporting of outcomes |
| ☐ | ☒ | Estimates of effect sizes (e.g. Cohen's *d*, Pearson's *r*), indicating how they were calculated |

*Our web collection on statistics for biologists contains articles on many of the points above.*

## Software and code

Policy information about availability of computer code

| Data collection | No software was used for data collection. |
|---|---|
| Data analysis | We used bcftools 1.19, samtools 1.3.1, bwa aln 0.7.17-r1188, GLIMPSEv1.1.1, Relate v1.2.1, and R packages stats (v3.6.2), admixtools2 (v2.0.4). Code for twigstats (v1.0.1) is available through https://github.com/leospeidel/twigstats and https://zenodo.org/records/13833120. |

For manuscripts utilizing custom algorithms or software that are central to the research but not yet described in published literature, software must be made available to editors and reviewers. We strongly encourage code deposition in a community repository (e.g. GitHub). See the Nature Portfolio guidelines for submitting code & software for further information.

## Data

Policy information about availability of data

All manuscripts must include a data availability statement. This statement should provide the following information, where applicable:
- Accession codes, unique identifiers, or web links for publicly available datasets
- A description of any restrictions on data availability
- For clinical datasets or third party data, please ensure that the statement adheres to our policy

All ancient DNA data used in this study was publically available and is listed in Extended Data Table 1. The corresponding accession codes are: ERS2540893, PRJEB11004, PRJEB11364, PRJEB11848, PRJEB11995, PRJEB13123, PRJEB14180, PRJEB14675, PRJEB14737, PRJEB18067, PRJEB20614, PRJEB20658, PRJEB21037, PRJEB21330, PRJEB21940, PRJEB22592, PRJEB23467, PRJEB26760, PRJEB29189, PRJEB29360, PRJEB29360 , PRJEB29603, PRJEB29700, PRJEB31045, PRJEB31249,

## Research involving human participants, their data, or biological material

Policy information about studies with human participants or human data. See also policy information about sex, gender (identity/presentation), and sexual orientation and race, ethnicity and racism.

| | |
|---|---|
| Reporting on sex and gender | We use DNA sequenced from archaeological remains and have inferred the genetic sex where possible. |
| Reporting on race, ethnicity, or other socially relevant groupings | We have grouped ancient DNA samples by expert assigned archaeological context, by time period, by geographic location and by genetic clustering. |
| Population characteristics | We have included samples from Western and Central Eurasia spanning the last 10,000 years. |
| Recruitment | We used publicly available ancient DNA samples. These are subject to sampling bias, that may arise for instance due to burial context. In particular, current technologies are unable to extract DNA from cremation burials which have been frequent in some cultural contexts. |
| Ethics oversight | N/A |

Note that full information on the approval of the study protocol must also be provided in the manuscript.

# Field-specific reporting

Please select the one below that is the best fit for your research. If you are not sure, read the appropriate sections before making your selection.

☒ Life sciences   ☐ Behavioural & social sciences   ☐ Ecological, evolutionary & environmental sciences

For a reference copy of the document with all sections, see nature.com/documents/nr-reporting-summary-flat.pdf

# Life sciences study design

All studies must disclose on these points even when the disclosure is negative.

| | |
|---|---|
| Sample size | We aimed to compile a close to exhaustive list of ancient genomes with Western and Central Eurasian ancestries and then filtered by sequencing technology (shotgun sequencing), sequencing coverage (>0.5x), and excluded close relatives. Our final dataset comprised 1,151 genomes in total. |
| Data exclusions | We only used samples that were sequenced genome-wide to an average sequencing coverage of 0.5x. We excluded close relatives. |
| Replication | We conducted two replication analyses of previous work (Extended Data Figure 8) to make sure our findings are consistent with current knowledge. We conducted non-parametric and parametric modeling to confirm that our findings are robust to some modeling assumptions. |
| Randomization | We ran ancestry models both on a per individual basis, as well as grouping individuals according to archaeological context provided by the reference and as detailed in SI Table 1. To select source groups in our ancestry modelling, we used a clustering approach described in the Methods section. |
| Blinding | We used existing data and so blinding was not possible. |

# Reporting for specific materials, systems and methods

We require information from authors about some types of materials, experimental systems and methods used in many studies. Here, indicate whether each material, system or method listed is relevant to your study. If you are not sure if a list item applies to your research, read the appropriate section before selecting a response.

## Materials & experimental systems

| n/a | Involved in the study |
|---|---|
| ☒ ☐ | Antibodies |
| ☒ ☐ | Eukaryotic cell lines |
| ☒ ☐ | Palaeontology and archaeology |
| ☒ ☐ | Animals and other organisms |
| ☒ ☐ | Clinical data |
| ☒ ☐ | Dual use research of concern |
| ☒ ☐ | Plants |

## Methods

| n/a | Involved in the study |
|---|---|
| ☒ ☐ | ChIP-seq |
| ☒ ☐ | Flow cytometry |
| ☒ ☐ | MRI-based neuroimaging |

## Plants

| | |
|---|---|
| Seed stocks | *Report on the source of all seed stocks or other plant material used. If applicable, state the seed stock centre and catalogue number. If plant specimens were collected from the field, describe the collection location, date and sampling procedures.* |
| Novel plant genotypes | *Describe the methods by which all novel plant genotypes were produced. This includes those generated by transgenic approaches, gene editing, chemical/radiation-based mutagenesis and hybridization. For transgenic lines, describe the transformation method, the number of independent lines analyzed and the generation upon which experiments were performed. For gene-edited lines, describe the editor used, the endogenous sequence targeted for editing, the targeting guide RNA sequence (if applicable) and how the editor was applied.* |
| Authentication | *Describe any authentication procedures for each seed stock used or novel genotype generated. Describe any experiments used to assess the effect of a mutation and, where applicable, how potential secondary effects (e.g. second site T-DNA insertions, mosiacism, off-target gene editing) were examined.* |

nature portfolio | reporting summary

April 2023

