## [Peer Review File · Nature]

High-resolution genomic history of early medieval Europe

Corresponding Author: Dr Pontus Skoglund

Version 1:

Reviewer comments:

Referee #1

(Remarks to the Author)

This is a ground-breaking paper, developing cutting-edge methods to shed light on important questions of broad and general interest. The new "Twigstats" method builds on recent breakthroughs in the inference of genome-wide genealogies, and cleverly combines tried-and-trusted f-stats methods with the information in these newly available genealogical trees. The application of this new method to the large dataset of ancient genomes provides a major increase in statistical power, and an array of new insights into the population structure of medieval Europe. The paper is very well written and clear on all points.

I have one point of mild methodological concern which could be addressed with some additional simulations, and would strongly encourage the authors to share their simulation and evaluation code (see below). I also have some minor comments on the text and presentation.

High level comments

- Although I'm not certain without being able to look at the code (see next point) I'm concerned that the evaluations here are being done on quite simplistic simulations, which may not reflect the properties of the data used in reality. Since Ancient DNA data is very noisy, evaluations based on trees built from perfectly clean simulated data may not be fully indicative of real world performance. I would like to see some additional analysis like Fig 1b, c, d where some errors are thrown on the data. Although methods for generating such noise realistically is an open problem, a reasonable first step could be to use the empirical model of Albers & McVean 2020, to at least show that the results are robust under *some* forms of error, if not the forms of error that we might expect from ancient DNA. Some discussion on these points in the methods would also be welcomed.

- I think the code used to run the simulations and evaluations here should be made available. Ideally all code and scripts used to generate the Relate trees from public data as well should be made available, both for scrutiny as well as to provide a resource to the community.

Minor comments

- L39. "However, tracing history using ancestry has remained challenging..." It's not clear to me what "ancestry" means here, and I'm not sure what the following sentence really refers to. Perhaps a sentence describing what you

mean by "ancestry" immediately after the first "ancient genome sequencing" sentence would help clarify this vital point.

- L70. The applications of ARGs is reasonably comprehensive here, I think the "among others" is unnecessary.

~L71. It would be good to cite the recent ARG reviews here somewhere as starting points for readers new to the literature. Perhaps at the end of the "Genealogical trees ..." sentence.

- The era of the ARG: An introduction to ancestral recombination graphs and their significance in empirical evolutionary genomics, PLOS Genetics 2024
- The Promise of Inferring the Past Using the Ancestral Recombination Graph, GBE 2024

- L156-163. Does this need some citations? This is entirely out of my expertise, so I don't know whether historians would agree with these statements. Some references would be reassuring.

- Fig 1. It took me a while to figure out that the colours in (c) are defined by the legend at the bottom. I think what confused me was the "dots" in the middle of the colour lines in the legend. Because there's no dots in panel (c), it wasn't obvious to me that they are the same. I think you could add a legend to fig C without losing too much? Another approach might be to use a dotted line in D for the theoretical prediction rather than a blue line, and then you wouldn't need a legend for D. It might then be more obvious that there's only one global colour scheme.

- Fig 2a. Caption. "between individuals calculated as conventional directly on genotype calls" reads awkwardly, consider rephrasing.

- Fig 2b. It's not clear what the SGDP panel is here. Explain. Generally the caption in this figure needs more clarity I think.

- Fig 3. "UK and Ireland" should probably be "Britain and Ireland".

- L564. I would like to see the code used to run these simulations and the various evaluations. While the code for Twigstats is mentioned in the Data availability section, the simulation and evaluation code for generating the figures in this paper is not.

Referee #2

(Remarks to the Author)

In this paper, Speidel and colleagues develop a new approach to examining patterns of admixture between populations, called Twigstats. Essentially the method combines two existing approaches for modeling population genetic data (coalescent tree reconstruction and f-statistics) to provide more powerful insights, particularly with regard to differentiating subtle population structure in ancient populations. They then demonstrate the robustness of this approach by testing it on various questions in human population genetic history, either using simulated data or replicating (and in many cases enhancing) analysis in well-established demographic scenarios. Finally they apply their method to existing, previously published Bronze Age, Roman and Medieval whole genomes to gain new insights into putative population movements described in the historical record, with a particular focus on the movement of Germanic-speaking barbarians from Northern Europe into the former Roman Empire and the impact on and by medieval Scandinavian populations such as the Vikings.

First of all I will say there is a lot to like about this paper from a purely population genetics methods perspective. The idea of using the underlying genealogies (as estimated by the primary author's own software, Relate) to estimate f-statistics, rather than the messier allele frequencies is really very clever. Further, the focus on using more shallow genealogies to extract younger coalescent events that are more relevant to discriminating more recently diverged populations is also a very powerful insight. The theory presented in the supplement appears sound to me, and the performance on simulated and real data prove this method works well and provides additional insights beyond that was possible using observed allele frequency-based f-statistics alone. I do agree with authors in that this paper is a good argument for why the human paleogenetics community will need to transition from capture-based genomes like 1240k to full whole genomes if we are maximize the insights gained from ancient DNA.

However, I do not believe the study rises to the level expected of a Nature paper for the following reasons.

- 1) The method itself is an incremental improvement on already existing methods (though done in a very neat way), providing a refinement in terms of insights rather than transforming how we can look at genetic data from a pop gen perspective (for example Li and Durbin 2011 with PSMC).
- 2) The application of this method is to existing published data. The results from this are a series of interesting additional insights to existing studies, but derived in a very post-hoc way. I can imagine that the application of this method will be extremely powerful when combined with a robust sampling of ancient whole genomes from multiple time points to answer a specific question (i.e. with a robust study design). But at the moment the conclusions are very coarse with lots of caveats that derive from the sampling time and geography (which of course the authors have no real control over). There is not any particular major take home message from the results of the analysis that is particularly striking or specific to me, but rather a series of intriguing vignettes that could be fruit for future in-depth study. The overall conclusion of the abstract that “our results are consistent with substantial mobility in Europe in the early historical period” is not a major leap in our understanding of the post-Roman Medieval period beyond what is known already from history, archaeology, or even recent previous paleogenetic work. In many ways this paper is equivalent of other recent papers (including by this primary author) where an exciting new method has been developed for aDNA and then applied to existing data (Ringbauer et al. 2021, Ringbauer et al. 2023, Speidel et al. MBE). I certainly do not think this manuscript rises above any of these, and I think a journal like Nature Genetics would be a good home for this paper.
- 3) I also am not sure if the general approach of this method (that population in the future are mixtures of distinct ancestral populations) is the best model of the data from the time periods being examined. The expectations of f-statistic-based approaches such as f4 ratio tests, qpwave and qpadm are based on a fairly explicit model of population divergence with a pulse-like admixture. While they can be robust to some violations of this model, probably one of the most extreme violations would be case where there is continuous geneflow between populations, for example in an isolation-by-distance or stepping-stone model. Indeed Harney et al. (2011) examined this and concluded that they “caution against ... analyzing population histories involving extended periods of gene flow”. The overall patterns of population structure in Europe from the Bronze-Age onwards (Antonio et al. 2024, which happens to be where much of the data used in this study are from) are very much in line with a model of isolation-by-distance and largely seem to extend to the kind of patterns we see in modern Europe (Novembre et al. 2008). Yet by nature of how the f-statistic approach is applied, the authors of this study must define discrete “ancestral populations” and that subsequent populations/individuals are then modeled by these discrete groups. While of course I recognize this can be a useful way of modeling the data, I argue that this is likely so far from the reality of actual spatial structure of the populations, that it provides a potentially false view of how these populations are derived at multiple levels. And often the general patterns observed by the authors very much seem like they are probably just isolation-by-distance (for example more Austria/France/German ancestry in southern Scandinavia), but the framework is forcing us to think of them as mixing of distinct sources. The authors have some interesting simulations looking at the effectiveness of their approach in situations with geneflow rather than pulses of admixture, but not sufficiently to deal with my concerns above. For example extended data figure 4a shows the impressive fine-scale resolution that can be obtained with their approach even for populations connected by high migration. But this only looks at one particular aspect derived from f-statistics, which is via an outgroup f3 statistic, so one shared drift path between two populations (that will represent a mix of geneflow and divergence, rather than just pure divergence). This does not necessarily tell us how the method will perform in a qpadm or qpwave context where now it is about the extent of the sharing of a complex network of drift paths under extended geneflow. Similarly, the method seems to do an impressive job at distinguishing deep African population structure with lots of past geneflow from a pulse of neanderthal admixture, but I do not think this is that relevant to the question of recent extended geneflow (and I assume much of the power comes from cutting off the trees to limit the impact of that deep structure, which will tend to extend coalescence times).
- 4) The authors define the source groups by applying qpwave on Bronze Age and Roman samples to find “homogenous groups” via UPGMA clustering. There are two potential issues or concerns I have with this. From my point above, the suitability for qpwave to do this on populations exhibiting extended geneflow is at best, unclear. However, this was also the approach followed by Antonio et al. so I understand why this was done, and could potentially provide at least some decent proxy of ancestral populations. However, Antonio et al. were largely identifying “cluster” to better characterize “outliers” from the same period, so the purpose is not really the same as used in this study, which is to model all subsequent individuals in the post-Roman period. And while I am sure the new Twigstats approach can increase the refinement of these “sources”, any real insight I think is limited by the very sparse sampling available to the authors. This leads to the somewhat weird description of sources that consist of Portugal and Lithuania as distinct sources, but also three sets of Austria/France and Germany samples being distinct sources, and five sets of sources from “Italy” that as far as I can tell all derive from a strip of central Italy in and around Rome. This kind of lumpy and uneven “source population” attribution is almost certainly driven by the very limited and uneven sampling. I highly doubt that such a distribution of groups in time and space would emerge with a more comprehensive sampling of Europe. While an argument could be made that the approach of the authors is a statistically “objective” way of choosing sources, the underlying choice of samples was not, which obviously contaminates for the former. There is also the question of what these sources even represent. If three samples from the same “population” (R11557, R11552, R11560, all France_Sarrebourg_LateAntiquity.SG) are put into two source “populations” by this approach, and one of these source populations also includes an individual from Austria (R10659, Austria_Klosterneuburg_Roman.SG) a 1000km away, what does it mean if a subsequent person has ancestry from both these sources? There appears to be much more resolution in the Scandinavian analysis due to the better, denser, sampling, and some of the inferences can be quite nuanced (hence I assume the emphasis on this section), showing the importance of sampling and study design to make this approach valuable.
- 5) There are some interesting insights about certain individuals and their ancestry, but usually we are left with very coarse descriptions of the actual processes that might have led to these patterns and their dynamics, usually not much beyond what

was described the original study the data derived from, and certainly not with any precision to make conclusive statements about historical events. This is not a fault of the authors, and in some ways I am glad they are (mostly) cautious, but is the limitation of the data and sampling. While I see Peter Heather is one of the co-authors, I think most medieval historians would be somewhat underwhelmed by the insights that derive from these ancestry inferences.

For example, I highlight some statements below that while sometimes intriguing, are very superficial from a historical point of view:

“The Wielbark archaeological complex has been associated with the Goths, a Germanic-speaking group, but this attribution has remained unclear. Our modelling supports the idea that some early Germanic-speaking groups expanded into the area between the Oder and Vistula rivers”

So does this resolve that they were Goths? Probably not.

“The presence of a southern European-like ancestry component (represented by Roman Italy), which is not present in the Lithuanian Roman Iron Age, could be consistent with models of admixture taken place further south and arriving in Poland through north-westerly Slavic expansions.”

What models of admixture are these exactly, are they associated with specific historical events?

“Later Migration period individuals only have partial Scandinavian-related ancestry, suggesting later broader ancestry shifts in the region.”

What exactly are these broader ancestry shifts? This almost is meaningless, other than saying, their ancestry seems to have change.

“we observe Scandinavian-related ancestry components in several Longobard burials (Longobard_EMED(I)) dating to the middle of the 5th century CE46 (Figure 3b)...The Longobards were dominated by Germanic-speakers, and our result is consistent with attestations that they originated in the area of present-day Northern Germany or Denmark. In contrast, several other Longobard-associated individuals from the region(Longobard_EMED(II)) show ancestry more closely related to continental Europe, putatively representing local ancestry prior to Langobardic influence. This is consistent with the notion that the Longobards of the fifth and sixth centuries headed a confederation of several different groups.”

This statement is actually quite troubling. It seems that the authors have decided that some burials are Longobard, because they have Scandinavian ancestry, but those that have Central European ancestry are simply Lombard-associated and are probably local. This to me is bordering on genetic essentialism and certainly somewhat circular.

“In southern Germany, the genetic ancestry of individuals from early medieval Bavaria likely associated with the Germanic-speaking Baiuvarii cannot be modelled as deriving ancestry solely from earlier groups in Iron Age Germany. Our current best model indicates a mixture with ancestry derived from EIA Peninsular Scandinavia, suggesting a regional ancestry shift and expansion of Scandinavian-related ancestry (Figure 3b)”

Again, a regional shift with no real explanation about the processes underlying it or the timing.

References

- Li and Durbin. Inference of human population history from individual whole-genome sequences. Nature 2011
Ringbauer et al. Parental relatedness through time revealed by runs of homozygosity in ancient DNA. Nature Communications 2021
Ringbauer et al. Accurate detection of identity-by-descent segments in human ancient DNA. Nature Genetics 2023
Speidel et al. A method for genome-wide genealogy estimation for thousands of samples. Nature Genetics 2023
Speidel et al. Inferring population histories for ancient genomes using genome-wide genealogies. MBE 2021
Harney et al Assessing the performance of qpAdm: a statistical tool for studying population admixture. Genetics 2021
Novembre et al. Genes Mirror geography within Europe. Nature 2008
Antonio et al. Stable population structure in Europe since the Iron Age, despite high mobility. ELife 2024

Referee #3

(Remarks to the Author)

This is an important paper and one that is publishable in Nature. The development of Twigstats is significant in moving away from using modern populations in aDNA research and will have a major impact on the field. The Data also points to an important but previously unrecognised migration into pre-Viking Age Scandinavia. There are however some points that the authors need to attend to. These are outlined below, and many relate to terminology which is inconstantly presented, and confusing it often mixes regions, periods and genetic groups and need to be made clearer. There is a tendency towards conflating individual mobility vs migration, but this may be the result of loading of interpretation onto individual burials. Terms like Tribe, Germanic speakers Slavic Speakers should be avoided (tribes are specify units of social organisation and genetics cannot tell you how an individual spoke) and genetic affinity and chronology become conflated and or are difficult to separate in some places. Perhaps adopt ancestry as a term as other papers have to avoid these mix-ups.

This is an enormously complex topic and paper, with considerable implications for 1st millennium Europe and is likely to be controversial, it is therefore important to be clearer. The early sections on Roman and iron age Scandinavia are the most problematic and too many words are spent on straw men for example, Neanderthals, the Roman Gladiator, and Female

settlement vs Male mobility interpretations tend to imply a bias in interpretation that is not favourable to the hard work of the authors. I'd recommend reducing and simplifying these sections so that there is more space for the details that are needed to convey the importance of the Ancestry Changes in late Iron Age Scandinavia onwards doing so will not remove from the importance of Twigstats. These sections convey the implication of then DNA data well, but the terminology used, the oversimplification of and absence of archaeological details weakens them. Because of the tendency to refer to people as genetic and not cultural groups the paper does not convey as well as it could the hugely complex melting pot of genetics and cultures that are present in Northern Europe that than it could and given that this is what it identifies this is a challenge. In this paper the Twigstats approach is able to distinguish between Scandinavian IA and Austrian-French- German (AFG) Iron Age ancestry. The published Isotopic Data (Leggett) and the early medieval English population in Gretzinger et al (their Ref 37) indicates that the CNE genetic groups identified were a mixture of these two it would be of considerable benefit then to perform a Twigstats analysis on that CNE population to see if there is a difference between those in SE England (which we might expect to be AFG) and North eastern England (which we might expect to be Scandi IA) particularly as these genetic groups seem to be together in England at the same point as there is conflict between them in southern Scandinavia (as presented here).

As an aside given the genetic tree analogy and the temporal and geographic focus of the paper I'm surprised they did not call Twigstats Yggstats after Yggdrasil the early medieval Scandinavian mythological tree of life.

Some Ref to follow up on in relation of the points raised in this review:

Leggett, S., Hakenbeck, S., & O'Connell, T. (2022, June 9). Large-scale Isotopic Data Reveal Gendered Migration into early medieval England c AD 400-1100. <https://doi.org/10.31219/osf.io/jzfv6>

Hamerow H, Leggett S, Tinguely C, Le Roux P. Women of the Conversion Period: a biomolecular investigation of mobility in early medieval England. *Antiquity*. 2024;98(398):486-501. doi:10.15184/aqy.2023.203

Hakenbeck S.E.(2009). 'Hunnic' modified skulls: Physical appearance, identity and the transformative nature of migrations. H. Williams and D. Sayer (eds.), *Mortuary Practices and Social Identities in the Middle Ages. Essays in Honour of Heinrich Härke*. Exeter: Exeter University Press. 64-80.

Detailed comments

98 theoretical prediction and simulation – be more specific please, what is theoretical prediction?

135-145 – Iron Age and Northern Europe - that is fine from a methodological perspective but it would be good to have the implications of this spelled out more clearly.

Fig 3 – the label used here are opaque – England Saxon is not a term used in this period, and what is England Gladiator? Why use an ethnic term that has no historical basis in one place and an interpretive term based on superstition in another – is the York sample a gladiator or a Roman soldier?

156-159 – this is a significant oversimplification of the fall of Rome and needs further clarification, and references. I'd also recommend avoiding loaded terms like the 'migration period' provide dates here instead. Which Roman historians have written about this?

160-16 – if the nature and scale of these things has been questioned then there should be references to this important debate here.

165-175 – very important and powerful, this is the key contribution of this paper so I'd recommend flagging this up more prominently in the introduction.

180-181 – great but perhaps this needs to be more transparent in the numbers and reality of these samples, there are not that many samples from Roman Britain presently, for example, how many are used here and are they representative of the Roman population or specific to cosmopolitan area.

203 – again where are the references to this contested association, what do you mean by 'Goths' where these one group by this time, and why associate them with Germanic speaking this is a supposition imposed by Roman writers and is potentially contested by these results so this needs to be developed and presented in a more representative and holistic fashion.

219 – represented by Roman Italy how? Roman Italy attracted immigrants from Northern Europe and North Africa as well as Asia so please explain why you can be sure your samples are not unique individuals.

221 – Slavic expansions needs justifying as a statement to genetics.

227 – why is this single 50-60 year old female attributed to nearby settlement and not long-distance exogamous marriage that we know to have taken place in the 1st century? It's not clear how the data supports this conclusion, or how these references used do either.

229 – later ancestry shifts – or perhaps migration and integration of Scandinavian and local populations suggesting settlement.

236-241 – or that the Lombards were a migratory group made up of people from different ancestries and not a biologically

district group but a culturally evolving conglomerate like other overland migrating populations.

248-253 this is a quite a fascinating result – are other sources available?

260-263 – very interesting result, be careful of a gender bias in the authors imprecations. Earlier a single female was evidence of nearby settlement, whereas now two men are soldiers and evidence of non-local mobility. What date are these soldiers, you provide dates elsewhere? Cavalry is also significant there were a number of auxiliary cavalry forts in Northern Britain's – Ribchester, Chesters, for example, these were populated early by Spanish or Sarmatians, that presumably intermixed with the local population.

267 – is this date secure? I thought there was no date for this person so they could be 2-4th century – also the way you talk about them highlights the inaccuracy of the label on fig 3, the gladiator interpretation is contested. It is interesting that this individual might have EIA Scandinavian ancestry – a result which confirms the point made in ref 56 where they are identified as probably CNE foederati, so this point is valid but entirely too strong, the presence was not missed as suggested but there is a difference between mobility in a few individuals and a 'substantial influx'. It is also unclear how a Gladiator 'condemned to die' would contribute significantly to the gene pool – many gladiator contests were staged and not fatal as Gladiators themselves were too valuable to kill in this way.

Moreover, what does this new data say about the CNE ancestry identified in ref 37 how much of that is Scandinavian? Esp in sites like Lakenheath in Suffolk where Swedish ancestry is implied in the historical sources (the Wuffings for example). Given lines 351-356 this would be important to address.

285-288 – the chronology and group terminology become conflated in a confusing way. Are Iron Age groups genetic groups similar to Iron Age groups, or chronological?

283-299 – Iron age groups from Austria, France or Germany. Moving into Scandinavia in 500-800CE and afterwards (see point above) it would be very interesting to know the genetic similarity between these AFG groups, and the CNE ancestry seen in ref 37 can the AFG and Scandinavian ancestry be separated if present?

This section is very important, potentially identifying Northern European Migration not attested in the contemporary literature (or at least not recognised) as that literature is very limited at this period. It would be useful to connect it broadly to the wider issue migration issue from these places the Lombards and CNE ancestries identified before in this paper.

301-306 needs expanding many deposits are so vague as to meaningless.

323-328 – what do you mean by continental-related or 'continental'-related ancestry and line 347 continental ancestry. Scandinavia is continental Europe and are these three different things or the same thing? Also when describing Sweden 28 out of 74, but Norway 2 out of how many from Norway? And central Sweden 'relatively few'. Meaning? Are the authors trying to hide the relatively few samples used in this study or is it just vague and slightly opaque? As with 301-306 many deposits, this need tightening up.

348-349 – after the Oland massacre and thus unconnected or was this part of the impact of migration? Along with the 324 – a map more clearly defining this ancestry and its impact would be helpful here. Fig4b is useful but large amounts of data are presented and not discussed.

354 – sure Iron Age Britain, but by the Viking age Britain is a mixture of Iron Age and CNE ancestry, so that IA ancestry may only describe the west of Britain. What about the rest? Could this presence in Norway be detected because of Norway's association with Ireland, NW England and Scotland, rather the British ancestry which would include England. The Dane Law is in Southern and Eastern England so it surprise not to see more – is it visible?. Although you go on to this in lines up to 361 this is unclear purchase instead of Britain – Eastern England, and REF the relevant publications. How close is CNE (REF 35) and Scandinavian ancestry. And 358 what is British-Related ancestry there is no such genetic group?

368-371 i don't think Imperial Roman and Portugal can be take as representative of southern Europe as a whole so tighten up this expression.

376 – do you mean in males via the Y chromosome, and therefore in male ancestry?

389-392 – great but you have not explained these sites or told us they are execution/massacres.

410 – there is no such thing as Anglo-Saxon related ancestry. English early Medieval perhaps?

415-417 – So an EIA Scandinavian ancestry, British Iron Age ancestry, and an abundance of continental ancestry – the Viking Age burials included a mixture of people with different ancestries – so there is no such thing as a genetic Viking but rather these were drawn from the diversity of people represents in Sweden or Norway? – this is an important point but the way this is presented it is not quite made clear.

And Finally given the point about AFG ancestry and its impact on Scandinavian genetics and the indistinguishability of English early medieval ancestry (as opposed to British Iron Age) from AFG ancestry, as well as the missing British (overall) ancestry that has been seen in previous studies it would be important to figure how to remove the geographic influence on the terminology so that it could be discussed in a way that does not imply that this migration might originate only in the places identified in the genetic terminology i.e Austria, France and Germany.

Version 2:

Reviewer comments:

Referee #1

(Remarks to the Author)

The authors have addressed my comments very well, and I appreciate the thorough explanations given. I am fully satisfied that this paper is ready for publication in Nature, and I am excited to see it in print.

Referee #2

(Remarks to the Author)

I appreciate the efforts of the author's to respond to my critiques. Clearly they have thought deeply about them.

The new isolation-by-distance simulations presented in Ext Fig 3F are very positive, and I am glad that they have such power to reject a pulse model when there is an IBD pattern. However, as far as I can tell, these simulations do not specifically get at the question of what the power of the method is when there is a pulse of migration on top of a background IBD pattern, which are the likely dynamics of the migration period. For example how would the model fair if after establishing the 1-D stepping stone model, there was a pulse of admixture from 1 into 4, and then a continuation of constant 1-d migration? It would be useful to see that the method was robust to such a process. However, the new simulations are still a great addition to the paper.

While I appreciate the additional work to define the source populations, it still seems problematic to me from an interpretation perspective. That clusters from qpwave fall in close proximity in an MDS is not really a surprise for me. What is essentially being described is the former is a particularly close genetic relationship, which then get recapitulated in the latter. I do not argue that the samples qpwave finds are not genetically coherent groups in terms of being highly similar to each other. However, if anything the MDS shows for Central European populations the overlap and proximity of some of these groups genetically, and then overlaying this on the map some very awkward geographic relations. So my same questions largely remain from my initial review in terms of how we then interpret an individual whose ancestry is determined to come from these groups. But I suppose if the authors remain fairly ambiguous and careful when an individual draws their ancestry from a "Central European" group then I guess this is ok, but not particularly ideal or powerful.

Regarding the appropriateness for Nature and the author's response, while I still maintain (without taking away from the quality of the work) my position that I personally do not believe this method is as great an advance as the authors purport in their response compared to other comparative pop gen methods, this is ultimately of course highly subjective and an editorial decision, and certainly would not begrudge or be disappointed if the paper were to appear in the journal if the editor decided.

The results still also read as a series of vignettes, though worded in a more appropriate way thanks to reviewer 3 in particular. While the authors describe changes in ancestry, the results still do not really get down to the process underlying these migrations. The take home message for the Germanic migration sections is essentially there have been different levels of germanic migrations in different parts of Europe, which I still feel is somewhat superficial. Yes, there are some new evidence of migration in Scandinavia not observed before, but that migrations have taken place during the medieval period has never really been in question. What has been of debate are the forces driving these migrations, the mechanistic processes by which they took place, their size, their impact and the legacy, not just with regard to modern genetic ancestry, but the development of nation states and their culture. The findings of this paper, though interesting do not really get at these facets of history. In addition it is not clear to what extent the results are evidence of large scale migration, perhaps establishing permanent settlements, versus small scale mobility of for example military units or elites. So much of understanding of what these ancestry profiles mean in the wider realm of migration versus mobility depends on the sampling strategy and intensity and interpreting the archaeological context of the samples (these are not just random samples from a particular place and time).

I am not a fan of the new title. It gives the impression that prior to this study, we did not believe there was any migration in medieval Europe, and the genetic analysis has finally revealed it to us. I also do not see evidence of "migration waves". Migration waves implies there are periods of high migration and low migration (peaks and troughs of high and low migration rates). But this study does not really characterize these dynamics through time, they just offer instances where ancestry changes based on sample availability

Finally, I understand and appreciate the huge work that has gone into the study and the desire to show all the results as much as possible and not relegate work to a largely unread 200 page supplementary text as has commonly been the norm in ancient DNA. However, I think the figures need some revisions. The various panels are often just so small. I had to spend a long time zooming in on specific sections to see them, and then trying to cross reference that with the legend. It made it prohibitively difficult and time consuming to cross reference the text and figures, and in an actual print copy many of them would probably require a magnifying glass to see. It would be better if some of the sub-panels were transformed into their own figures. Almost every figure bar Extended data figure 2 had some section that was difficult to read and required a lot of

zooming, though extended figures 3-5 are probably the worst culprits.

Referee #3

(Remarks to the Author)

Thank you for your patience, I always juggle with family and digging commitments.

The authors have made a very good and detailed response to the comments, they have expanded and explained, removed inaccurate or misleading terms and increased the detail for key archaeological examples in a useful way. The paper is easier to follow, and more precise, its contribution - via new statistical approaches and the results that have been flagged up in their samples are important and suitable for publication in Nature. Being able to use aDNA as statistical sources without modern populations is very important, and the case studies highlight differences in migration patterns, and unknown migrations that will be debated and explored for many years behind the small samples presented here. I'd be very happy to read this paper in Nature, and look forward to it being published.

Response to reviews

Referee #1 (Remarks to the Author):

This is a ground-breaking paper, developing cutting-edge methods to shed light on important questions of broad and general interest. The new "Twigstats" method builds on recent breakthroughs in the inference of genome-wide genealogies, and cleverly combines tried-and-trusted f-stats methods with the information in these newly available genealogical trees. The application of this new method to the large dataset of ancient genomes provides a major increase in statistical power, and an array of new insights into the population structure of medieval Europe. The paper is very well written and clear on all points.

We thank Referee #1 for the positive comments and recognition of the advance that the study represents.

I have one point of mild methodological concern which could be addressed with some additional simulations, and would strongly encourage the authors to share their simulation and evaluation code (see below). I also have some minor comments on the text and presentation.

High level comments

- Although I'm not certain without being able to look at the code (see next point) I'm concerned that the evaluations here are being done on quite simplistic simulations, which may not reflect the properties of the data used in reality. Since Ancient DNA data is very noisy, evaluations based on trees built from perfectly clean simulated data may not be fully indicative of real world performance. I would like to see some additional analysis like Fig 1b, c, d where some errors are thrown on the data. Although methods for generating such noise realistically is an open problem, a reasonable first step could be to use the empirical model of Albers & McVean 2020, to at least show that the results are robust under *some* forms of error, if not the forms of error that we might expect from ancient DNA. Some discussion on these points in the methods would also be welcomed.

Response:

Thank you for this suggestion, we agree that it is important to demonstrate robustness to realistic data quality in these simulations. We have now added several simulations to evaluate this (described below).

In general, we believe that these ancient imputed genomes are of relatively good data quality. This is because we restricted our analysis to genomes of $>0.5x$ sequencing coverage, which has been demonstrated to yield good imputation accuracy, and excluded transition SNPs which are subject to cytosine-deamination derived degradation errors. Please refer to e.g. da Mota et al. 2023 (<https://www.nature.com/articles/s41467-023-39202-0/figures/2>) for an overall evaluation of imputation accuracy for ancient genomes.

To emulate the data quality we expect in imputed ancient genomes, we implemented a simple error model where every haploid genotype at any segregating site can switch with a certain error probability. We can theoretically compute the predicted squared correlation coefficient (r^2) between the true simulated genotypes and the genotypes including error, stratified by minor allele frequency, to generate a plot similar to those used for evaluating imputation accuracy using downsampled high-coverage samples (<https://www.nature.com/articles/s41467-023-39202-0/figures/2>).

As imputation accuracy varies by individual in real settings, we randomly sampled the error probability for each individual uniformly between $1e-4$ and $1e-3$. This yields r^2 curves that are comparable to those observed in real data. We additionally simulated error probabilities between $1e-3$ and $1e-2$, corresponding to very high error rates.

In our new simulations, we combine two haploid sequences to construct a diploid individual. We then computationally rephased these diploid individuals without a reference panel. This approach is expected to result in suboptimal phasing, and we therefore believe that this is a good setup to further test robustness to phase switch errors.

For the above simulations, we observe no or very limited bias in *Twigstats* when estimating admixture proportions, while rare variant ascertainment and coalescence painting approaches were significantly biased. These results are now presented in Extended Data Fig 2.

It is notable that the high-error rate simulations reveal a new direction of bias for the rare-variant cutoff approach, where it significantly underestimates the admixture proportion under the presence of high-error rates.

We have updated the main text with the following:

"We further confirm with simulations that genealogy-based f -statistic estimates are robust to sequencing and phase-switch errors of expected magnitude (Extended Data Figure 3d). In fact, while sequence errors can impact in SNP-based population genetic approaches substantially, errors can be 'corrected' in genealogies as they take all variants in a region into account³². "

In conclusion, we thank the reviewer for suggesting this check of the effect of error rates, which demonstrates further advantages of the *Twigstats* approach.

- I think the code used to run the simulations and evaluations here should be made available. Ideally all code and scripts used to generate the Relate trees from public data as well should be made available, both for scrutiny as well as to provide a resource to the community.

Response:

We have uploaded all scripts to a github repository including documentation on how to run these scripts. This includes scripts to reproduce our simulations, as well as the scripts we used to run Relate on imputed ancient genomes, and downstream analyses, such as computation of f_2 statistics, and qpAdm. We now reference this in the "Code availability" section of our manuscript:

"All scripts to reproduce simulations, as well as to run Relate on imputed ancient genomes, and downstream analyses including computation of f-statistics, and running qpAdm models are available through github https://github.com/leospeidel/twigstats_paper."

In addition, we have expanded our existing documentation further, in particular adding details to run a small real data example analysis: <https://leospeidel.com/twigstats/articles/real-data-example.html>

Minor comments

- L39. "However, tracing history using ancestry has remained challenging,..". It's not clear to me what "ancestry" means here, and I'm not sure what the following sentence really refers to. Perhaps a sentence describing what you mean by "ancestry" immediately after the first "ancient genome sequencing" sentence would help clarify this vital point.

Response:

Thank you for this comment, we agree that it is important to clarify what we mean by ancestry here and throughout this study. First, we have clarified that we mean "genetic" ancestry in the highlighted sentence. Second, we are now elaborating on what we mean specifically by genetic ancestry and cite a recent paper discussing these concepts (<https://arxiv.org/abs/2207.11595>):

"Genealogical trees can be thought of as containing essentially full, time-resolved information about genetic ancestry, including information typically captured by recent haplotype sharing or identity-by-descent. Genetic ancestry here refers to the full collection of genetic ancestors of individuals⁴¹, and genealogical trees reveal how and when these are shared across individuals."

- L70. The applications of ARGs is reasonably comprehensive here, I think the "among others" is unnecessary.

Response:

Thank you, we have removed this.

--L71. It would be good to cite the recent ARG reviews here somewhere as starting points for readers new to the literature. Perhaps at the end of the "Genealogical trees ..." sentence.

- The era of the ARG: An introduction to ancestral recombination graphs and their significance in empirical evolutionary genomics, PLOS Genetics 2024

- The Promise of Inferring the Past Using the Ancestral Recombination Graph, GBE 2024

Response:

Thank you for suggesting these references, we agree that these are a valuable resource and we are now referencing these:

“One class of methods that use haplotype information is full genealogical tree inference^{28,29}”

- L156-163. Does this need some citations? This is entirely out of my expertise, so I don't know whether historians would agree with these statements. Some references would be reassuring.

Response:

Thank you, we have rephrased this paragraph and have now added two references:

Heather, P. *Empires and Barbarians: Migration, Development and the Birth of Europe*. (Pan Macmillan, 2010).

Halsall, G. *Barbarian Migrations and the Roman West, 376-568*. (Cambridge University Press, 2007).

"In the first half of the 1st Millennium CE, Roman historians such as Tacitus and Ammianus Marcellinus described the geographic distribution and movements of groups beyond the imperial frontier, and suggest a potential role for them in the political dismantling of the western Roman Empire⁵². However, the exact nature and scale of these historically attested demographic phenomena—and their genetic impact—have been questioned⁵³, and have been difficult to test with genetic approaches due to the close relations shared between many groups that were ostensibly involved. Less is understood at further distances from the Roman frontier due to lack of historical accounts. The improved statistical power of time-restricted ancestry in *Twigstats* thus offers an opportunity to revisit these questions. "

- Fig 1. It took me a while to figure out that the colours in (c) are defined by the legend at the bottom. I think what confused me was the "dots" in the middle of the colour lines in the legend. Because there's no dots in panel (c), it wasn't obvious to me that they are the same. I think you could add a legend to fig C without losing too much? Another approach might be to use a dotted line in D for the theoretical prediction rather than a blue line, and then you wouldn't need a legend for D. It might then be more obvious that there's only one global colour scheme.

Response:

Thank you for this suggestion, we agree that this would improve clarity and have changed the theoretical prediction to a black and dashed line in panel d).

- Fig 2a. Caption. "between individuals calculated as conventional directly on genotype calls" reads awkwardly, consider rephrasing.

Response:

Thank you for this suggestion. As part of our revisions, we have now moved Figure 2 to the Extended Data Figures. Fig 2a is now Extended Data Figure 4b and we have rephrased the caption as follows:

“Fine-scale genetic structure in Neolithic Europe quantified using an MDS calculated on a symmetric matrix that contains all pairwise outgroup f_3 statistics (outgroup: YRI) between individuals. These are either calculated directly on genotypes or calculated using *Twigstats* on *Relate* genealogies with a cutoff of 1000 generations. Individuals were selected by filtering based on Steppe and WHG ancestry (**Methods**).”

- Fig 2b. It's not clear what the SGDP panel is here. Explain. Generally the caption in this figure needs more clarity I think.

Response:

We have now added the following sentence to the caption to clarify our data sources:

“We compute these on genetic variation data from the Simon’s Genome Diversity Project (SGDP) ⁷⁹ and use the high-coverage Altai and Vindija Neanderthals ^{80,81}. “

- Fig 3. "UK and Ireland" should probably be "Britain and Ireland".

Response:

We have updated this label, thank you for pointing out.

- L564. I would like to see the code used to run these simulations and the various evaluations. While the code for *Twigstats* is mentioned in the Data availability section, the simulation and evaluation code for generating the figures in this paper is not.

Response:

We have uploaded all simulation scripts to a github repository https://github.com/leospeidel/twigstats_paper and we are now referencing this repository in this Methods section.

Referee #1 (Remarks on code availability):

I have (lightly) reviewed the code for the *twigstats* method itself, and it seems to be high-quality package with good documentation.

However, I would very much like to see the code used in the *evaluation* of the method.

Response:

Thank you very much for reviewing our code. We have expanded our documentation further now including several examples on how to run Twigstats, which we hope will be of use to the community.

Referee #2 (Remarks to the Author):

In this paper, Speidel and colleagues develop a new approach to examining patterns of admixture between populations, called Twigstats. Essentially the method combines two existing approaches for modeling population genetic data (coalescent tree reconstruction and f-statistics) to provide more powerful insights, particularly with regard to differentiating subtle population structure in ancient populations. They then demonstrate the robustness of this approach by testing it on various questions in human population genetic history, either using simulated data or replicating (and in many cases enhancing) analysis in well-established demographic scenarios. Finally they apply their method to existing, previously published Bronze Age, Roman and Medieval whole genomes to gain new insights into putative population movements described in the historical record, with a particular focus on the movement of Germanic-speaking barbarians from Northern Europe into the former Roman Empire and the impact on and by medieval Scandinavian populations such as the Vikings.

First of all I will say there is a lot to like about this paper from a purely population genetics methods perspective. The idea of using the underlying genealogies (as estimated by the primary author's own software, Relate) to estimate f-statistics, rather than the messier allele frequencies is really very clever. Further, the focus on using more shallow genealogies to extract younger coalescent events that are more relevant to discriminating more recently diverged populations is also a very powerful insight. The theory presented in the supplement appears sound to me, and the performance on simulated and real data prove this method works well and provides additional insights beyond that was possible using observed allele frequency-based f-statistics alone. I do agree with authors in that this paper is a good argument for why the human paleogenetics community will need to transition from capture-based genomes like 1240k to full whole genomes if we are maximize the insights gained from ancient DNA.

We fully agree with the reviewers sentiments, and thank them for their comments.

However, I do not believe the study rises to the level expected of a Nature paper for the following reasons.

1) The method itself is an incremental improvement on already existing methods (though done in a very neat way), providing a refinement in terms of insights rather than transforming how we can look at genetic data from a pop gen perspective (for example Li and Durbin 2011 with PSMC).

Response:

Our method provides a new conceptual approach for studying ancestry in ancient DNA, that we show with real-world data can provide an order-of-magnitude improvement in resolution of ancestry models with ancient genomes. We thus see this as the largest advance in how to study ancient DNA since the introduction of 'qpAdm' models in Haak et al. 2015, almost a decade ago.

Since the introduction of f-statistics-based modelling in Reich et al. 2009, Nature, and then qpAdm-based approaches in 2014-2015 (Lazaridis et al. 2014, Haak et al. 2015), no

generalised method has been able to broadly improve the resolution of unbiased ancestry estimates. In practice, ancient DNA projects are increasingly hindered by low statistical power to study the ancestry questions they are interested in. Our aim was to introduce a potentially transformative new technique that addresses these challenges.

IBD- and chromosome-painting-based methods are increasingly used for ancient DNA, but were previously available for modern data (Lawson et al. 2012; Palamara et al. 2012), and as we show in our current manuscript are unable to provide unbiased ancestry estimates (see more in response below). Our *Twigstats* approach goes beyond these improvements in being a new approach that has not been applied either to ancient DNA or modern genomes, and is able to provide unbiased ancestry models with high resolution.

We have included in the text that the bias observed for chromosome painting would also be expected to affect IBD-based approaches, since they are conceptually similar.

“We also demonstrate that a widely-used 'chromosome painting' approach, and any conceptually similar modelling based on identity-by-descent (IBD), that finds the nearest neighbours between chromosomal segments in a sample and model groups using a non-negative least squares of genome-wide painting profiles² is also prone to bias, when source groups have undergone strong drift since the admixture event (Figure 1b, Extended Data Figure 3).”

We believe that our method is transformative for empirical population genetic studies, as it opens up the potential for a new generation of genetic history studies with much-improved resolution. We demonstrate this directly in our application to Northern European history. Indeed, the approach is also fully applicable to non-human organisms where whole-genomes or imputation is available.

2) The application of this method is to existing published data. The results from this are a series of interesting additional insights to existing studies, but derived in a very post-hoc way. I can imagine that the application of this method will be extremely powerful when combined with a robust sampling of ancient whole genomes from multiple time points to answer a specific question (i.e. with a robust study design). But at the moment the conclusions are very coarse with lots of caveats that derive from the sampling time and geography (which of course the authors have no real control over).

Response:

Our empirical analysis of European genomes focusing on the early medieval period resulted in several robust advances in our understanding of genetic history in the period. For example, we find a previously unknown expansion into Scandinavia for which sufficiently comprehensive sampling is already available. In the manuscript, we have indicated throughout the strength of evidence in the light of currently available samples. In the original manuscript we used all relevant data available, and in the revised version we have added ~405 recently published new genomes in order to have more comprehensive sampling.

There is not any particular major take home message from the results of the analysis that is particularly striking or specific to me, but rather a series of intriguing vignettes that could be fruit for future in-depth study. The overall conclusion of the abstract that “our results are consistent with substantial mobility in Europe in the early historical period” is not a major leap in our understanding of the post-Roman Medieval period beyond what is known already from history, archaeology, or even recent previous paleogenetic work.

Response:

We appreciate this comment and have worked to improve presentation in our revised manuscript to highlight several important general and novel conclusions about Germanic expansions through the lens of genetic history. We have moved Figure 2 into the Extended Data Figures and now devote three main text figures to our findings. Specifically

1) We trace Germanic expansions using ancestry signals in the period 1-500 CE, which was previously suggested by historical accounts from Roman authors and archaeological material culture, but not demonstrated on a continental scale with ancient DNA. Our modelling demonstrates expansions eastwards, with individuals in Poland, Slovakia and the Balkans having clear evidence of Scandinavian-Iron-Age-related ancestry. We see it extending southwestwards, in the Baiuvarii in southern Germany, the Langobards in Hungary and Northern Italy, and the Saxons of the Netherlands and England.

In the revised manuscript, we now provide a new finding, demonstrating that we can distinguish ancestry signatures likely related to the expansion of now-extinct Eastern Germanic languages in the direction towards the Black Sea, and separately Western Germanic languages (Figure 2d and discussed in lines 294 - 305)

“Across Europe, we see regional differences in the southeastern and southwestern expansions of Scandinavian-related ancestries. Early medieval groups from present-day Poland and Slovakia carry specific ancestry from one of the Scandinavian EIA groups—the one with individuals primarily from the northern parts of Scandinavia in the Early Iron Age— (Figure 2d), with no evidence of ancestry related to the other primary group in

more southern Scandinavia. In contrast, in southern and western Europe, Scandinavian-related ancestry either derives from EIA southern Scandinavia - as in the cases of the probable *Baiuvarii* in Germany, Lombard-associated burials in Italy, and early medieval burials in southern Britain - or cannot be resolved to a specific region in Scandinavia. If these expansions are indeed linked to language, this pattern is remarkably concordant with the main "branches" of Germanic languages, with the now-extinct 'eastern Germanic' attested by Goths in Ukraine on the one hand, and 'western Germanic' languages such as Old English and Old High German recorded in the early medieval period."

2) Ancestry influx into Scandinavia by 500 CE of ancestry related to continental Europe. This has previously not been observed with ancient DNA. Instead, it reveals a major new phenomenon in genetic history. We integrate this with stable isotope data and historical records to show that it was not a phenomenon of transient mobility.

To reflect this, we have changed the title of the paper to be "**High-resolution genomic ancestry reveals migration waves in medieval Europe**"

In many ways this paper is equivalent of other recent papers (including by this primary author) where an exciting new method has been developed for aDNA and then applied to existing data (Ringbauer et al. 2021, Ringbauer et al. 2023, Speidel et al. MBE). I certainly do not think this manuscript rises above any of these, and I think a journal like Nature Genetics would be a good home for this paper.

Response:

Our new approach effectively unlocks new information from ancient DNA data, and we believe that this is similar in spirit to how sequencing unlocks new information from archaeological remains. We are keen to demonstrate the utility of this new information, and have worked to improve presentation of our findings through revising our main text figures in our updated manuscript.

We are encouraged by the fact that IBD inference (using methods introduced by Ringbauer and colleagues) has had an increasing impact and is emerging as a standard analysis step in ancient DNA studies to identify direct relatives in the past few generations, highlighting the potential impact of new methods. Conceptually, IBD inference in aDNA has primarily been enabled by the ability to impute genomes, while the underlying algorithms given such data have conceptually remained similar (e.g. Gusev et al. 2009; Palamara et al. 2012). We believe that our method is a conceptual advance and is not just limited to applications in ancient DNA. We are now able to access previously unattainable time-stratified information about ancestry.

Importantly, we believe that our approach addresses different questions to IBD inference. IBD inference is well suited for detecting close genetic relatives within the past few generations, but is less well suited to quantify more distant relationships that do not result in medium to long IBD tracts, as highlighted by authors of the leading IBD inference techniques referenced by the reviewer (<https://theconversation.com/dna-says-youre-related-to-a-viking-a-medieval-german->

jew-or-a-1700s-enslaved-african-what-a-genetic-match-really-means-222833). In comparison, we believe that ancestry modelling on whole genome genealogies can robustly capture genome-wide ancestry, which has not been possible previously.

3) I also am not sure if the general approach of this method (that population in the future are mixtures of distinct ancestral populations) is the best model of the data from the time periods being examined. The expectations of f-statistic-based approaches such as f4 ratio tests, qpwave and qpAdm are based on a fairly explicit model of population divergence with a pulse-like admixture. While they can be robust to some violations of this model, probably one of the most extreme violations would be case where there is continuous geneflow between populations, for example in an isolation-by-distance or stepping-stone model. Indeed Harney et al. (2011) examined this and concluded that they “caution against ... analyzing population histories involving extended periods of gene flow”. The overall patterns of population structure in Europe from the Bronze-Age onwards (Antonio et al. 2024, which happens to be where much of the data used in this study are from) are very much in line with a model of isolation-by-distance and largely seem to extend to the kind of patterns we see in modern Europe (Novembre et al. 2008). Yet by nature of how the f-statistic approach is applied, the authors of this study must define discrete “ancestral populations” and that subsequent populations/individuals are then modeled by these discrete groups. While of course I recognize this can be a useful way of modeling the data, I argue that this is likely so far from the reality of actual spatial structure of the populations, that it provides a potentially false view of how these populations are derived at multiple levels. And often the general patterns observed by the authors very much seem like they are probably just isolation-by-distance (for example more Austria/France/German ancestry in southern Scandinavia), but the framework is forcing us to think of them as mixing of distinct sources. The authors have some interesting simulations looking at the effectiveness of their approach in situations with geneflow rather than pulses of admixture, but not sufficiently to deal with my concerns above. For example extended data figure 4a shows the impressive fine-scale resolution that can be obtained with their approach even for populations connected by high migration. But this only looks at one particular aspect derived from f-statistics, which is via an outgroup f3 statistic, so one shared drift path between two populations (that will represent a mix of geneflow and divergence, rather than just pure divergence). This does not necessarily tell us how the method will perform in a qpAdm or qpwave context where now it is about the extent of the sharing of a complex network of drift paths under extended geneflow. Similarly, the method seems to do an impressive job at distinguishing deep African population structure with lots of past geneflow from a pulse of neanderthal admixture, but I do not think this is that relevant to the question of recent extended geneflow (and I assume much of the power comes from cutting off the trees to limit the impact of that deep structure, which will tend to extend coalescence times).

Response:

We believe that the best way to think of ancestry in practical terms is through questions such as: What proportion of the genetic ancestors of this 500 CE individual lived in Scandinavia in 100 CE? Such a question is in principle independent of demographic models of isolation-by-distance or punctual mixture. F4-ratios and qpAdm in fact do not assume punctual admixture and instead quantify the cumulative ancestry tracing to different groups; using Twigstats, we are able to

time-stratify this to exclude any potential deeper migration events that could confound more recent events. The qpAdm framework allows one to test whether such models fit, quantify the space of plausible models, and infer proportions.

We agree that testing for patterns resulting from isolation by distance and understanding the behaviour of our modelling in such settings is important. We have therefore added a new simulation, described below, and find that we are able to robustly identify continuous gene flow scenarios and are still able to accurately model groups as mixtures of their most proximal neighbours in such settings.

In our empirical analysis, we detect clear divergence from constant isolation by distance patterns and identify waves of migration between groups across time. This includes between geographically distant groups as in the case of Northern European ancestry arriving in southeastern Europe in the first half of the first millennium, as well as between geographically more proximate groups, such as with the arrival of central European ancestry in Denmark.

Isolation by distance simulation:

We evaluated the robustness of our approach in a simulation with extensive migration and resembling isolation by distance. We adapted the simulation of Harney et al (<https://academic.oup.com/view-large/figure/247988639/iyaa045f11.tif>), where six populations are organised on a 1-d grid and exchange migrants. We simulated under migration rates of 0.001 and 0.005, corresponding to average F_{st} values of 0.01 and 0.002, respectively. Empirically, we observe average F_{st} values in this range in early medieval Europe (Ext Data Figure 1c). We then used qpAdm to model population 2 as a mixture of two other populations. As in our real data application, we employed a rotational qpAdm scheme, such that unused populations were used as outgroups.

An important assumption in qpAdm is that no substantial gene flow occurs between reference and source populations post admixture and we expect *qpAdm* to reject models that violate this assumption. Averaging over 50 replicates for each migration rate, we find that *twigstats qpAdm* now strongly rejects all models ($p < 1e-100$), which is consistent with the fact that this *qpAdm* assumption is violated. Using conventional *qpAdm*, we are only able to do so when $F_{st} = 0.01$ but lack power when $F_{st} = 0.002$. This suggests that we are well powered to reject continual isolation by distance scenarios at F_{st} values resembling early Medieval Europe only using *Twigstats*.

In addition, we find that in close to 75% of all simulations, the 'best' model (i.e., the source group combination with the highest p-value) is the indeed the model where the two adjacent populations '1' and '3' are chosen as sources. This illustrates that even in these scenarios, we are able to accurately recapitulate migration routes and reject less proximal sources.

We have included this new simulation in Extended Data Figure 3f.

4) The authors define the source groups by applying qpwave on Bronze Age and Roman samples to find “homogenous groups” via UPGMA clustering. There are two potential issues or concerns I have with this. From my point above, the suitability for qpwave to do this on populations exhibiting extended gene flow is at best, unclear. However, this was also the approach followed by Antonio et al. so I understand why this was done, and could potentially provide at least some decent proxy of ancestral populations. However, Antonio et al. were largely identifying “cluster” to better characterize “outliers” from the same period, so the purpose is not really the same as used in this study, which is to model all subsequent individuals in the post-Roman period. And while I am sure the new Twigstats approach can increase the refinement of these “sources”, any real insight I think is limited by the very sparse sampling available to the authors. This leads to the somewhat weird description of sources that consist of Portugal and Lithuania as distinct sources, but also three sets of Austria/France and Germany samples being distinct sources, and five sets of sources from “Italy” that as far as I can tell all derive from a strip of central Italy in and around Rome. This kind of lumpy and uneven “source population” attribution is almost certainly driven by the very limited and uneven sampling. I highly doubt that such a distribution of groups in time and space would emerge with a more comprehensive sampling of Europe. While an argument could be made that the approach of the authors is a statistically “objective” way of choosing sources, the underlying choice of samples was not, which obviously contaminates for the former. There is also the question of what these sources even represent. If three samples from the same “population” (R11557, R11552, R11560, all France_Sarrebouurg_LateAntiquity.SG) are put into two source “populations” by this approach, and one of these source populations also includes an individual from Austria (R10659, Austria_Klosterneuburg_Roman.SG) a 1000km away, what does it mean if a subsequent person has ancestry from both these sources? There appears to be much more resolution in the Scandinavian analysis due to the better, denser, sampling, and some of the inferences can be quite nuanced (hence I assume the emphasis on this section), showing the importance of sampling and study design to make this approach valuable.

Response:

Thank you for raising two potential issues which we will address below:

- 1) Suitability of qpwave for clustering individuals
- 2) Interpretation of resulting source groups

We agree that a suitable choice of source groups is indeed a common challenge in ancestry modelling. We were particularly worried about bias in our results when hand-curating these source groups. Our current strategy is to identify groups based on cladality testing using qpwave, however we have now worked to better characterise the identified source groups in a new Figure panel (Figure 2a).

Figure 2a illustrates the genetic relationship between our source groups and their geographic spread to aid interpretability. We find that clusters that were identified using a qpwave-based cladality test also fall in close proximity in an non-parametric MDS, supporting that we are identifying real genetic affinities among individuals. Additionally, clusters overlap well with geopolitical or archaeological descriptive data. This is despite the fact that our approach is agnostic to this information when clustering samples.

Particularly in central Europe, we agree with the reviewer that the full extent of ancestry variation remains to be explored. Our manuscript is therefore careful to speak about this type of central European ancestry jointly.

We note however that currently all central European clusters fall in relatively close proximity in our MDS, with adjacent regions of Roman Portugal and Britain clustering alongside these groups. This indicates that the bulk of ancestry in Iron and Roman Age central Europe is likely captured by the samples available. We further note that we employ a rotational qpAdm scheme where unused source groups are assigned as reference groups. Therefore, we are encouraged

that we see fitting models with these central European groups chosen as sources over other adjacent groups.

We have included the following sentence to elaborate on the ancestry variation in central Europe:

“Sparse sampling limits our understanding of the full extent of regional ancestry variation in central Europe and some other regions, but the continental ancestries differentiated in the MDS suggests that major ancestry variation across Europe in this period is already well captured.”

5) There are some interesting insights about certain individuals and their ancestry, but usually we are left with very coarse descriptions of the actual processes that might have led to these patterns and their dynamics, usually not much beyond what was described the original study the data derived from, and certainly not with any precision to make conclusive statements about historical events. This is not a fault of the authors, and in some ways I am glad they are (mostly) cautious, but is the limitation of the data and sampling. While I see Peter Heather is one of the co-authors, I think most medieval historians would be somewhat underwhelmed by the insights that derive from these ancestry inferences.

Response:

Thank you for this comment. We are keen to provide interpretable results in light of historical events and our paper is therefore the product of close collaborations involving leading historians and archaeologists.

We are pleased that Referee 3, whose expertise is indeed listed as 'historical archaeology', is not underwhelmed but write in their report that "*This is an important paper and one that is publishable in Nature*", with "*considerable implications for 1st millennium Europe*" that reports "*an important but previously unrecognised migration into pre-Viking Age Scandinavia*".

We have now worked to improve our characterisation of the genetic signatures we identify and we share the sentiment of the reviewer that it is important to be cautious but also important to provide plausible scenarios and context that can explain our inferred ancestry patterns.

We list these as responses to your comments below.

For example, I highlight some statements below that while sometimes intriguing, are very superficial from a historical point of view:

“The Wielbark archaeological complex has been associated with the Goths, a Germanic-speaking group, but this attribution has remained unclear. Our modelling supports the idea that some early Germanic-speaking groups expanded into the area between the Oder and Vistula rivers”

So does this resolve that they were Goths? Probably not.

Response:

While we cannot answer the link between the Wielbarks and Goths given current data, we believe that there is merit in discussing this link. We have rephrased this paragraph as follows:

“The Wielbark archaeological complex has been linked to the later Chernyakhov culture and early Goths, an historical Germanic group that flourished in the 2nd - 5th centuries CE⁵⁶. Our modelling supports the idea that some likely Germanic-speaking groups from Scandinavia expanded south across the Baltic into the area between the Oder and Vistula rivers in the early centuries CE, although whether these expansions can be linked specifically with historical Goths is still debatable. In addition, since a considerable proportion of Wielbark burials during this period were cremations, the possible presence of individuals with other ancestries can not be strictly rejected if they were exclusively cremated (and therefore invisible in the aDNA record). ”

“The presence of a southern European-like ancestry component (represented by Roman Italy), which is not present in the Lithuanian Roman Iron Age, could be consistent with models of admixture taken place further south and arriving in Poland through north-westerly Slavic expansions.”

What models of admixture are these exactly, are they associated with specific historical events?

Response:

We agree that this particular hypothesis about the precise geographic origin of Slavic expansions was speculative, and have removed it from the manuscript.

We have rephrased this section as follows:

“A present-day individual from Poland is similar in ancestry to the medieval individuals in lacking detectable Scandinavian-related ancestry (Figure 3a). Some of the ancestry detected in individuals from later medieval Poland may have persisted through the late 1st Millennium CE in the cremating portion of the population, but regardless this points to large-scale ancestry transformation in medieval Poland. Future data would shed light on to what extent this reflects the influence of Slavic-speaking groups in the region.”

“Later Migration period individuals only have partial Scandinavian-related ancestry, suggesting later broader ancestry shifts in the region.”

What exactly are these broader ancestry shifts? This almost is meaningless, other than saying, their ancestry seems to have change.

Response:

The genetic legacy of northern European ancestry arriving in these regions during the migration period is debated and we feel that it is of value to describe the genetic ancestry of people in this time period. We have rephrased this sentence to clarify:

“Later early medieval individuals from Slovakia have partial Scandinavian-related ancestry, providing evidence for the integration between expanding and local groups.”

“we observe Scandinavian-related ancestry components in several Longobard burials (Longobard_EMED(I)) dating to the middle of the 5th century CE (Figure 3b)...The Longobards were dominated by Germanic-speakers, and our result is consistent with attestations that they originated in the area of present-day Northern Germany or Denmark. In contrast, several other Longobard-associated individuals from the region (Longobard_EMED(II)) show ancestry more closely related to continental Europe, putatively representing local ancestry prior to Langobardic influence. This is consistent with the notion that the Longobards of the fifth and sixth centuries headed a confederation of several different groups.”

This statement is actually quite troubling. It seems that the authors have decided that some burials are Longobard, because they have Scandinavian ancestry, but those that have Central European ancestry are simply Lombard-associated and are probably local. This to me is bordering on genetic essentialism and certainly somewhat circular.

Response:

Thank you very much for pointing this out, this was an unintended accident in our formulation and agree that this should be rephrased:

“Nearby, in present-day Hungary, we observe Scandinavian-related ancestry components in several burials dating to the middle of the 6th century CE associated with Longobards (Longobard_earlyMED(I))¹⁰ (Figure 2c). This is consistent with the original study¹⁰, which reported affinity to present-day groups from northwestern Europe (GBR, CEU, and FIN in the 1000 Genomes Project)¹⁰, but which we can resolve with higher-resolution using earlier genomes. Several other individuals from these Longobard burials (Longobard_earlyMED(II)) show no detectable ancestry from northern Europe and instead are more closely related to Iron Age groups in continental central Europe, putatively representing descendants of local people buried in a Longobard style. Our results are consistent with attestations that the Longobards originated in the areas of present-day Northern Germany or Denmark, but that by 6th Century CE they incorporated multiple different cultural identities, and mixed ancestries.”

“In southern Germany, the genetic ancestry of individuals from early medieval Bavaria likely associated with the Germanic-speaking Baiuvarii cannot be modelled as deriving ancestry solely from earlier groups in Iron Age Germany. Our current best model indicates a mixture with ancestry derived from EIA Peninsular Scandinavia, suggesting a regional ancestry shift and expansion of Scandinavian-related ancestry (Figure 3b)”

Again, a regional shift with no real explanation about the processes underlying it or the timing.

Response:

The origins of the Baiuvarii people, who appear in the region around the 5th century, remains unclear from historical documents, although links to Longobards and other Germanic speaking groups exist. We have added this into the paragraph”

"In southern Germany, the genetic ancestry of individuals from early medieval Bavaria likely associated with the historical Germanic-speaking *Baiuvarii*⁵⁹ cannot be modelled as deriving ancestry solely from earlier groups in Iron Age central Germany ($p < 1e-36$). The *Baiuvarii* likely appeared in the region in the 5th century CE⁵⁹, but their origins remain unresolved."

References

- Li and Durbin. Inference of human population history from individual whole-genome sequences. *Nature* 2011
- Ringbauer et al. Parental relatedness through time revealed by runs of homozygosity in ancient DNA. *Nature Communications* 2021
- Ringbauer et al. Accurate detection of identity-by-descent segments in human ancient DNA. *Nature Genetics* 2023
- Speidel et al. A method for genome-wide genealogy estimation for thousands of samples. *Nature Genetics* 2023
- Speidel et al. Inferring population histories for ancient genomes using genome-wide genealogies. *MBE* 2021
- Harney et al. Assessing the performance of qpAdm: a statistical tool for studying population admixture. *Genetics* 2021
- Novembre et al. Genes Mirror geography within Europe. *Nature* 2008
- Antonio et al. Stable population structure in Europe since the Iron Age, despite high mobility. *ELife* 2024

Referee #2 (Remarks on code availability):

The code is available. But I do not have sufficient time to test it as this would require substantial analysis of genomic-scale data. However, the maths is sound as far as I can tell. There is extensive documentation.

Response:

Thank you and we really appreciate the time taken to evaluate our work.

Referee #3 (Remarks to the Author):

This is an important paper and one that is publishable in Nature. The development of Twigstats is significant in moving away from using modern populations in aDNA research and will have a major impact on the field.

We appreciate the positive evaluation of our work by reviewer #3 and are grateful for the comments which we address below.

The Data also points to an important but previously unrecognised migration into pre-Viking Age Scandinavia. There are however some points that the authors need to attend to. These are outlined below, and many relate to terminology which is inconstantly presented, and confusing it often mixes regions, periods and genetic groups and need to be made clearer. There is a tendency towards conflating individual mobility vs migration, but this may be the result of loading of interpretation onto individual burials. Terms like Tribe, Germanic speakers Slavic Speakers should be avoided (tribes are specify units of social organisation and genetics cannot tell you how an individual spoke) and genetic affinity and chronology become conflated and or are difficult to separate in some places. Perhaps adopt ancestry as a term as other papers have to avoid these mix-ups.

Response:

Thank you for these suggestions. We have amended the paper throughout to avoid assuming tribal membership or languages spoken.

This is an enormously complex topic and paper, with considerable implications for 1st millennium Europe and is likely to be controversial, it is therefore important to be clearer. The early sections on Roman and iron age Scandinavia are the most problematic and too many words are spent on straw men for example, Neanderthals, the Roman Gladiator, and Female settlement vs Male mobility interpretations tend to imply a bias in interpretation that is not favourable to the hard work of the authors. I'd recommends reducing and simplifying these sections so that there is more space for the details that are needed to convey the importance of the Ancestry Changes in late Iron Age Scandinavia onwards doing so will not remove form the importance of Twigstats.

Response:

Thank you for these comments. We have reduced and simplified the section on other empirical examples such as Neanderthals and Neolithic Europe, as well as moved Figure 2 that displayed these results to instead be a Extended Data Figure 3. This allows more focus and space for the importance of ancestry changes in Iron Age and early medieval Europe, as the reviewer suggests.

These sections convey the implication of then DNA data well, but the terminology used, the oversimplification of and absence of archaeological details weakens them. Because of the

tendency to refer to people as genetic and not cultural groups the paper does not convey as well as it could the hugely complex melting pot of genetics and cultures that are present in Northern Europe that than it could and given that this is what it identifies this is a challenge. In this paper the Twigstats approach is able to distinguish between Scandinavian IA and Austrian-French-German (AFG) Iron Age ancestry. The published Isotopic Data (Leggatt) and the early medieval English population in Gretzinger et al (their Ref 37) indicates that the CNE genetic groups identified were a mixture of these two it would be of considerable benefit then to perform a Twigstats analysis on that CNE population to see if there is a difference between those in SE England (which we might expect to be AFG) and North eastern England (which we might expect to be Scandi IA) particularly as these genetic groups seem to be together in England at the same point as there is conflict between them in southern Scandinavia (as presented here).

Response:

Indeed, we applied the Twigstats approach to different sets of individuals identified by Gretzinger et al., and see a difference in inferred ancestry for those that were identified as "high-CNE" compared to those identified as "low-CNE" or "mid-CNE".

At present, our models do not indicate an "AFG" ancestry component in these individuals, however it is statistically challenging to detect a minority component tracing to AFG in this case. Increased sample sizes may allow us to study the ancestry make-up of early medieval England in more detail in future.

As an aside given the genetic tree analogy and the temporal and geographic focus of the paper I'm surprised they did not call Twigstats Yggstats after Yggdrasil the early medieval Scandinavian mythological tree of life.

Response:

We thank the reviewer for this imaginative idea, perhaps in future iterations of the approach! For the moment, we will keep the original name, also since the method will be applicable to the genetic history of all regions of the world.

Some Ref to follow up on in relation of the points raised in this review:

Leggett, S., Hakenbeck, S., & O'Connell, T. (2022, June 9). Large-scale Isotopic Data Reveal Gendered Migration into early medieval England c AD 400-1100.

<https://doi.org/10.31219/osf.io/jzfv6>

Hamerow H, Leggett S, Tinguely C, Le Roux P. Women of the Conversion Period: a biomolecular investigation of mobility in early medieval England. *Antiquity*. 2024;98(398):486-501. doi:10.15184/aqy.2023.203

Hakenbeck S.E.(2009). 'Hunnish' modified skulls: Physical appearance, identity and the transformative nature of migrations. H. Williams and D. Sayer (eds.), *Mortuary Practices and Social Identities in the Middle Ages. Essays in Honour of Heinrich Härke*. Exeter: Exeter University Press. 64-80.

Detailed comments

98 theoretical prediction and simulation – be more specific please, what is theoretical prediction?

Response:

Thank you, the theoretical prediction is the dashed line in Figure 1d, which is the expected ideal "cutoff time" given the mathematical model developed in the Supplementary Information. The simulations are the yellow dots in the same figure. We have added a reference to Figure 1d in this sentence.

135-145 – Iron Age and Northern Europe - that is fine from a methodological perspective but it would be good to have the implications of this spelled out more clearly.

Response:

Thank you, we have removed this reference from that part of the manuscript, in order to be able to discuss Iron Age Northern Europe in full in the later section of the manuscript.

Fig 3 – the label used here are opaque – England Saxon is not a term used in this period, and what is England Gladiator? Why use an ethnic term that has no historical basis in one place and an interpretive term based on superstition in another – is the York sample a gladiator or a Roman soldier?

Response:

Thank you, we have revised all labels in order to carry less assumptions, and reflect modern scholarship in archaeology and history better.

In particular, we have relabelled "England Saxon" as "England_EMED" and "England Gladiator" as "England.Driffield.Terrace", whose ancestry we now discuss as follows:

"We used this improved resolution to demonstrate that an earlier Roman individual (6DT3) dating to ~second to fourth century CE from the purported gladiator or military cemetery at Driffield Terrace in York (Roman *Eboracum*), England⁶⁰, who was previously identified as an ancestry outlier^{61,62}, specifically carried ~25% EIA Scandinavian Peninsula-related ancestry (Figure 2c)."

156-159 – this is a significant oversimplification of the fall Rome and needs further clarification, and references. I'd also recommend avoiding loaded terms like the 'migration period' provide dates here instead. Which Roman historians have written about this?

Response:

Thank you, we have revised this paragraph to avoid oversimplification, added reference to the Roman authors Tacitus and Ammianus Marcellinus, and removed the reference to the migration period.

“In the first half of the 1st Millennium CE, Roman historians such as Tacitus and Ammianus Marcellinus described the geographic distribution and movements of groups beyond the imperial frontier, and suggest a potential role for them in the political dismantling of the western Roman Empire⁵². However, the exact nature and scale of these historically attested demographic phenomena—and their genetic impact—have been questioned⁵³, and have been difficult to test with genetic approaches due to the close relations shared between many groups that were ostensibly involved. Less is understood at further distances from the Roman frontier due to lack of historical accounts. The improved statistical power of time-restricted ancestry in *Twigstats* thus offers an opportunity to revisit these questions.”

160-16 – if the nature and scale of these things has been questioned then there should be references to this important debate here.

Response:

We have added a reference e.g. to G Halsall 2007 "Barbarian Migrations and the Roman West, 376–568". We note that due to journal guidelines, we are limited to approximately 50 citations in the main text.

165-175 – very important and powerful, this is the key contribution of this paper so I'd recommend flagging this up more prominently in the introduction.

Response:

We thank the reviewer for this positive feedback and suggestions. Based on their feedback we have moved this paragraph to the introduction of the manuscript (l. 48 - 62), with slight modifications, to highlight the issue of studies using modern populations to model past ancestry-which our new method addresses.

“Several recent studies have documented substantial mobility and genetic diversity in these time periods, suggesting stable population structure despite high mobility⁵, and documenting genetic variation in Viking Age Scandinavia^{6–8}, early medieval England^{3,9}, early medieval Hungary^{10,11}, and Iron Age and medieval Poland¹², but largely had to resort to using large-sample-size modern cohorts to study ancestry change through time and space. This is because the differentiation between Iron Age groups in central and northern Europe ($F_{ST}=0.1-0.7\%$; Extended Data Figure 1) is an order of magnitude lower than e.g. the more commonly studied hunter-gatherer, early farmer, and steppe-pastoralist groups that shaped the ancestry of Stone- and Bronze Age Europe^{13–16} ($F_{ST}=5\%-9\%$ ^{13,17}). Modern populations provide ample power to detect differences, but their genetic affinity to ancient individuals may be confounded by later gene flow, *i.e.* after the time of the ancient individual(s)¹⁸. The most principled approach is thus to build ancestry models where

source and 'outgroup/reference' populations are older than, or at least contemporary with, the target genome or group which we are trying to model¹⁸. However, this has been challenging, due to the limited statistical power offered by the thousands-fold lower sample sizes and reduced sequence quality of ancient genomes."

180-181 – great but perhaps this needs to be more transparent in the numbers and reality of these samples, there are not that many samples from Roman Britain presently, for example, how many are used here and are they representative of the Roman population or specific to cosmopolitan area.

Response:

Thank you. We have added sample numbers to region.

"This resulted in a set of model ancestry sources that included Iron Age and Roman Britain (n=11), the Iron Age of central European regions of mostly Germany, Austria, and France (n=10), Roman Portugal (n=4), Roman Italy (n=10), Iron Age Lithuania (n=5), the Early Iron Age Scandinavian Peninsula (Sweden and Norway, n=10), as well as several other more eastern groups dating to the Bronze- and Early Iron Ages (n=25) (Figure 2a, Extended Data Figure 1)."

Additionally, we now show in Figure 2a the geographic distribution of these individuals used as sources in our ancestry modelling.

We have added a caveat about the potentially limited representation of these samples:
"Sparse sampling limits our understanding of the full extent of regional ancestry variation in central Europe and some other regions, but the continental ancestries differentiated in the MDS suggests that major ancestry variation across Europe in this period is already well captured."

203 – again where are the references to this contested association, what do you mean by 'Goths' where these one group by this time, and why associate them with Germanic speaking

this is a supposition imposed by Roman writers and is potently contested by these results so this needs to be developed and presented in a more representative and holistic fashion.

Response:

We have updated the sentence to remove mention of language. Indeed, our results show a strong Scandinavian link in the ancestry of the Wielbark-associated individuals, which to us seems supportive of their association with the Goths. This paragraph now reads:

“The Wielbark archaeological complex has been linked to the later Chernyakhov culture and early Goths, an historical Germanic group that flourished in the 2nd - 5th centuries CE⁵⁶. Our modelling supports the idea that some likely Germanic-speaking groups from Scandinavia expanded south across the Baltic into the area between the Oder and Vistula rivers in the early centuries CE, although whether these expansions can be linked specifically with historical Goths is still debatable. In addition, since a considerable proportion of Wielbark burials during this period were cremations, the possible presence of individuals with other ancestries can not be strictly rejected if they were exclusively cremated (and therefore invisible in the aDNA record).”

219 – represented by Roman Italy how? Roman Italy attracted immigrants from Northern Europe and North Africa as well as Asia so please explain why you can be sure your samples are not unique individuals.

Response:

We removed the mention of Roman Italy here. In these models, individuals from Roman Italy were in practice used as a source. While there is indeed evidence of ancestry variation spanning the Near East and North Africa in the Roman genomes that are available, that wouldn't be expected to change the observation of a more "southerly" component in the Medieval Poland sample.

221 – Slavic expansions needs justifying as a statement to genetics.

Response:

We changed "expansions" for "migration" here to be more general, and updated the sentence to be less speculative about Slavic expansions. This sentence now reads:

“Future data would shed light on to what extent this reflects the influence of Slavic-speaking groups in the region.”

227 – why is this single 50-60 year old female attributed to nearby settlement and not long-distance exogamous marriage that we know to have taken place in the 1st century? It's not clear how the data supports this conclusion, or how these references used do either.

Response:

We took these interpretations from the original publications, but the reviewer is right that they are hard to evaluate. We have removed the statement about the individual being from a nearby settlement. We still think the individual is a relevant example of Scandinavian-like ancestry appearing increasingly in a southeasterly direction in the early 1st Millennium CE, but as the reviewer notes, we can't exclude that the individual is a rarer example of long-distance mobility.

The sentence now reads:

"However, a 1st century burial of a 50-60-year-old female from Zohor is modelled only with Scandinavian-related ancestry, providing evidence of ancestry related to the Scandinavian EIA appearing southwest of the range of the Wielbark archaeological complex^{5,57} (Figure 3b)."

229 – later ancestry shifts – or perhaps migration and integration of Scandinavian and local populations suggesting settlement.

Response:

We agree. We have updated the sentence according to this suggestion to read:

"Later early medieval individuals from Slovakia have partial Scandinavian-related ancestry, providing evidence for the integration between expanding and local groups."

236-241 – or that the Lombards were a migratory group made up of people from different ancestries and not a biologically distinct group but a culturally evolving conglomerate like other overland migrating populations.

Response:

Yes, we completely agree with this view, and in the original manuscript noted this view. We have in the revised version further emphasised this, writing that

"Our results are consistent with attestations that the Longobards originated in the areas of present-day Northern Germany or Denmark, but that by 6th Century CE they incorporated multiple different cultural identities, and mixed ancestries."

248-253 this is a quite a fascinating result – are other sources available?

Response:

We agree with the reviewer that this is fascinating and an example of a clear ancestry change that would have been too subtle to detect with standard methods, but nevertheless clearly had a large demographic impact. The original article presenting the Baiuvarii data by Veeramah et al. 2018 compared them to modern populations alone, and only broadly described their ancestry as "ancestry that closely resembles modern northern and central Europeans". Since that publication, earlier Iron Age genomes from further north in Germany are now available, which together with our higher resolution methods allowed us to reveal this ancestry change.

260-263 – very interesting result, be careful of a gender bias in the authors imprecations. Earlier a single female was evidence of nearby settlement, whereas now two men are soldiers and evidence of non-local mobility. What date are these soldiers, you provide dates elsewhere? Cavalry is also significant there were a number of auxiliary cavalry forts in Northern Britain's – Ribchester, Chesters, for example, these were populated early by Spanish or Sarmatians, that presumably intermixed with the local population.

Response:

We agree with the reviewer that this is interesting, and the reasons to be wary of gender bias in interpretation. We have removed the sentence that was speculating about one individual's potential equestrianism. However, they are still from the context of a Roman military fortress site from the 1st-2nd century CE, so we have kept the two hypotheses of them either having local ancestry, or being non-local soldiers.

The sentences now read:

"However, two 1st to 2nd century CE burials from a Roman military fortress site in Austria (Klosterneuburg)⁵ carry ancestry which is currently indistinguishable from Iron Age/Roman populations of Britain, to the exclusion of other groups (qpWave cladality p-value = 0.11). While one option is that they had ancestry from Britain, an alternative is that currently unsampled populations from western continental Europe carried ancestries similar to Iron Age southern Britain."

267 – is this date secure? I thought there was no date for this person so they could be 2-4th century – also the way you talk about them highlights the inaccuracy of the label on fig 3, the gladiator interpretation is contested. It is interesting that this individual might have EIA Scandinavian ancestry – a result which confirms the point made in ref 56 where they are identified as probably CNE foederati, so this point is valid but entirely too strong, the presence was not missed as suggested but there is a difference between mobility in a few individuals and a 'substantial influx'. It is also unclear how a Gladiator 'condemned to die' would contribute significantly to the gene pool – many gladiator contests were staged and not fatal as Gladiators themselves were too valuable to kill in this way.

Moreover, what does this new data say about the CNE ancestry identified in ref 37 how much of that is Scandinavian? Esp in sites like Lakenheath in Suffolk where Swedish ancestry is implied in the historical sources (the Wuffings for example).

Given lines 351-356 this would be important to address.

Response:

Thank you for pointing this out, we have updated the date to "second to fourth century CE".

We have changed all figure labels to no longer refer to "gladiator", and have added "Although it is uncertain whether this individual was a Gladiator or soldier, [...]" to the text. We agree that this is a highly interesting question about CNE ancestry, but we think more data from Roman and early Medieval Britain will be needed to address it.

285-288 – the chronology and group terminology become conflated in a confusing way. Are Iron Age groups genetic groups similar to Iron Age groups, or chronological?

Response:

We have tried to clarify this section and its terminology. This section now reads

“The observation of a shift in ancestry in Denmark cannot be confounded by potentially earlier unknown gene flow into Iron Age source groups in Austria, France, and Germany, but such gene flow could affect the exact ancestry proportions.”

283-299 – Iron age groups from Austria, France or Germany. Moving into Scandinavia in 500-800CE and afterwards (see point above) it would be very interesting to know the genetic similarity between these AFG groups, and the CNE ancestry seen in ref 37 can the AFG and Scandinavian ancestry be separated if present?

Response:

Currently, we don't see a clear connection of this influx of 'AFG' ancestry, as the reviewer terms it, and the CNE ancestry present in Britain. However, this is complicated by it being relatively difficult to separate British Iron Age ancestry from AFG, and the individuals from Britain with CNE ancestry also have British Iron Age ancestry. We hope that additional ancient genomes in the future will allow this to be addressed.

This section is very important, potentially identifying Northern European Migration not attested in the contemporary literature (or at least not recognised) as that literature is very limited at this period. It would be useful to connect it broadly to the wider issue migration issue from these places the Lombards and CNE ancestries identified before in this paper.

Response:

We completely agree, and mention this find as the major empirical advance in our paper e.g. in the abstract.

301-306 needs expanding many deposits are so vague as to meaningless.

Response:

Thank you, we have expanded this sentence to read

“Interestingly, in southern Scandinavia, the archaeological record has yielded considerable evidence for periodic conflict and socio-political stress during the pre-Roman and Roman Iron Ages, as seen most explicitly in the large quantities of weaponry and human remains found deposited in bogs at sites such as Hjortspring, Kragehul, Vimose, Alken Enge, and Illerup Ådal in Denmark⁶⁵⁻⁶⁷.”

323-328 – what do you mean by continental-related or ‘continental’-related ancestry and line 347 continental ancestry. Scandinavia is continental Europe and are these three different things or the same thing?

Response:

We are referring to what the reviewer has above termed Iron Age 'AFG' ancestry (Austria-France-Germany), which we have struggled a bit to find a shorthand label for. We are now referring to this ancestry in the text as ‘Iron Age central European-related’, or variations thereof.

Also when describing Sweden 28 out of 74, but Norway 2 out of how many from Norway? And central Sweden ‘relatively few’. Meaning? Are the authors trying to hide the relatively few samples used in this study or is it just vague and slightly opaque? As with 301-306 many deposits, this needs tightening up.

Response:

We have expanded this to state how many individuals from Norway were tested (n = 24 individuals), and that almost no individuals in Central Sweden show this ancestry.

“Almost no such individuals are noted in eastern Central Sweden, which was a focus of regional power of the Svear (Figure 4a).”

348-349 – after the Oland massacre and thus unconnected or was this part of the impact of migration? Along with the 324 – a map more clearly defining this ancestry and its impact would be helpful here. Fig 4b is useful but large amounts of data are presented and not discussed.

Response:

It is indeed a fascinating idea that the Oland massacre was a consequence of upheaval associated with the ancestry change we see. We have discussed this before, but with the reviewer's comment we have now added a sentence

" Indeed, one speculative hypothesis is that the massacre at Sandby Borg could represent conflict associated with movements of people which contributed to later ancestry change, although other scenarios are possible and further synthesis of biomolecular and archaeological data is necessary to test this hypothesis."

We have removed Fig 4b as we believe that the same information is captured by other panels of this Figure.

354 – sure Iron Age Britain, but by the Viking age Britain is a mixture of Iron Age and CNE ancestry, so that IA ancestry may only describe the west of Britain. What about the rest? Could this presence in Norway be detected because of Norway's association with Ireland, NW England and Scotland, rather than the British ancestry which would include England. The Dane Law is in Southern and Eastern England so it is surprising not to see more – is it visible?. Although you

go on to this in lines up to 361 this is unclear purchase instead of Britian – Eastern England, and REF the relevant publications.

How close is CNE 9REF 35) and Scandinavian ancestry. And 358 what is British-Related ancestry there is no such genetic group?

Response:

Most individuals in early medieval Britain were the product of contacts and had a proportion of ancestry that would indeed appear as Iron Age Britain in our analyses. Such Early Medieval English could be represented as individuals with partial ancestry matching Iron Age Britain. However, several individuals have almost all their ancestry related to Iron Age Britain. We have tried to clarify throughout that we refer to "Iron Age British ancestry" instead of "British ancestry".

368-371 i don't think Imperial Roman and Portugal can be take as representative of southern Europe as a whole so tighten up this expression.

Response:

We have removed mention of Imperial Rome and Portugal here to not cause confusion.

376 – do you mean in males via the Y chromosome, and therefore in male ancestry?

Response:

No, we mean autosomal genomic ancestry found in male individuals. We have added "male individuals" to the sentence to make it more clear.

389-392 – great but you have not explained these sites or told us they are execution/massacres.

Response:

Thank you, we have expanded this text to describe the sites briefly.

“In Britain, most of the individuals recovered from the two late Viking Age mass graves identified at Ridgeway Hill, Dorset, and St. John's College, Oxford⁶ show ancestries typical of those seen in Viking Age southern Scandinavia. The violent manner of their death, which at Ridgeway Hill was characterised by formal execution and at Oxford by combat or a massacre, indicates that those killed could have been predominantly migrants or members of Viking raiding parties from Scandinavia (Figure 4f). “

410 – there is no such thing as Anglo-Saxon related ancestry. English early Medieval perhaps?

Response:

We have indeed changed this to "early English".

415-417 – So an EIA Scandinavian ancestry, British Iron Age ancestry, and an abundance of continental ancestry – the Viking Age burials included a mixture of people with different ancestries – so there is no such thing as a genetic Viking but rather these were drawn from the diversity of people represents in Sweden or Norway? – this is an important point but the way this is presented it is not quite made clear.

Response:

Thank you, yes we think that it is possible that the ancestry profiles we observe at the Staraya Ladoga site in western Russia, which was an important trading town in the Viking age, were 'drawn' from the diversity of people represented in Sweden or Norway at the time. However further sampling would be required to strengthen this hypothesis. We certainly agree that there is no such thing as a genetic Viking.

We have amended the text as follows:

"The relative absence of Iron Age central European ancestry, which was largely restricted to southern Scandinavia during the Viking Age, is thus inconclusive but indicative that these individuals originated in the central/northern parts of Sweden or Norway, where we observe similar ancestry profiles in the Viking Age."

And Finally given the point about AFG ancestry and its impact on Scandinavian genetics and the indistinguishability of English early medieval ancestry (as opposed to British Iron Age) from AFG ancestry, as well as the missing British (overall) ancestry that has been seen in previous studies it would be important to figure how to remove the geographic influence on the terminology so that it could be discussed in a way that dose not imply that this migration might originate only in he places identified in the genetic terminology i.e Austria, France and Germany.

Response:

We completely agree about this. We think it will be important to expand the samples of ancient genomes, to pin down closer the geography of these migrations. At the moment, it is hard for us to coin particular labels. We are now referring to this ancestry as 'CentralEurope.IronRoman' and have added the following sentence to the manuscript

"The geographic origin of this ancestry is currently difficult to discern, as the available samples from Iron Age Central Europe remain sparse."

Response to reviews

Referee #1 (Remarks to the Author):

The authors have addressed my comments very well, and I appreciate the thorough explanations given. I am fully satisfied that this paper is ready for publication in Nature, and I am excited to see it in print.

Referee #1 (Remarks on code availability):

I have lightly reviewed the code. While I am not an R expert (or even basically competent) the package layout looks like it follows R standards, and I believe it should be straightforward to install. The code looks well factored and readable.

Thank you very much for the detailed comments and we are pleased to hear about the positive evaluation.

Referee #2 (Remarks to the Author):

I appreciate the efforts of the author's to respond to my critiques. Clearly they have thought deeply about them.

The new isolation-by-distance simulations presented in Ext Fig 3F are very positive, and I am glad that they have such power to reject a pulse model when there is an IBD pattern. However, as far as I can tell, these simulations do not specifically get at the question of what the power of the method is when there is a pulse of migration on top of a background IBD pattern, which are the likely dynamics of the migration period. For example how would the model fair if after establishing the 1-D stepping stone model, there was a pulse of admixture from 1 into 4, and then a continuation of constant 1-d migration? It would be useful to see that the method was robust to such a process. However, the new simulations are still a great addition to the paper.

We agree that the suggested simulations have added genuinely important insight into the behaviour of *qpAdm* and *Twigstats* in scenarios of continuous migration between populations. Thank you for these suggestions.

We have now conducted an additional simulation where nine populations are organised on a 1d grid, with bidirectional migration occurring between adjacent populations from 1000 generations ago to today.

We updated Extended Data Fig 4d with this new simulation (with no additional mixture events), and as before, find that *qpAdm* accurately captures the underlying relationships between these populations.

We have then added a punctual mixture event from population 1 into population 7 occurring 10 generations ago, such that population 7 received 50% gene flow at this time from population 1.

We then modelled population 7 in a rotational *qpAdm*. Whenever a model had infeasible admixture proportions (i.e. not falling within [0,1]), we set the corresponding p-value to 0. The plot below shows the proportion of simulations (out of 20 replicates), for which each combination of source groups received the highest p-value, as well as the median $-\log_{10}$ p-value across these 20 replicates. We can see that *Twigstats* increases power by many fold, and in most cases, a three-way mixture of $p_1 + p_6 + p_8$ or $p_0 + p_6 + p_8$ is chosen as the fitting model. This accurately reflects the genetic history of population 7.

Given space constraints, we have opted to not include this simulation in our current manuscript, but believe that there is a lot of scope to explore the behaviour of *qpAdm* and *Twigstats* in similar simulation models in future.

While I appreciate the additional work to define the source populations, it still seems problematic to me from an interpretation perspective. That clusters from *qpwave* fall in close proximity in an MDS is not really a surprise for me. What is essentially being described is the former is a particularly close genetic relationship, which then get recapitulated in the latter. I do not argue that the samples *qpwave* finds are not genetically coherent groups in terms of being highly similar to each other. However, if anything the MDS shows for Central European populations the overlap and proximity of some of these groups genetically, and then overlaying this on the map some very awkward geographic relations. So my same questions largely remain from my initial

review in terms of how we then interpret an individual whose ancestry is determined to come from these groups. But I suppose if the authors remain fairly ambiguous and careful when an individual draws their ancestry from a “Central European” group then I guess this is ok, but not particularly ideal or powerful.

Thank you for these comments. We were careful not to further pinpoint the exact geographic source of this ancestry based on the sampling locations of these individuals. Our observations of large-scale ancestry shifts e.g. in southern Scandinavia are robust to this.

Regarding the appropriateness for Nature and the author’s response, while I still maintain (without taking away from the quality of the work) my position that I personally do not believe this method is as great an advance as the authors purport in their response compared to other comparative pop gen methods, this is ultimately of course highly subjective and an editorial decision, and certainly would not begrudge or be disappointed if the paper were to appear in the journal if the editor decided.

The results still also read as a series of vignettes, though worded in a more appropriate way thanks to reviewer 3 in particular. While the authors describe changes in ancestry, the results still do not really get down to the process underlying these migrations. The take home message for the Germanic migration sections is essentially there have been different levels of germanic migrations in different parts of Europe, which I still feel is somewhat superficial. Yes, there are some new evidence of migration in Scandinavia not observed before, but that migrations have taken place during the medieval period has never really been in question. What has been of debate are the forces driving these migrations, the mechanistic processes by which they took place, their size, their impact and the legacy, not just with regard to modern genetic ancestry, but the development of nation states and their culture. The findings of this paper, though interesting do not really get at these facets of history. In addition it is not clear to what extent the results are evidence of large scale migration, perhaps establishing permanent settlements, versus small scale mobility of for example military units or elites. So much of understanding of what these ancestry profiles mean in the wider realm of migration versus mobility depends on the sampling strategy and intensity and interpreting the archaeological context of the samples (these are not just random samples from a particular place and time).

I am not a fan of the new title. It gives the impression that prior to this study, we did not believe there was any migration in medieval Europe, and the genetic analysis has finally revealed it to us. I also do not see evidence of “migration waves”. Migration waves implies there are periods of high migration and low migration (peaks and troughs of high and low migration rates). But this study does not really characterize these dynamics through time, they just offer instances where ancestry changes based on sample availability

On reflection we propose the revised and shortened title of: "High-resolution genomic history of early medieval Europe", but welcome the editor's thoughts as well.

Finally, I understand and appreciate the huge work that has gone into the study and the desire to show all the results as much as possible and not relegate work to a largely unread 200 page supplementary text as has commonly been the norm in ancient DNA. However, I think the figures need some revisions. The various panels are often just so small. I had to spend a long time zooming in on specific sections to see them, and then trying to cross reference that with the legend. It made it prohibitively difficult and time consuming to cross reference the text and figures, and in an actual print copy many of them would probably require a magnifying glass to see. It would be better if some of the sub-panels were transformed into their own figures. Almost every figure bar Extended data figure 2 had some section that was difficult to read and required a lot of zooming, though extended figures 3-5 are probably the worst culprits.

Thank you for pointing this out, we have now increased font sizes in all of our figures and hope that these are now easier to read.

Referee #3 (Remarks to the Author):

Thank you for your patience, summers always a juggle with family and digging commitments.

The authors have made a very good and detailed response to the comments, they have expanded and explained, removed inaccurate or misleading terms and increased the detail for key archaeological examples in a useful way. The paper is easier to follow, and more precise, its contribution - via new statistical approaches and the results that have been flagged up in their samples are important and suitable for publication in Nature. Being able to use aDNA as statistical sources without modern populations is very important, and the case studies highlight differences in migration patterns, and unknown migrations that will be debated and explored for many years behind the small samples presented here.

I'd be very happy to read this paper in Nature, and look forward to it being published.

We are pleased to hear about the positive feedback and thank you very much for the detailed comments throughout this review process.